# Energy flows reveal declining ecosystem functions by animals across Africa

Ty Loft[1,2✉], Imma Oliveras Menor[1,2], Nicola Stevens[1,3,4], Robert Beyer[5], Hayley S. Clements[6,7], Luca Santini[8], Seth Thomas[9,10], Joseph A. Tobias[11] & Yadvinder Malhi[1,4]

A key challenge for ecological science is to understand how biodiversity loss is changing ecosystem structure and function at scales that are relevant for policy[1]. Almost all biodiversity metrics are challenging to disaggregate into animal-mediated ecosystem functions such as pollination, seed and nutrient dispersal, and predation. Here we adopt an ecosystem energetics approach[2] as a physically meaningful method of translating animal species composition into a suite of ecosystem functions. Drawing on new datasets that estimate biodiversity intactness and species population densities[3–5], we quantify historical changes to energy flows through mammal- and bird-mediated ecosystem functions across sub-Saharan Africa. In total, trophic energy flows have decreased by more than one-third. The pattern of decreasing function varies by historical biome, driven by arboreal birds and primates in forests, terrestrial herbivores in grassy systems, and burrowing mammals in arid systems. Functions performed by megafauna in particular have collapsed outside protected areas. Compared with other biodiversity metrics, an energetics approach highlights the ecological importance of smaller animals and keystone species. The results can help practitioners conserve and restore functionally diverse, energetically intact ecosystems across land uses and biomes. By relating biodiversity intactness to energy and material flows, ecosystem energetics can also advance efforts to integrate animal-driven functions into biosphere and earth system models, helping us to understand possible regional or planetary boundaries[6] for biodiversity.

Ecologists have devised multiple metrics to track species loss and recovery[7–9], but alone these metrics can be poor indicators of changes in ecological function[10]. The influence of a species on its ecosystem depends on the species' abundance and on the specific functions that it performs[11]. To assess how changing biodiversity affects ecosystem function at large scales, ecologists must develop consistent methodologies that account for species' changing abundances and their diverse impacts on ecosystems. Doing so is central to predicting how biodiversity change affects the ability of ecosystems to provide services, such as storing carbon, supporting food production and buffering natural disasters[12].

Many ecosystem functions are moderated by animals, yet most ecosystem function literature that addresses regional or larger scales focuses exclusively on vegetation functions such as carbon storage or water cycle modification. Animals perform multiple functions including herbivory, seed and nutrient dispersal, and predation[13,14], which shape ecosystems by controlling species abundances and flows and patterns of carbon, nutrients and water[15,16]. Since the pre-industrial Holocene epoch, animal populations have declined as intensively human-modified landscapes have expanded fivefold[8,17]. Tracking how biodiversity loss alters animal-moderated ecosystem functions is challenging because species respond unevenly to land use change depending on their traits[18]. For example, agricultural conversion depletes populations of large or frugivorous animals faster than small and omnivorous ones[18,19]. These asymmetric population declines change the trophic structures of ecosystems: the partitioning of energy and biomass between plant and animal guilds. Ecosystems with simplified trophic structures provide a reduced range of functions and services[20]. Ultimately, they become less capable of recovering from external shocks and supporting human wellbeing and livelihoods[21].

To track species abundances (a prerequisite for measuring functional changes) conservationists have developed biodiversity intactness indices (BIIs)[3,22,23]. These indices estimate how human activity has changed the richness or abundance of species relative to remaining highly intact landscapes such as wilderness areas, which are assumed to be representative of historical animal abundances, nominally in the pre-colonial and pre-industrial period. Local intactness scores can be aggregated to determine BIIs across ecoregions, countries and taxonomic groups. BIIs have been proposed as a metric for biodiversity in the planetary boundaries framework, which seeks to identify safe

[1]Environmental Change Institute, School of Geography and the Environment, University of Oxford, Oxford, UK. [2]AMAP, Université de Montpellier, CIRAD, CNRS, INRAE, IRD, Montpellier, France. [3]Center for African Ecology, University of the Witwatersrand, Johannesburg, South Africa. [4]Leverhulme Centre for Nature Recovery, University of Oxford, Oxford, UK. [5]Department of Zoology, University of Cambridge, Cambridge, UK. [6]Centre for Sustainability Transitions, Stellenbosch University, Stellenbosch, South Africa. [7]Helsinki Lab of Interdisciplinary Conservation Science, Department of Geosciences and Geography, University of Helsinki, Helsinki, Finland. [8]Department of Biology and Biotechnologies "Charles Darwin", Sapienza University of Rome, Rome, Italy. [9]Department of Biology, University of Oxford, Oxford, UK. [10]Nature-based Solutions Initiative, Department of Biology, University of Oxford, Oxford, UK. [11]Department of Life Sciences, Imperial College London, Ascot, UK. ✉e-mail: tyloft25@gmail.com

environmental conditions for human societies[24]; although BIIs were more recently abandoned because of the difficulty of relating them to ecosystem function[6]. Metrics such as BIIs, which aggregate changes to species abundances or richness, cannot estimate changes to ecosystem function because they weight each species equally. In reality, some species disproportionately affect ecosystem function owing to their population densities, body sizes, dietary preferences, rates of food consumption and behavioural features[13]. To quantify how the changing animal populations estimated by BIIs alter ecosystem function, an approach is needed that accounts for species' variable ecosystem impacts using a common unit of measurement[25].

One option is to adopt an ecosystem energetics approach to compare how much energy species consume across land uses[2,25]. Such an approach quantifies energy flow through the trophic web by calculating the annual food energy consumed by each species per unit area. Species within an ecosystem can be classified into functional groups. Changes to energy flows through these groups indicate changes to the provision of associated ecosystem functions[16,26]. This approach relies on the ubiquity of energy: in all terrestrial ecosystems, energy, captured as sunlight by plants, flows up the trophic web through guilds of herbivores, carnivores, scavengers and detritivores. Compared with abundance and richness-based assessments of biodiversity loss, an energetics approach has three advantages: (1) it weights species impacts based on the ecologically meaningful metric of food consumption; (2) the common currency of energy enables functions performed by different taxonomic or functional groups of species to be quantitatively compared across time and space; and (3) energy flows can be quantitatively related to earth system processes such as changing net primary productivity and carbon cycling.

Although energetics approaches have been previously used to measure how human-modified landscapes alter ecosystem trophic structure[2,27], they have not been scaled beyond a few model ecosystems, as they require extensive data measuring species abundances across different land uses. Energetics approaches have never been applied at regional or continental scales. Here we focus on sub-Saharan Africa as a case study, owing to the region's wide range of ecosystems, large gradients in ecological intactness and (in some areas) relatively intact megafauna communities, which provide an opportunity to assess a wider range of animal-mediated ecosystem functions than elsewhere. We take advantage of new datasets that: (1) model population densities of bird and mammal species[4,5]; and (2) estimate BIIs, or the impacts of land use changes on species abundances, across sub-Saharan Africa[3,28]. The BII estimates are derived from a new dataset that aggregates 30,000 expert estimates of how African species abundances respond to land use change. We use these datasets to quantify how biodiversity loss has degraded a suite of ecosystem functions across sub-Saharan Africa.

## Approach

Our approach is to quantify how human modification of land uses has changed the distribution of energy among trophic guilds and functional groups. Energy flows are calculated for African bird and mammal species under contemporary versus pre-colonial and pre-industrial (approximately 1700 CE) conditions, which we refer to hereafter as historical conditions. Energy flows are aggregated across biomes (mapped before land use change) to compare how the dominant vegetation structures in Africa moderate the ecological impacts of land use change. To clarify the broad relationships between biodiversity loss and ecosystem functionality, we also assess how well biodiversity intactness predicts the intactness of energy flows through specific ecosystem functions and the entire community of birds and mammals, as well as whether energy flows can be predicted from species richness. We focus on birds and mammals because they are important components of animal biomass[29], as well as data-rich groups with well-understood

ecological functions, while acknowledging that invertebrates have a major but data-challenged role in ecosystem energetics. Although populations of some species, especially megafauna, declined substantially long before the colonial and industrial period, these declines appear to have been less severe in Africa than elsewhere, thus Africa's contemporary association with megafauna[30,31]. Africa therefore provides the unusual opportunity to examine how human activities have changed ecosystem function in historically near-intact ecosystems.

To estimate energy flows through African ecosystems, we calculated historical energy consumption by each species present in each 8 km × 8 km cell across sub-Saharan Africa (317,000 cells in total). We used (1) modelled average species population densities[4,5] based on around 10,000 averaged empirical estimates of species population densities; and (2) habitat-adjusted International Union for Conservation of Nature (IUCN) range maps[32,33] to predict historical species abundances in each cell. To calculate the average absolute energy consumption of sub-Saharan Africa's approximately 3,000 bird and mammal species, we used published allometric equations[34] and datasets on species traits, diets and food assimilation efficiencies (Supplementary Tables 1–3 and Supplementary Data 2). We quantified current energy flows according to the remaining abundance of each species in each cell estimated by the BII[3,28]. To calculate the energy flows through animals performing ecosystem functions, we grouped species into trophic guilds and functional groups, based on their diets, lifestyles, body sizes and, for mammals, social group sizes. Using these categories, we identified 23 unique ecosystem functions (11 for birds and 12 for mammals), which we aggregated across classes into 10 major functions[14,31,35,36] (Extended Data Table 1). These include both consumption functions directly correlated with energy intake including granivory, carnivory, browsing, grazing and insectivory, and behavioural functions such as seed and nutrient dispersal, pollination and soil disturbance, for which energy intake is a potentially less accurate proxy but can still usefully indicate changes in a function's strength. For each functional group, we calculated the absolute historical and current energy flows in kJ m$^{-2}$ year$^{-1}$, as well as the proportional energetic intactness, defined as the percentage of historical energy flows through a group of species remaining in an ecosystem. In all cases, we estimated changes to energy flows through groups of birds or mammals performing a function (for example, granivory and seed dispersal), rather than to the function itself, and we refer to changes in functions themselves only as a shorthand. Because uncertainty around energy flows decline as a greater number of species are analysed (Methods), we report and discuss guild-level rather than species-level results.

## Changing trophic structure and function

Changes to the total flow of energy through animal populations and to its distribution among guilds, can alter ecosystem functionality[20]. Energy flow through food consumption by wild African birds and mammals has decreased to 64% of historical values (54–74%; all ranges reported are 95% confidence intervals). Energy flow decreased most in high intensity land uses, falling to 27% (18–35%) of historical levels in settlements, 41% (30–53%) in croplands, 67% (56–76%) in unprotected untransformed lands (comprising rangelands and near-natural lands) and 88% (81–96%) in strict protected areas (Fig. 1). Conversion to croplands and cities was responsible for 25% of the total decline in energy flows. In aggregate, birds were more resilient to land use change than mammals, with the fraction of energy flowing through birds (as opposed to mammals) increasing from 37% to 41%. The greater reported decline of mammals was driven entirely by the collapse of large herbivores (including grazers, browsers and frugivores), which historically accounted for over one-quarter of mammalian energy consumption. Energy flows through large herbivores decreased by 72% (61–85%), compared with a decrease of 29% (19–38%) for other mammals and 29% (20–38%) for birds. Large herbivorous mammals

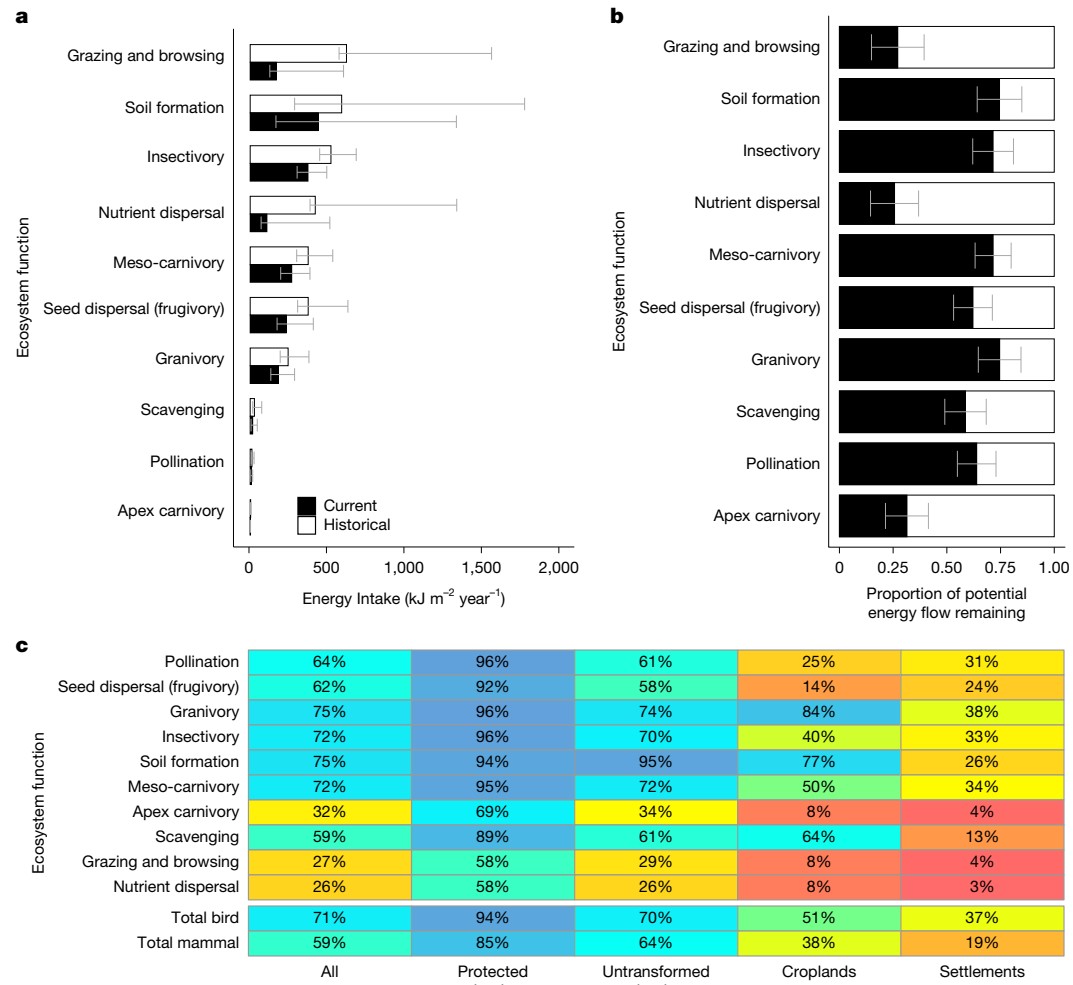

**Fig. 1 | Intactness of energy flows through birds and mammals performing key ecosystem functions.** Species were allocated to functional groups based on diet, lifestyle, body size and group size. The intactness of each ecosystem function indicates the intactness of energy flows through the birds and mammals that perform the ecosystem functions. **a**, Total absolute historical (white) and modern (black) mean energy flow through ten bird and mammal functional cohorts across sub-Saharan African. Bars denote mean values with 95% confidence intervals derived from 10,000 Monte Carlo simulation estimates incorporating uncertainty in body mass, population density, the daily energy expenditure equation, assimilation efficiency of different food types, composition of the diet of each species, and the biodiversity intactness of each species within each land use. **b**, Average energy flow through modern sub-Saharan Africa (black) as a proportion of historical energy flow (white). Error bars denote 95% confidence intervals derived from 10,000 Monte Carlo simulation estimates of the biodiversity intactness of each species within each land use. **c**, Average remaining proportion of pre-industrial energy flows through birds, mammals and each functional group, across sub-Saharan Africa and within its predominant land uses. Cells with higher blue intensity indicate more intact functional groups, and those with higher red intensity indicate less intact groups.

have undergone substantial population declines even in protected areas (Fig. 1). Therefore, estimated energy flows have fallen well below historical levels even in Africa's relatively wilder regions.

Partitioning energy transfer by habitat and broad ecological niches, we find that energetic intactness has collapsed across all biomes, (which were mapped based on their historical occurrences before land use change). Total (bird and mammal) energy flows are estimated to be 63% (54–73%) intact in historical grassy systems, 65% (57–76%) intact in historical forests and 69% (60–80%) intact in historical arid systems. Despite the similar magnitude of these declines, the trophic guilds that are most responsible for energy loss varied according to each biome's dominant feeding pathways, suggesting that biome moderates changes to function (Fig. 2). Arboreal species account for significantly more energy flow in historical forests, where they can take advantage of greater vertical space and habitat complexity[37] (Fig. 2). Of these arboreal species, birds make up nearly half of energy flow despite their much smaller body sizes than most primates and other arboreal mammals. Overall, reduced arboreal species populations

accounted for 37% of energy decline in forests, compared with 10% and 5% of energy decline in grassy and arid systems, respectively. Fossorial species account for a large but highly uncertain proportion of energy flow in arid systems, perhaps because burrowing helps animals regulate temperature and conserve moisture[38] (Extended Data Fig. 5). Fossorial mammals accounted for 26% of energy decline in historical arid systems, versus 10% and 3% of energy decline in grassy systems and forests, respectively; however, the highly uncertain abundances of fossorial mammals means that this should be analysed cautiously. Large terrestrial herbivorous mammals were major contributors to energy decline (29–36%) in all historical biomes. However, the total fraction of loss attributable to terrestrial birds and mammals was notably lower in forests (50%) than in arid (61%) and grassy (68%) systems.

When we consider the proportion of total energy flow contributed by broad trophic guilds, we find that overall patterns have changed little over time, with flows through herbivorous birds and mammals (including leaf-, grass-, seed-, fruit- and nectar-eating animals) falling from 64% to 61% of total energy flow, flows through insectivores

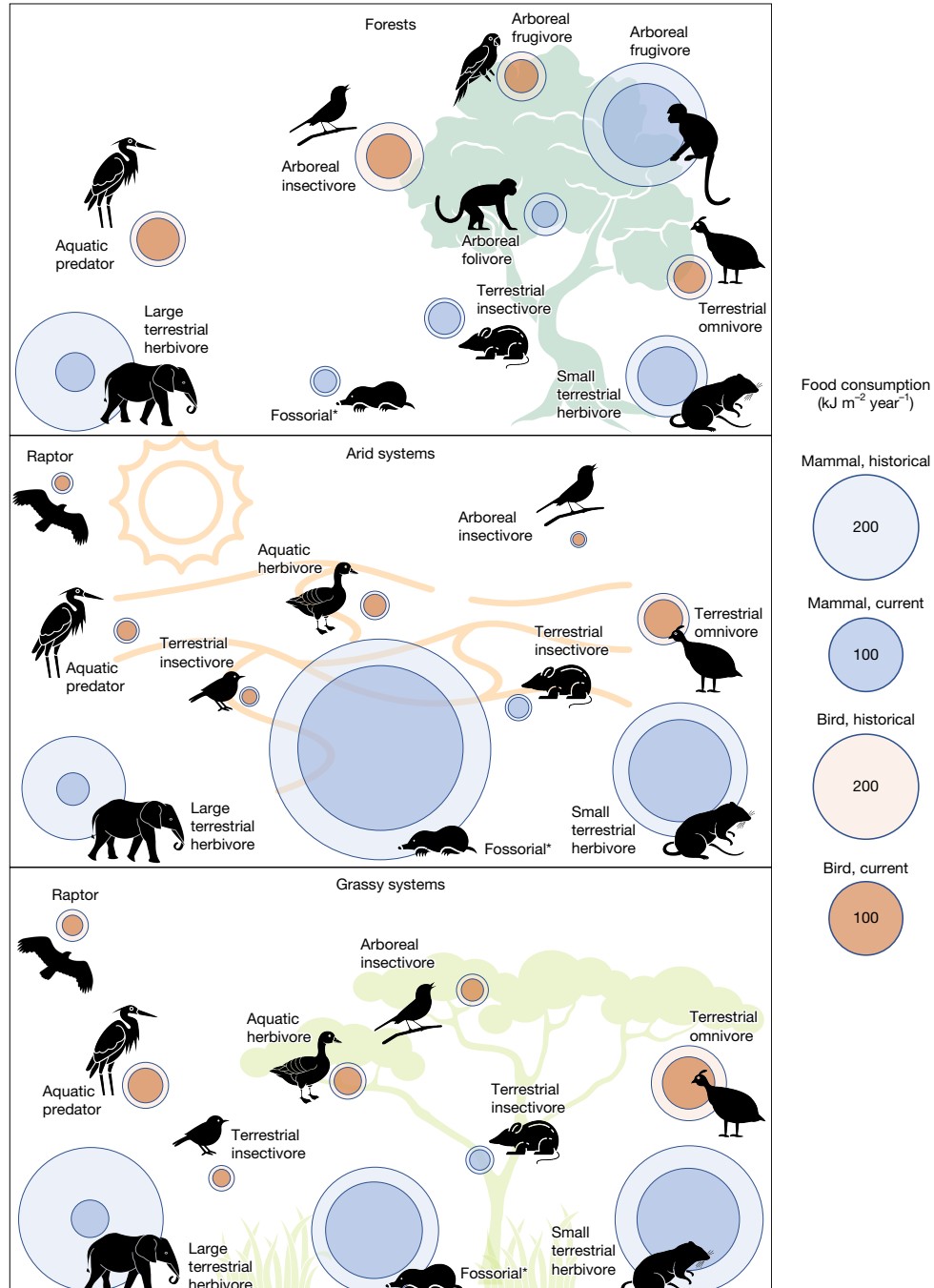

**Fig. 2 | Mean magnitude of energy flows through bird and mammal trophic guilds across African biomes.** Values represent the mean energy flow through all cells given historical species abundances (light shading) and current species abundances accounting for human land use (dark shading). The size of the circle indicates the magnitude of food energy consumption by species in animal guilds. Flows through present day ecosystems were calculated using the mean intactness of bird and mammal groups according to their BIIs. For clarity, guilds with small energetic flows are not shown, and complete results as well as uncertainty are available in Supplementary Data 1. The asterisk associated with the fossorial mammals indicates the high uncertainty around energy flows through that trophic guild.

increasing from 12% to 14%, and flows through carnivores remaining at 4%. However, the intactness of energy flows through functional groups varied widely, ranging from 26% (14–37%) intact for nutrient dispersal by large herbivores to 75% (64–85%) intact for granivory, when averaged across sub-Saharan Africa (Fig. 1b).

Energy flows through megafauna-dominated functional groups, which include nutrient dispersers, grazers, browsers and apex carnivores, were notably less intact (from 26–32%) than other ecosystem function groups (from 59–75% intact), and were twice as depleted in non-protected untransformed lands as in strict protected areas (Fig. 1c).

Megafauna have a key role in controlling vegetation structure, both directly through grazing, browsing and nutrient deposition by large herbivores, and indirectly through the control of herbivore populations by apex carnivores[13,36]. Given that 80% of Africa is unprotected untransformed land, the collapse of large herbivores and carnivores is probably altering vegetation on a continental scale[36,39]. Megafauna extinctions on other continents have been estimated to have reduced mammal herbivory[26] by 42% and lateral nutrient flow[16] by more than 90%. Although some of this lost functionality is substituted by domestic herbivores, total (domestic and wild) herbivore biomass

has decreased across most of Africa, as has the functional diversity of herbivore guilds[40].

Beyond megafauna decline, untransformed lands are relatively functional, with other ecosystem functions persisting above 55% intactness. However, the large absolute decreases in energy flows through arboreal guilds in historical forests, shared roughly equally between birds and mammals, translates to notable absolute declines in seed dispersal by frugivores outside of protected areas (Extended Data Fig. 2). Flows through seed dispersers are only 58% (50–66%) intact in untransformed lands and 14% (10–18%) intact in croplands. Flows through pollinators, dominated among vertebrates by birds and bats, also decline notably outside of protected areas, and are 63% (53–70%) intact in untransformed lands and 25% (17–32%) intact in croplands. Seed dispersal and pollination strongly influence plant community composition, vegetation structure and biomass. In tropical forests, primate and megafauna-dispersed tree species tend to have higher biomass than other species[41]. Reduced seed dispersal is likely to change the vegetation structure of defaunated forests and hinder ecosystem recovery[42].

The most energetically intact functional groups are dominated by small and mid-sized herbivores, which account for a large proportion of energy in all biomes. These functional groups include granivores, soil disturbers and avian grazers (mostly by water birds) (Extended Data Fig. 1). Energy flows through soil disturbers are potentially more intact in rangelands (95% intact, 82–107%) than in protected lands (94% intact, 85–103%). The sole ecosystem function to become amplified after land use change was avian granivory, which reached 108% (84–135%) of historical levels in croplands, where birds can take advantage of seed-rich crops (Extended Data Fig. 5). These relatively stable functions can generate ecosystem disservices when they harm agricultural production.

The resilience of guilds and functions dominated by small birds and mammals (defined as <3 kg)[43] has created a striking pattern: African ecosystems are becoming dominated by smaller species. On average, the proportion of energy consumed by small birds and mammals increased from 69% of total energy to 78% of total energy, whereas the proportion consumed by megafauna (>65 kg) fell from 16% to 7% (Extended Data Table 2). Moreover, even in intact systems, rodents and passerines account for a much larger proportion of energy flow than of biomass, owing to their high energy consumption per unit mass[2]. Even as the absolute energy flows through small animals have marginally decreased, their proportion of total energy flow has increased: rodents accounted for 31% of total historical energy consumption but only 17% of biomass, increasing to 36% of energy and 24% biomass today. Passerine birds accounted for 8% of historical energy consumption but only 2% of biomass, the same proportions as today. As large herbivores decline, these smaller animals are likely to exert greater relative control over the flows of nutrients, water and material that structure their ecosystems' vegetation, while not compensating for the attributes (such as large seed dispersal and greater daily transport ranges) that are particular to larger animals.

The species that contribute most to total energy flow are elephants (family Elephantidae), which historically accounted for 16% (9–26%) of total bird and mammal biomass across the region and 10% (6–26%) of total energy flow across sub-Saharan Africa. Savanna elephants (*Loxodonta africana*) historically consumed by far the most energy of any single species. Elephants perform the grazing and browsing and nutrient dispersal functions; however, the confidence intervals around functions performed by elephants are highly uncertain, owing to their disproportionate energy consumption: uncertainty around energy flow values are lower for guilds and functions that have more species and more even energy consumption among species (Methods). The high energy flows through elephants supports prior findings that they are the dominant large herbivore in Africa[44] and a keystone species with the potential to change ecosystem vegetation at the landscape scale and to affect continental-scale carbon sequestration[45]. Elephants and their close relatives were also widespread and abundant across Eurasia

and the Americas until the Late Pleistocene megafaunal extinctions[1]; our finding indicates just how much of terrestrial vertebrate energy flow and associated functions once occurred through elephants. Our results also indicate that fossorial rodents, particularly mole rats, may consume an outsized proportion of energy, owing to their high abundance and high food consumption per unit body mass (Extended Data Fig. 6). However, energy flows through fossorial rodents, and through their associated soil disturbance function, are highly uncertain as confidence intervals around modelled mole rat population densities span 2.5 orders of magnitude[4]. There is some evidence that fossorial rodents provide important belowground ecosystem functions[46], but in general their role in ecosystems is difficult to measure and poorly understood, (as indicated by their high uncertainty in this study) and is a ripe subject for further investigation.

## Biodiversity and ecosystem energetics

Understanding how changes to biodiversity intactness alter an ecosystem's suite of functions can help guide conservation and restoration. Biodiversity intactness equally weights changes to species abundances, whereas energetic intactness weights changes to ecosystem functions based on each associated species' energy consumption. The ability of biodiversity intactness to predict changes to ecosystem functions varies widely among functions (Extended Data Fig. 1). For some functions, BII is a strong proxy. These tend to be functions that are performed by a large number of species, and which are not dominated by a few keystone species. They include avian aquatic carnivory, avian terrestrial and perching insectivory, avian granivory, avian grazing and mammalian insectivory. For other functions, BII is a noisy proxy, such that biodiversity intactness accurately predicts energetic intactness when averaged across Africa, but inaccurately predicts energetic intactness in any given cell. These functions include those dominated by arboreal species (avian and mammalian seed dispersal, canopy engineering, and ecosystem engineering and cavity creation), as well as mammalian granivory, mammalian scavenging and pollination. Pollination in particularly unpredictable: in landscapes in which BIIs are 50% intact, energy flows through pollinators range from 22–75% intact. For still other functions BII poorly predicts even overall energetic intactness. BIIs substantially overestimate the intactness of the megafauna-performed functions: grazing, browsing, nutrient cycling, megaherbivore impacts and apex carnivory. BIIs also underestimate the intactness of avian aerial insectivory, which is predominantly performed by swifts, martins, and swallows. At the ecosystem scale, biodiversity intactness is a strong predictor of total energetic intactness for birds ($R^2 = 0.97$), and a weaker predictor for mammals ($R^2 = 0.86$) (Fig. 3a,b). The difference for mammals is driven by changes to large herbivore populations, which account for 16% of historical energy consumption, but just 3% of BIIs. Because large herbivores are 32% more intact within protected areas than outside them, protected areas conserve a higher proportion of energy flow (and therefore ecological functionality) than of biodiversity intactness. Thus, whereas BIIs can reasonably predict total energy flow, they cannot predict which ecosystem functions remain intact and which have been depleted. This is a key knowledge gap for practitioners who are working to conserve and restore ecosystem functionality that is uniquely addressed by our approach.

In addition to assessing intactness (Extended Data Fig. 8), an energetics approach can be used to estimate the absolute historical magnitude of energy flows through African landscapes (Fig. 3). Large absolute energy flows supported by a rich diversity of species and guilds can indicate exceptionally vibrant ecosystems. One key question for clarifying the causes of functional resilience is whether species richness predicts absolute energy flows through animals, as it does other aspects of ecosystem function, including net primary productivity[47]. We found that the species richness–energy relationship differed between

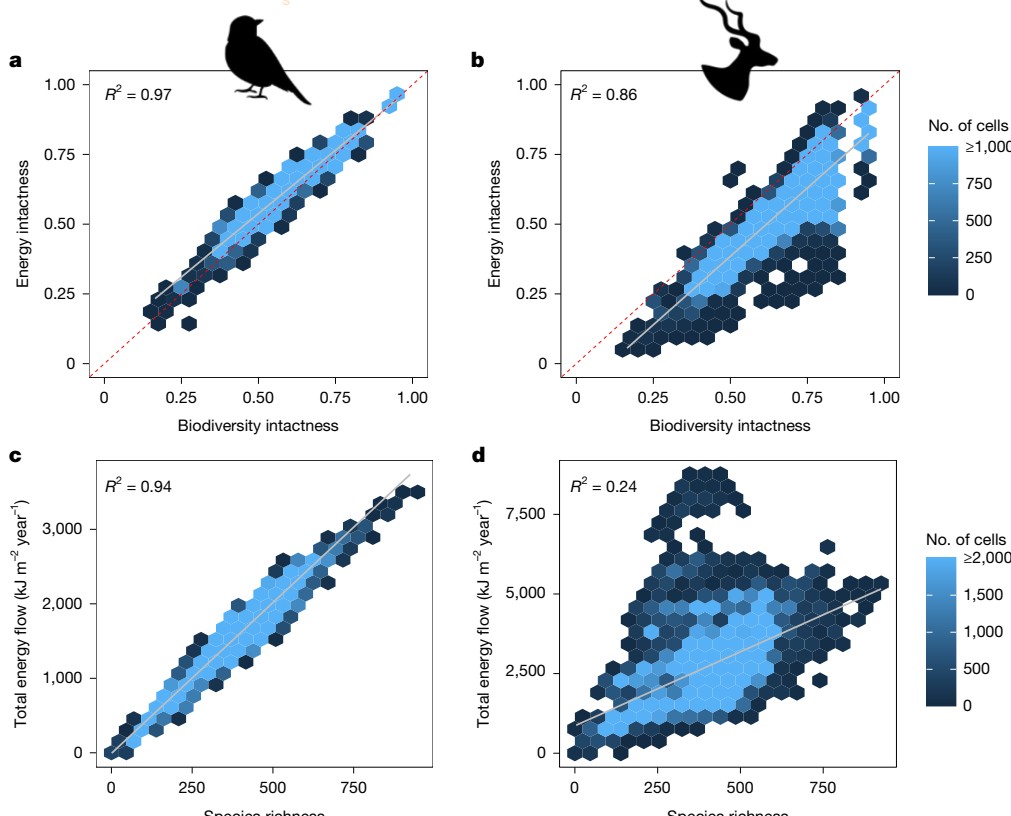

**Fig. 3 | Relationships between biodiversity intactness, energetic intactness, total energy flow and species richness. a–d,** Relationships were calculated for each of the approximately 317,000 cells that compose sub-Saharan Africa, and colour indicates the number of cells with a given relationship. $R^2$ values were calculated using linear regression. **a,b,** Relationships between energetic intactness and biodiversity intactness were calculated over each cell for birds (**a**) and mammals (**b**). Energetic intactness indicates the proportion of historical pre-modern energy flow remaining under modern landscapes, with species weighted on the basis of their contribution to energy flow. Biodiversity intactness indicates the changes in species population abundances between pre-modern and modern land uses, with each species weighted equally. **c,d,** Relationships between species richness and total energy flow were calculated over each cell for birds (**c**) and mammals (**d**). Species richness for each cell was calculated by summing species occurrences based on IUCN range maps.

birds and mammals (Fig. 3c,d). Richness predicts class-wide energy consumption strongly for birds ($r^2 = 0.92$), but weakly for mammals ($R^2 = 0.15$). This discrepancy is caused by the lower richness and less even apportionment of energy consumption among mammal species. Mammals account for 64% (57–75%) of total historical energy flows but just 36% of species, meaning that proportionally more energy is consumed by the average mammal than bird species. In addition, the 5% most energy-consuming mammals consumed 72% (64–84%) of energy flows through mammals, whereas the 5% most energy-consuming birds consumed only 48% (41–59%) of flows through birds. By contrast, the bottom 50% of mammals consumed just 0.7% (0.4–1%) of mammal-mediated flows, whereas the bottom 50% of birds consumed 7% (4–10%) of bird-mediated flows. These differences exists because species-level energy flows per unit area span a greater range of values for mammals (4.3 orders of magnitude) than for birds (2.1 orders of magnitude), owing to mammal species' wider ranges of population densities, body masses and assimilation efficiencies. Mammalian energy consumption thus appears to be more driven by the presence of keystone consumer species. These include large herbivores, especially elephants, highly abundant rodents, and primates, which dominate the arboreal guilds important in forests (Extended Data Fig. 3). The uneven allocation of energy among species has implications for the biodiversity–function relationship: bird-driven functions are likely to be far more resilient to biodiversity loss, as they are supplied more evenly by a wider range of species.

There are also clear biogeographical patterns in absolute historical energy flows. For birds, these tend to be highest in East Africa (Fig. 4a),

where high-energy landscapes broadly overlap with regions of volcanic soils and moderate to high rainfall along the Great Rift Valley. Birds and the insects that many birds consume may benefit from nutrient-rich vegetation growing on fertile soils. Absolute historical energy flows through birds tend to be lowest in Africa's arid regions: the southwest, the Sahel and the Horn of Africa. The map of historical absolute energy flow through mammals is more difficult to interpret, owing to the high uncertainty associated with the energy consumption of dominant species (Fig. 4b). Still, mapping absolute energy flows across sub-Saharan Africa opens up a number of interesting questions about the biogeographical factors that control total energy abundance. For example, future research might explore why birds consume a greater fraction of total energy in forests than in arid systems, or whether climatic and soil variables predict how energy flows are distributed among trophic guilds, taxonomic groups or ecosystem functions.

## Restoration and biodiversity assessments

As we advance through the United Nations Decade on Ecosystem Restoration (2021–2030), energy flows can contribute to ongoing efforts[48] to meet urgent demand from governmental and corporate sectors for metrics that can set and track progress towards nature restoration targets. In particular, energetics provides a novel and useful approach to quantitatively compare the restoration of ecosystem functions in recovering ecosystems in which species composition has changed substantially. When ecosystems recover in human-dominated landscapes, the allocation of biomass among remaining species can change substantially[49].

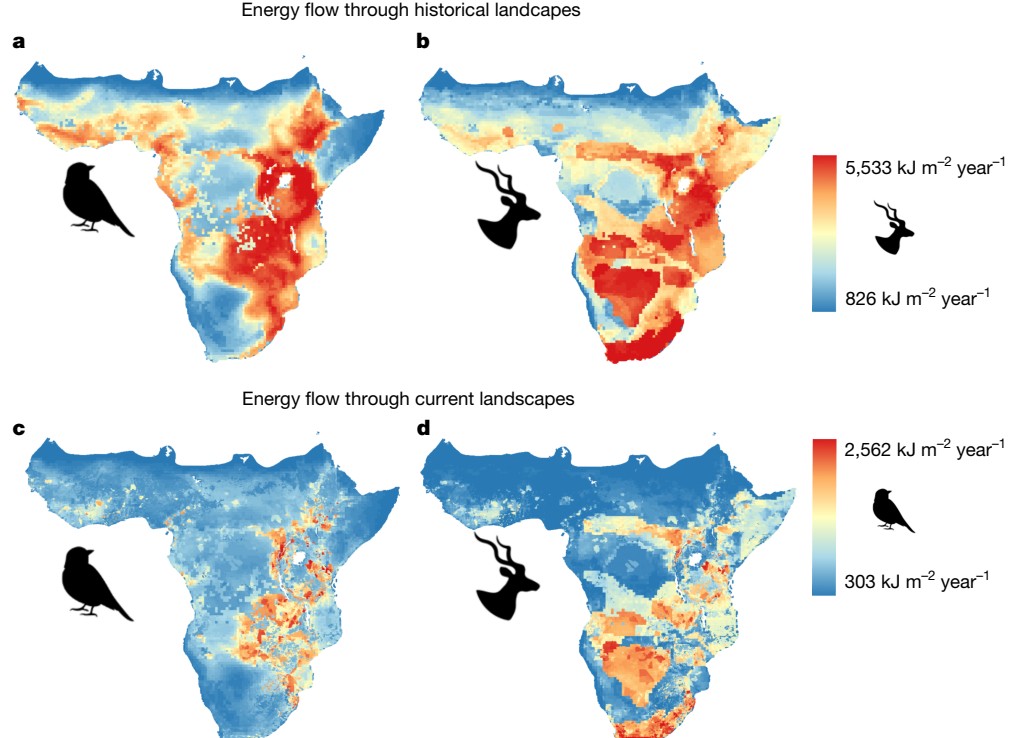

**Fig. 4 | Aggregate energy flow through birds and mammals mapped across sub-Saharan Africa. a,b,** To map historical energy flows, energy flows were summed for each bird (**a**) and mammal (**b**) species occurring in each 0.5° cell, assuming that they existed in pre-industrial population densities. **c,d,** To map energy flows through modern, transformed landscapes for birds (**c**) and mammals (**d**), historical energy flows for each species were multiplied by the BII value of each species, which is calculated independently for each species in each landscape based on the species response group and the cell's modern land use class and intensity. Colour gradients indicating aggregate energy flows were scaled independently for birds and for mammals.

In addition, extirpated or extinct species often become replaced with functionally similar substitutes through migration, inadvertent introduction and active rewilding[50]. Change in species abundances and composition can alter the strength of animal-performed ecosystem functions in an ecosystem, even after biomass and species richness recover to historical levels. Mapping ecosystem energy flows allows organizations to quantify the intactness of an ecosystem's trophic guilds and functions, independent of species composition. This information can complement richness or abundance-based metrics, such as the various BIIs, which cannot directly translate species composition to function. Once functional changes are mapped, practitioners can focus on restoring the most energetically depleted guilds across land uses and biomes to restore ecosystem function and trophic complexity. For example, amplifying energy flows through depleted arboreal birds and primate guilds might be prioritized in forests, where seed dispersers consume a large proportion of energy and are important for ecosystem resilience[42]. By comparing energy flows between these guilds in intact and recovering landscapes, practitioners can set quantitative benchmarks for restoring functions, independent of species composition.

Another application of energetics approaches to studying biodiversity loss is to advance global biodiversity assessments. In particular, ecosystem energetics can provide metrics that track how animal-mediated ecosystem functions are changing. The novel scale of this study, which expands previous plot-based energetics analyses to an area of more than 20 million km², allows these metrics to be integrated into global assessments of biodiversity loss. For example, energy-based metrics of ecosystem change can complement the richness and abundance-based metrics that currently underpin assessments by the Intergovernmental Science Policy Platform on Biodiversity and Ecosystem Services[21]. Ecosystem function metrics are additionally valuable because they can serve as an intermediate step that links changing biodiversity intactness to changing ecosystems services[21]. Energetics can also improve

efforts to integrate biodiversity loss into the planetary boundaries framework[6]. It is contested whether a planetary boundary for biodiversity is meaningful, because of the heterogeneous, local and spatially disconnected nature of ecological functions[12]; boundaries and thresholds may be more meaningful at local scales. Biodiversity intactness was proposed as a metric for such a planetary boundary[24], although recent planetary boundaries literature has abandoned BIIs as a metric because of the difficulty of relating them to ecological functions, favouring human appropriation of net primary productivity (HANPP) instead[6] (Supplementary Fig. 1) However, it is challenging to see how HANPP can be mechanistically related to actual declines in ecological function, beyond a broadly positive relationship between two indicators of human impact. The energetics approach that we have outlined here shows a way forward that enables biodiversity and its intactness to be related to a suite of ecosystem functions, whether at local, regional or planetary scales.

## Caveats

Like any other biodiversity metric produced by modelling, energy flows have a number of caveats in addition to their advantages. Relying on energy consumption alone as a metric does not capture the intrinsic value of rare and/or native species, or the ecological impacts of species that translate energy into function exceptionally efficiently or that perform unusual functions that are not easily captured by their traits. High energy flows through a guild or function do not necessarily indicate that ecosystems are functional, stable or resilient. The diversity of species contributing to an ecosystem function also matters. When energy flows through a function are dominated by a single species, the increase in function is likely to be accompanied by a substantial increase in instability, as individual species are more subject to sudden shifts in abundance or extirpation than are groups of species. This caveat

is crucial for restoration: when assessing the recovery of a function, practitioners should evaluate not only the strength of an energy flow but also the diversity of contributing species.

There is also evidence that some forms of anthropogenic disturbance can amplify animal energy consumption[2] at local scales, even as land use change has degraded biodiversity intactness and ecosystem function at the sub-continental scale. It is not sufficient to assume that land use change degrades overall ecosystem functionality or individual ecosystem functions. Here we find that some species and functions do better in rangelands and croplands than in protected areas. Logging has previously been found to amplify vertebrate energy consumption in some tropical forests by increasing vegetation palatability and accessibility to herbivores[2]. Positive relationships between some functions and forms of human modification also seem likely in disturbance-dependent grassy ecosystems. The broad patterns revealed in regional and continental-scale analyses should be tested and refined through empirical plot-based studies. Researchers can use energetics approaches to quantitatively test which types of disturbance maintain (or even enhance) biodiversity intactness, trophic complexity, or ecosystem function.

In addition, the large-scale approach used here is built by combining a number of datasets, each of which relies on assumptions and simplifications. There are therefore caveats associated with each dataset used, namely the species population density estimates, the range maps, the BII, the data on species traits and allometric equations, and the process used to sort species into functional groups. These methodological caveats are systematically stated in the caveats section of the Methods section and should guide application. In broad terms, the large-scale approach used here does not capture many kinds of local and regional variation in historical species population densities and intactness values, limiting its application at local scales. In addition, the estimates are highly uncertain for individual species, but decline as more species are analysed in a guild, allowing us to report reasonable uncertainties around guild-level energy consumption.

## Conclusion

This analysis has demonstrated how an energetics approach can quantify the decline and recovery of ecological functions mediated by birds and mammals. It fills an important gap: although it is increasingly clear that animals shape landscapes, biomes and the Earth system in important ways, their functions have been inadequately quantified over large spatial scales and have not been integrated into Earth system models. As a tool to estimate changes to animal-mediated ecosystem function, the energetics approach presented here provides at least three advantages over approaches that consider species richness or abundances alone. These advantages clarify ecosystem function at the scale of the species, the ecosystem and the Earth system. Together, they provide a framework to relate changes to animal-mediated ecosystem functions across these scales.

First, instead of weighting species equally, an energetics approach weights species based on food consumption, an ecologically meaningful indicator of a species' ecological importance within a landscape. Identifying each species' trophic importance can help reveal where keystone species contribute disproportionately to various ecosystem functions, and how the importance of key species varies across biomes, land uses and functions. The energetics lens used here also reveals the importance of smaller species, including birds and rodents, which have higher metabolic rates and are important contributors to energy flows and many ecosystem functions, but which other mass-based metrics often underrepresent owing to their minor contribution to biomass.

Second, as a common currency, energy can be used to quantitatively compare how changes in species composition alter the strength of ecosystem function across land uses and biomes. For example, the distribution of energy among species differs between arid, grassy and forested biomes, causing these biomes to have very different suites of functions even where overall species richness or animal biomass are similar. By contrast, metrics that aggregate changes in species richness or abundances are unable to compare the variable pathways through which biodiversity loss alters functionality, pathways that are moderated by the species composition of an ecosystem and the vulnerability of those species to various kinds of land use change.

Third, energetics provides a mechanism for bringing animal activity into the quantitative, mechanistic framework of biosphere modelling and earth system science, which to date has been dominated by the ecological functions provided by vegetation and is largely blind to the functions provided or modified by animals. Using energetics, changes in animal populations can be related to Earth system processes by estimating how animals affect vegetation structure, both directly through grazing and browsing, and indirectly through seed and nutrient dispersal, insectivory (of herbivorous insects) and soil disturbance. Changes in herbivory, insectivory or nutrient dispersal by birds and mammal can be related to changes in net primary productivity, and from there to the effects of changing vegetation on albedo, fire regimes or carbon cycling. These analyses rely on translating biodiversity intactness into energy flows, as energetics can be physically related to Earth system processes, whereas species richness, abundance and biomass alone cannot.

Like any metric (for example, carbon stock), energy flows should be considered as only one lens on the multifaceted nature and value of biodiversity and ecological function. In particular, energy flow needs to remain coupled with consideration of the number of species contributing to energy flow to avoid wrongly labelling the persistence of a few generalist species as ecological intactness. Some future steps could include integrating energy flows into global biodiversity assessments, expanding this energetic analysis to a planetary scale, incorporating domesticated animals within the same framework, extending to the much more data-challenged question of invertebrate energetics, and relating animal energy flows to vegetation structure captured through vegetation plots and through dynamic global vegetation models. We believe that the sub-continental-scale analysis presented here presents an important step forward in the challenge of relating animal biological richness and intactness to large-scale ecological and planetary function.

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

# Methods

## Study scope

The geographical scope of the study was sub-Saharan Africa, defined as comprising ecoregions[51] in the Afrotropic realm within continental Africa. We calculated energy flows for the 1,088 mammal and 1,955 bird species for which data were available, composing 98% of total African species excluding seabirds. Energy flows were calculated independently for each 8 km × 8 km grid cell, the scale at which biodiversity intactness data are available. The study area comprises ~317,000 cells. To assess change over time, energy flows were calculated twice for each cell: once based on estimated historical species abundances in the pre-industrial and pre-colonial Holocene (approximately 1700 CE), and once based on contemporary abundances, given human land use, according to the population changes estimated by the BII[28]. A visual overview of the methods is presented in Extended Data Fig. 5.

## Historical species abundances

To determine which bird and mammal species were historically present in each 8 km × 8 km grid cell, we used historical IUCN/Birdlife range maps, adjusted for species habitat requirements following ref. 52. For each species, the initial IUCN/Birdlife range maps were divided into 1/12° grid cells, and grid cells were excluded from the species range when the cell's historic, natural biome did not match the species' habitat requirements, as documented by the IUCN. The 1/12° grid cells were then aggregated into 1/2° cells, and cells were included if they contained any available habitat. For a few small-range species, 1/2° habitat-adjusted maps eliminated all available range due to their exclusion from 1/12° cells, so unadjusted IUCN/Birdlife ranges were used. For the 11 large mammal species for which historical range maps are not available within the IUCN database, we adapted vector maps from other sources, following ref. 44, and then applied the gridded habitat filter detailed in ref. 52 (Supplementary Information).

To calculate historical species abundances, we used published median population density estimates for bird[5] and mammal[4] species. These were modelled as a function of trait, environmental, and phylogenetic predictors, using additive mixed-effect models and Bayesian inference, based on ~36,000 empirical records of bird and mammal population densities across 737 mammal and 1,853 bird species. Population density estimates for species lacking empirical data were extrapolated using the model. To estimate species abundances across sub-Saharan Africa, we used mean species population densities[2]. Mean densities were calculated using log-normal distributions based on published median densities and uncertainty intervals. Because population density distributions for most species are left-skewed, mean species population densities are higher than median values for species with wide confidence intervals. Given that ~75% of the global terrestrial surface is modified by humans to some extent[53], the exclusion of non-natural population densities is not realistically possible, and is not necessarily desirable given that hominids have modified African species population densities for millions of years.

## Contemporary species abundances from the BII

To estimate contemporary species abundances we multiplied historical abundances by the proportional intactness of each species in each 8 km × 8 km cell under modern land use. We used the intactness values for species under various land uses that are published in the BII for Africa dataset[3]. The BII employs a structured expert elicitation process to estimate and validate the proportional changes to species abundances under nine land uses: strict protected areas, near-natural lands, rangelands, intensive croplands, smallholder croplands, tree croplands, timber plantations, dense settlements and urban areas. The BII allocates each species into one of 17 bird and 76 mammal 'response groups' containing species that respond similarly to land use change. The average impact of each land use class on the abundance of species in each response group was calculated from ~30,000 individual estimates produced by 200 experts on African biodiversity. To map changes in abundance, each cell was assigned a land use class and intensity according to the land use classification outlined by ref. 28. Cells within protected areas and timber plantations were classified categorically, and cells within croplands, rangelands, and settlements were classified and then scaled along a land use intensity gradient. In cases where land use change benefited a species, the intactness of that species was greater than 1, and its abundance increased compared to the historical baseline.

## Daily energy expenditure and food uptake

To calculate ecosystem energy flows, we first calculated the short-term equilibrium rate of food consumption for each species following ref. 2. For each species, daily energy expenditure was calculated from body mass using multi-species allometric equations[34] (Extended Data Fig. 7; see Supplementary Table 1 for equations). Food consumption was calculated from energy expenditure based on published assimilation efficiency values for each food type and taxonomic group of birds and mammals (Supplementary Table 2). Where available, assimilation efficiency values were assigned at the family level; otherwise they were assigned at the order or class level. Values for the body mass of each species, and for the composition of food types within each species' diet, were derived from the Elton Traits database for mammals[54] and from the Avonet database for birds[55]. Energetic food intake was calculated in units of $kJ\,m^{-2}\,year^{-1}$ and then averaged across cells.

## Allocation of species into trophic guilds and functional groups

To understand how human land use has altered ecosystem trophic structure, we allocated species into trophic (that is, feeding) guilds. Each species was allocated to a single trophic guild, to shed light on how an ecosystem's trophic structure, defined as the distribution of energy among guilds, varies between biomes and land uses. Species were allocated to guilds based on their taxonomic class, their diet (for example, omnivore, carnivore, nectarivore, folivore or frugivore), and their lifestyle (for example, arboreal or terrestrial). Data on diet and lifestyle were extracted from the Elton Traits database for mammals[54] and from the Avonet database for birds[55]. Throughout the text, herbivore is used as an umbrella term to capture species eating any kind of plant matter, including foliage, seeds and nuts, nectar and fruit. The terms folivore, granivore, nectarivore and frugivore are used to refer to these groups independently. In addition, large and small terrestrial herbivores were split according to a published list of African large herbivores[44] to better isolate how the distinctive vulnerability of large herbivores to human activity alters ecosystem trophic structure.

To understand how human land use has altered ecosystem function, we allocated species into 23 functional groups: 11 for birds and 12 for mammals. Species that perform multiple functions were allocated to multiple groups, so that the sum of energy flows through functional groups is greater than the total flow through the ecosystem's birds and mammals. By contrast, the energy flows through guilds sum to the total energy flow through birds and mammals. We adapted a list of 11 bird functions from a published list of major avian ecosystem functions[14]. We added a function for aquatic carnivory and subdivided the invertivory function based on species lifestyle (for example, insessorial, aerial, terrestrial), as invertivory is performed by over half of all bird species. We sorted birds into functional groups based on their lifestyles and diets (see Extended Data Table 1 for sorting criteria for both birds and mammals). Unlike for birds, there is no single authoritative source on functions performed by mammals. After reviewing the literature we designated twelve mammal functions performed by large herbivores[13,31], carnivores[36,56], primates[35], bats[57], fossorial mammals[46] and other small mammals[58]. We sorted mammals into functional groups based on their diet, body mass and lifestyle. For

the grazing and browsing functions performed by large terrestrial herbivores we used published data on the leaf versus grass component of large herbivore diets[44], and included large, terrestrial, herbivorous primates (*Gorilla* spp. and *Theropithecus gelada*) based on the expert knowledge of the authors. We additionally used published data on herd size[44] to select herbivores that perform a nutrient dispersal function, as herd forming species have a distinctive effect on nutrient distribution within ecosystems[13]. We included in the megafauna impacts function those species that have unique ecological impacts because their large body size frees them from predation[31]. We determined the diet thresholds for each function iteratively, running the species allocation process multiple times and refining thresholds based on the authors' expert knowledge. To clarify our results in the main text, we further aggregated our 23 preliminary functions into 10 aggregate functions, some of which are performed by both birds and mammals (Extended Data Table 1).

## Comparison of energy flows across functions, biomes and land uses

To calculate energy flows through functions, we summed the energy flows through all species that contribute to each function. This approach weights the contributions of species to associated functions based on species' average daily energy consumption. The proportionate contribution of each species to its functions therefore changes depending on whether energy flows are calculated based on historical species abundances or based on present day, human impacted species abundances. Beyond energy flow, we did not scale species-level contributions to functions based on other metrics of functional efficiency: for example, based on pollen deposition rates, seed dispersal distance, or diet proportion. These causes of efficiency vary widely between functions and species[14] and are difficult to measure consistently. To avoid biases, we therefore assumed that all species use energy equally efficiently to perform their associated ecosystem functions. For the analysis, we compared energy flow within specific functions across space and time. It is not meaningful to compare energy flows across ecosystem functions (for example, predation vs soil disturbance) as how each function uses energy is very different.

We also calculated the average energy flows through functional groups and trophic guilds across biomes and land uses. The biome is commonly proposed as the appropriate unit of analysis for assessing biodiversity trends, because biomes are biologically coherent subunits of the biosphere with structures and functions that respond to land use change in relatively consistent ways[12]. We allocated cells into biomes based on the biome map of the RESOLVE Ecoregions dataset[51]. To allow for broad comparisons between vegetation types, we further aggregated biomes into forests, grassy systems comprising savannas and grasslands, and arid systems comprising deserts and shrublands. For the biomes analysis, we excluded cells falling into the fynbos and thicket biomes, which are not easily classifiable and make up less than 2% of sub-Saharan Africa. We also excluded cells falling into mosaic biomes, as the low accuracy of available continent-scale vegetation maps makes it infeasible to subdivide mosaics into component biomes within the study scope. We calculated average energy flows through each guild and functional group across each of these three aggregated biomes under historical conditions and under modern land use conditions.

We allocated cells into land uses using an adapted version of the 8 km × 8 km resolution African land use map created for the BII for Africa[28]. Following source[28], cells were allocated to four land uses: strict protected areas (IUCN categories I:III); settlements (>20% urban cover or a population density over 1,000 per km²); croplands (>20% crop cover); and unprotected untransformed land (remaining cells). We calculated average energy flows through each guild and functional group across each of these four land uses.

## Comparison of energy flows to biodiversity intactness and species richness values

To understand how well biodiversity intactness values predict functional intactness, we related the BII of each cell to the intactness of energy flows through each cell. We related the BII of birds and mammal species to the intactness of total energy flows through bird and mammal species (Fig. 4a,b) and to the intactness of energy flows through species in each functional group (Extended Data Fig. 1). We identified functional groups for which biodiversity intactness is a good proxy (slope approximately 1 and high $r^2$ value), a noisy proxy (slope approximately 1 and low $r^2$ value), and a poor proxy (a non-linear slope or a slope that is not near 1). Functional groups with shallower slopes maintain high levels of energy consumption as biodiversity intactness declines, and were deemed more resilient to human impacts. We also related total energy flows to native bird and mammal species richness, to understand the extent to which high-energy keystone species versus rich communities of species drive ecosystem function (Fig. 4c,d). We analysed these relationships across all cells using linear regression.

## Uncertainty calculation

Following ref. 2, we quantified uncertainty in our estimates of energetic intake by running 10,000 Monte Carlo simulations of energy flow through animal species and groups. For each simulation, we replaced the values in our original calculations with values drawn from random distributions. We assumed there was uncertainty in the following variables: species body mass, population density, daily energy expenditure equation (DEE), assimilation efficiency, and fractional diet composition of each species. Following ref. 3, we also assumed there was uncertainty in the estimated intactness of each species in each land use.

For body mass, we drew values from a truncated normal distribution (lower bound = 1 g) in which the mean was published mean body mass[54,55] and standard deviation was 15% as described in ref. 2 For population densities, we drew from a log-normal distribution, using mean and uncertainty values for each species published in refs. 4,5. For DEE, we estimated the 95% confidence intervals following the methods described in ref. 34. For assimilation efficiency, we drew from a random beta distribution using the mean and standard deviation by taxonomic group and food type in the literature (Supplementary Table 1). For diet composition, we drew from a symmetrical beta distribution with uncertainty parameters assigned following ref. 2. For the proportional intactness of species abundances in each land use, we drew from a random beta distribution using the mean intactness values and standard deviations published in ref. 3. Intactness values were previously validated in ref. 3 through a structured expert elicitation process.

The uncertainty in each of these variables captures the natural variability occurring within species among individuals and groups, as well as ecologists' uncertainty about mean values. For example, the population density of a given species will naturally vary geographically based on habitat suitability, resource availability, and competition. But there is also absolute uncertainty about the mean species population density of each species based on limitations on empirical data and model accuracy. This division of uncertainty into geographic and absolute components is true of the other variables as well. The uncertainty derived from natural variability decreases as there is an increasing number of analysed landscapes in which the species occurs. We assumed that half the uncertainty in species energy flow in a given landscape is from natural variability and that half is from absolute uncertainty about mean values, which does not decline as geographic area increases.

To account for the effects of area in our uncertainty estimates, we grouped species-level energy flows into 1° grid squares (-12,000 km² at the equator) following ref. 44. We treated uncertainty about natural variability as independent in each 1° square in which a given species occurs and drew from independent distributions in each square. For each species, we calculated range-wide spatial means of energy flow

for each of the 10,000 Monte Carlo simulations, and then propagated this area-scaled uncertainty into the absolute uncertainty about mean energy flow values generated from the area-independent Monte Carlo simulation estimates. We estimated total uncertainty by assuming uncertainty in all variables simultaneously, and calculated the 2.5th and 97.5th centiles intervals to derive 95% confidence intervals for our estimates.

## Caveats

There are a number of caveats to our analysis, which we present here systematically in the order of their associated dataset or analysis. To estimate historical species abundances, we use range-wide average population densities for each species. For the vast majority of 3,000 bird and mammal species we analysed, there are insufficient data to predict how population densities vary along environmental gradients. We assumed that using flat densities would not substantially alter our results, hypothesizing that the intra-species variation in population densities would even out when summing energy flows across tens, hundreds, or thousands of species. To test these assumptions, we calculated the declines in energy flows through the 92 large herbivore species for which spatially variable population densities are available over the whole of sub-Saharan Africa[44]. Using flat instead of variable densities changed the total energy flow through all 92 species by less than 1%, justifying our assumption (Supplementary Discussion). Another issue with our flat densities is that they do not account for intra-specific competition. It is expected that species reach higher densities when competitors are missing. The approach may therefore overestimate energy flows through species-rich guilds in species-rich cells, although this was not supported in our sensitivity test.

A second set of caveats regards our use of range maps. Because these range maps are coarse (0.5° grid cells), they can overestimate abundances of species restricted to specialist habitats. Energy flows through colonial species including some fossorial rodents and water birds may be overstated. Another challenge is that the accuracy of the range map polygons varies between species groups and subregions, with maps of well-known species and well-known regions better accounting for fine-scale habitat heterogeneity. As a result, both historical energy flows and declines may be overestimated in poorly known areas, where maps are likelier to include inappropriate habitat not occupied by species. By contrast, where maps for poorly known species do not include historical ranges, declines in energy flows may be underestimated.

A third set of caveats regards our data on species traits and allometric equations. Our analysis also assumes that species' diets, body masses and assimilation efficiencies do not vary consistently across land uses and biomes. However, a prior analysis of bird and mammal energetics across a land use gradient showed that shifts in diet had negligible effects on total energy flows. In addition, we accounted for substantial uncertainties around all three of these variables in our uncertainty analysis. A similar caveat is that the allometric equations, which we used to predict energy requirements based on species body sizes, do not account for environmental variables, for example the impact of temperature on energy needs. Consequently, the analysis may underestimate energy flows through cold regions, particularly afro-alpine and afro-temperate ecoregions. However, these regions make up a very small part of sub-Saharan Africa, which is overwhelmingly tropical or subtropical.

A fourth set of caveats regards the Biodiversity Intactness Index, used to estimate species responses to land use change. Because the BII averages responses from experts in different countries and regions, it does not account for how national political factors impact species abundances. These factors include war, protected area management, wildlife legislation, and cultural differences about hunting. The analysis may therefore overestimate energy flows in regions where unique national factors cause anomalously high overexploitation of wildlife, independent of land use transformation (and the converse where regions have anomalously low exploitation of wildlife such as taboos against bushmeat). In addition, only protected areas within IUCN categories I-III were designated as protected areas on the BII map. The analysis may therefore have overestimated energy intactness across the region's *de jure* protected areas and underestimated intactness in some de facto strictly protected areas, for example private reserves. This study analyses continent-wide average energy flows through guilds in different land use classes and biomes, which are less likely to be affected by national factors. However, an effort to use this approach to analyse energy flows over smaller areas (for example, within a country or protected area) would need to account for regional and national variables affecting species abundances.

Finally, a fifth set of caveats regards species' allocation into functional groups and the estimation of functional intactness. The study assumed that species contribute evenly to an ecosystem function, accounting for population density and allometry. For many functions, species were allocated based on their diets, after reaching a certain diet threshold (for example, 25% or 50%). The analysis may therefore overestimate absolute energy flows through functions performed by species that have a broader array of diets. However, this should not substantially affect comparisons of energy intactness within functions across time and space, the core aim of the study. In addition, for the vast majority of species and functions, there were insufficient data to estimate how variations in behaviour moderates the efficiency with which species use energy when performing functions. This caveat is less important for diet-based functions such as grazing, browsing, insectivory, granivore, and carnivory, where energy consumption by definition correlates closely with functionality. However the lack of data about behaviour may create more uncertainty for functions dependent on movement, such as seed dispersal, nutrient dispersal, and soil disturbance. The analysis may underestimate declines in these movement-dependent functions where species movements are highly constrained by habitat fragmentation (that is, in forests), even when landscapes and abundances remain relatively intact.

### Reporting summary

Further information on research design is available in the Nature Portfolio Reporting Summary linked to this article.

## Data availability

The data on energy flows through each species, trophic guild, and functional group are available in Supplementary Data 1 and 2. Input data on species population densities are available through the TetraDENSITY dataset (https://ecaslab.com/tetradensity-database/). Input data from the biodiversity intactness index are available through the BII4Africa project (https://bii4africa.org/). Input data on species ranges are available in the associated data from ref. 52 and from the IUCN database (https://www.iucnredlist.org/resources/spatial-data-download). Input data on species traits (that is, diet, body mass, lifestyle) are available through the Elton Traits database (mammals) (https://opentraits.org/datasets/elton-traits.html) and through the Avonet database (birds) (https://opentraits.org/datasets/avonet.html). Input data on ecoregions is available through the RESOLVE ecoregion dataset (https://developers.google.com/earth-engine/datasets/catalog/RESOLVE_ECOREGIONS_2017).

## Code availability

Data were analysed and visualized with a custom code using R (version 4.5.0 and earlier) and the following R packages: ggplot2, tidyverse, rgdal, readxl, sp, raster, terra, sf, lwgeom, magrittr and parallel. R Scripts as well as adapted data are available online at the project's Mendeley Data site (https://data.mendeley.com/datasets/myw63hks9b/1) or from the corresponding author upon reasonable request.

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

**Acknowledgements** T.L. was supported by a grant from the Clarendon Fund at the University of Oxford. H.S.C. was supported by a Jennifer Ward Oppenheimer Research Grant. We thank G. Hempson for helpful advice on part of the analysis.

**Author contributions** T.L. led the analysis, drafted the paper, and conceptualized the visualizations, with input and supervision from Y.M., N.S. and I.O.M., as well as input from H.S.C., L.S., S.T. and J.A.T. Y.M. conceived the analysis, developed the energetics approach and provided feedback on the analysis, visualization and structure of the paper. R.B. produced the habitat-adjusted species range maps. H.S.C. collected and supplied the data on biodiversity intactness and provided input on the designation of land uses. N.S. provided input on the designation of biomes, land uses and ecosystem functions. I.O.M. guided and provided input on the uncertainty analysis. L.S. collected, modelled and supplied the data on species population densities. S.T. aggregated data on bird and mammal ecosystem functions and worked on the ecosystem function classification. J.A.T. provided input on the ecosystem function classification as well as on the discussion of bird trophic guilds and ecosystem functions performed by birds. All authors commented on the draft.

**Competing interests** The authors declare no competing interests.

**Additional information**
**Correspondence and requests for materials** should be addressed to Ty Loft.

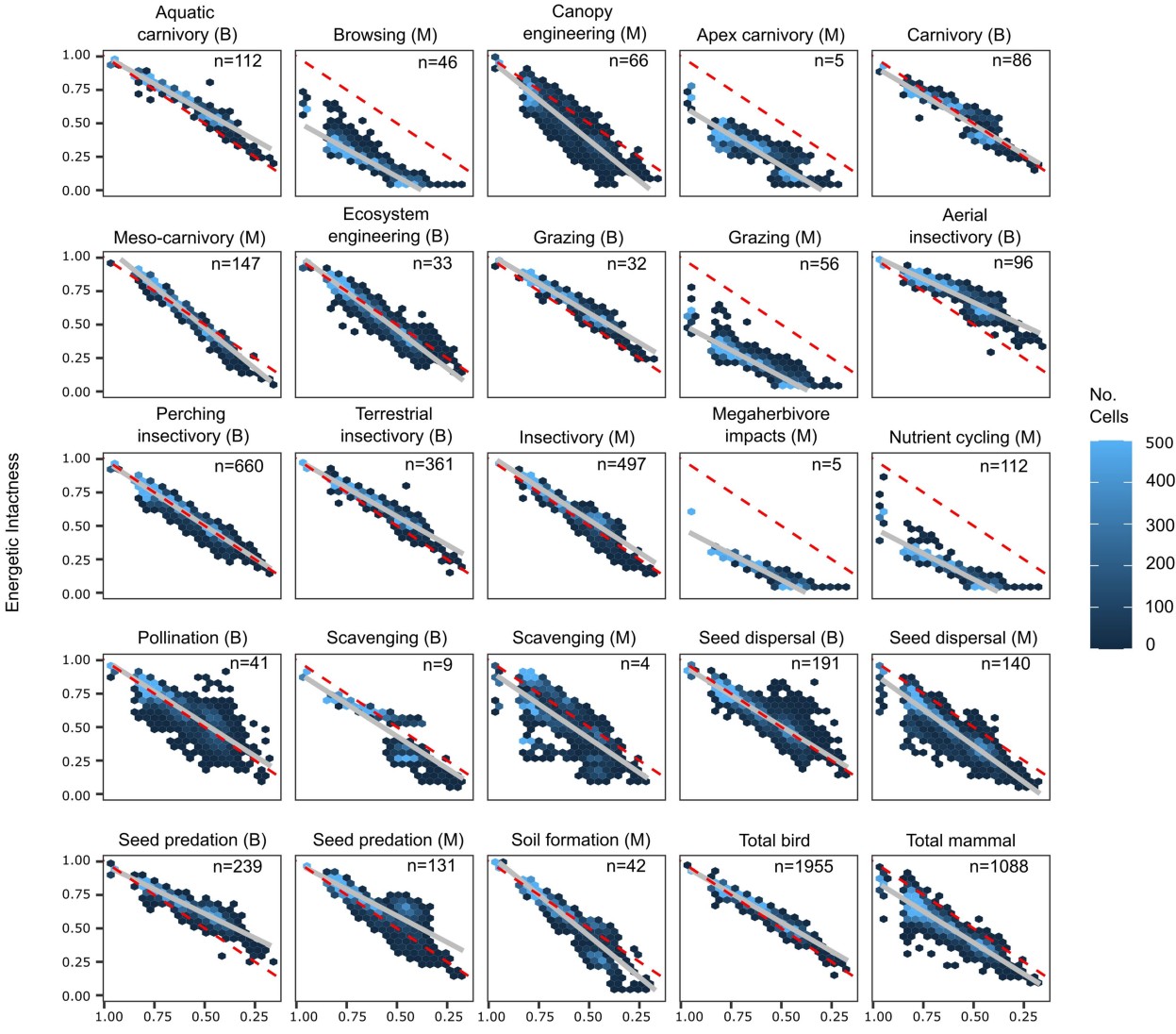

**Extended Data Fig. 1 | Relationships between declining biodiversity intactness (BII) and the intactness of energy flows through functional groups.** Relationships between total (bird and mammal) BII and energy flow intactness were calculated for each functional group in each 8 × 8 km² cell. The color of the hexagon indicates the number of cells with a given relationship between biodiversity intactness and energy flow intactness. Functions performed by birds are marked (B) and functions performed by mammals are marked (M). N refers to the number of species in each functional group. Gray lines signify the linear regression relationship between BII and energetic intactness, and the red dashed line indicates a reference slope of 1. BII refers to the changes in species abundances, with each bird and mammal species weighted equally.

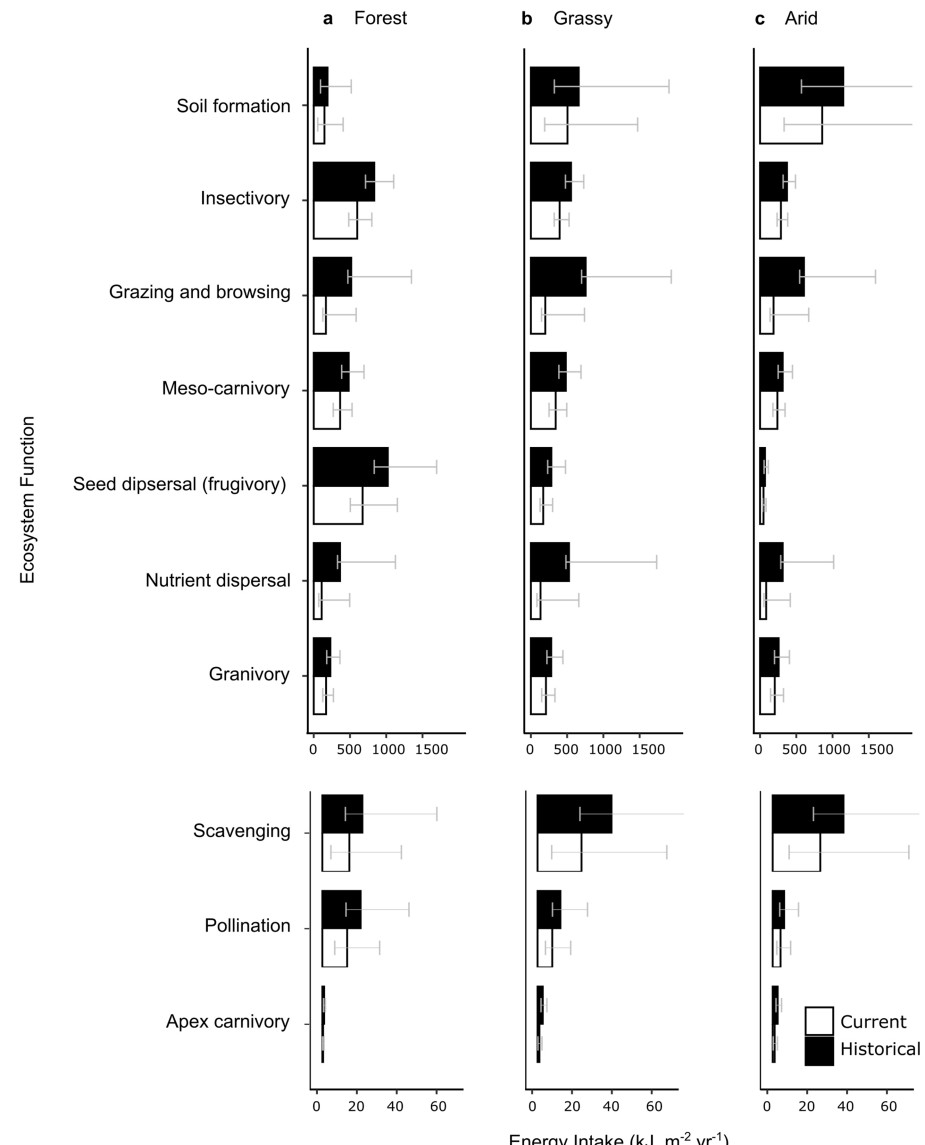

**Extended Data Fig. 2 | Intactness of energy flows through animal-mediated ecosystem functions across biomes.** Bars represent mean energy flows through functional groups for historical (black) and current (white) land uses within three groups of African biomes: forests (**a**), grassy systems including savannas and grasslands (**b**), and arid systems including deserts and shrublands (**c**). A separate scale is used for the scavenging, pollination, and apex carnivory functions. Error bars denote 95% confidence intervals derived from 10,000 monte-carlo simulations incorporating various sources of uncertainty.

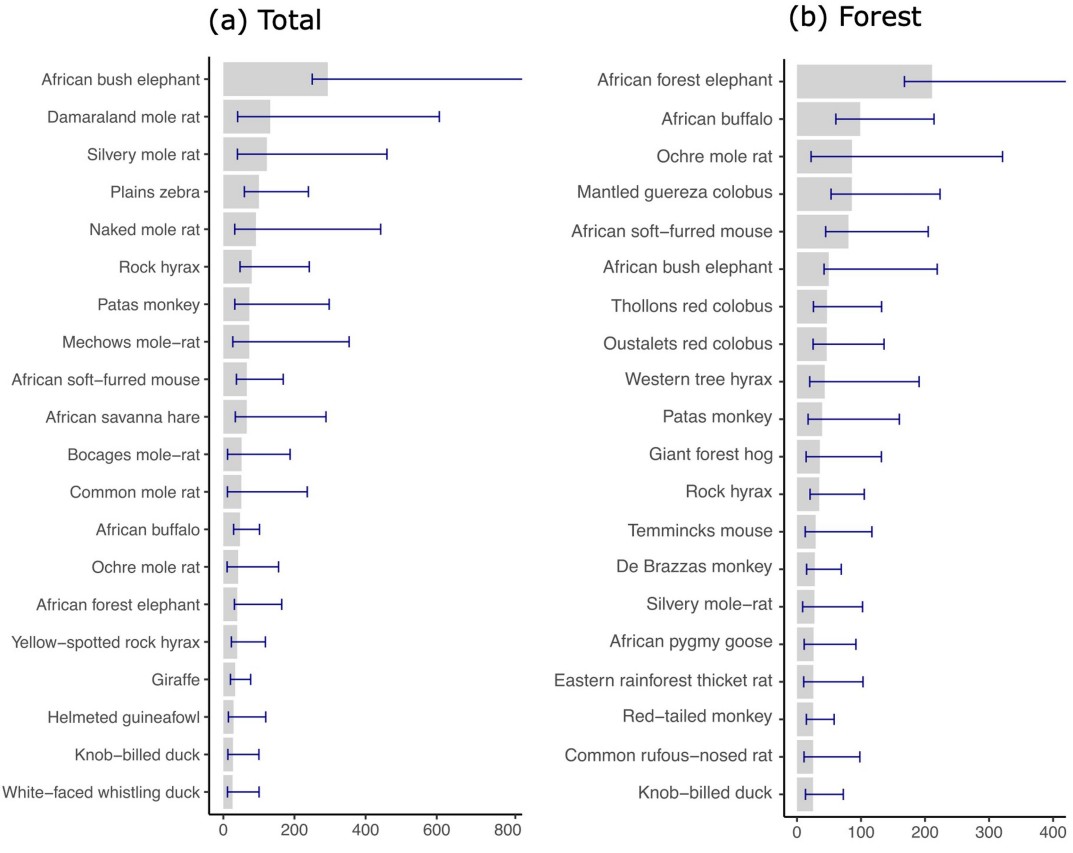

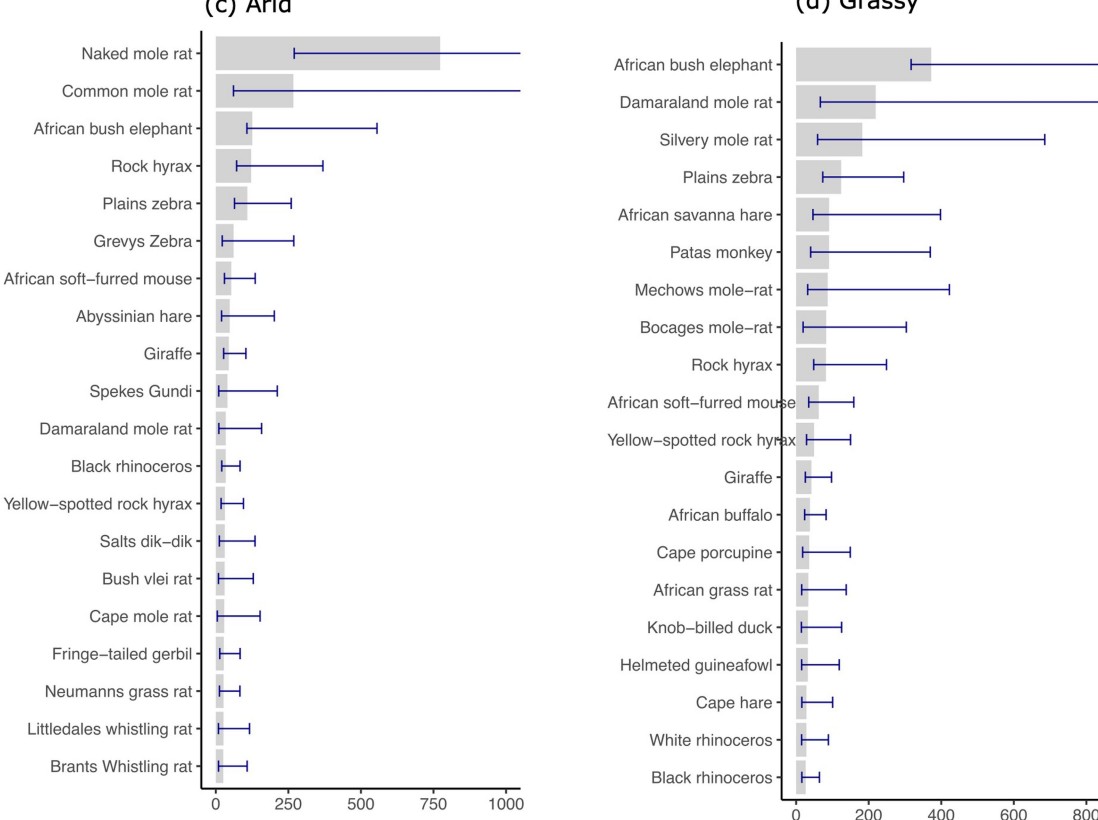

**Extended Data Fig. 3 | The historically dominant bird and mammal species in terms of food energy consumption in sub-Saharan Africa and across biomes.** Species-level energy consumption by the top 20 energy consumers in sub-Saharan Africa (a) and across major biomes (b–d). Error bars denote mean values and 95% confidence intervals derived from 10,000 monte-carlo simulations incorporating various sources of uncertainty.

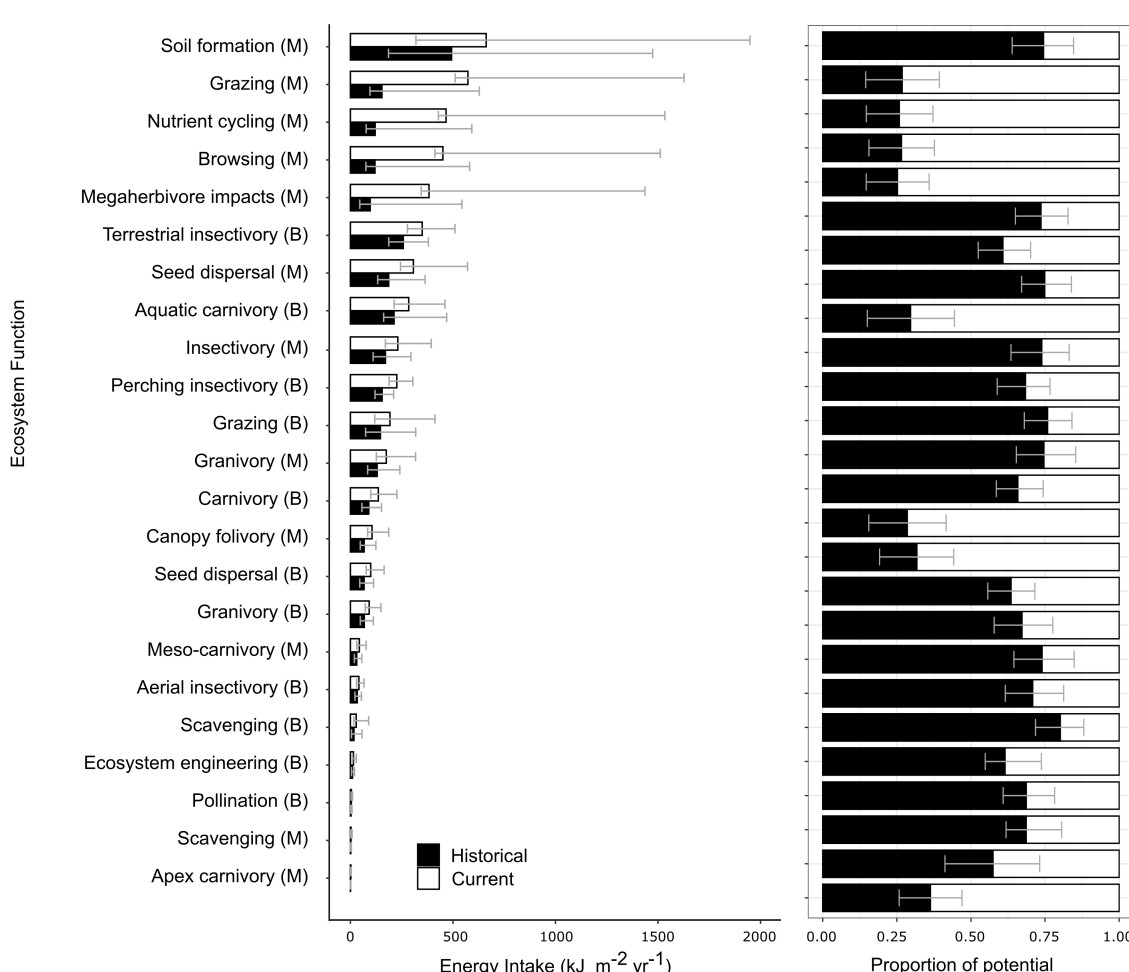

**Extended Data Fig. 4 | Energy flows through groups of species performing 23 fine-scale ecosystem functions, separated by birds and mammals.** Functions performed by birds are marked (B) and functions performed by mammals are marked (M). **a**, Total historical (white) and modern (black) mean energy flow through ten bird and mammal functional cohorts across sub-Saharan African. **b**, Average energy flow through modern sub-Saharan Africa (black) as a proportion of historical energy flow (white). Error bars denote 95% confidence intervals derived from 10,000 monte-carlo simulations incorporating various sources of uncertainty.

| Ecosystem Function | All | Protected Lands | Untransformed Lands | Croplands | Settlements |
|---|---|---|---|---|---|
| Aerial insectivory (B) | 24.74% | 9.35% | 14.86% | 16.97% | 80.40% |
| Grazing (B) | 18.06% | 9.12% | 22.17% | 66.15% | 41.61% |
| Aquatic carnivory (B) | 16.57% | 8.11% | 13.03% | 30.93% | 40.51% |
| Granivory (M) | 16.15% | 10.23% | 10.83% | 85.31% | 17.56% |
| Soil formation (M) | 15.94% | 6.60% | 42.64% | 75.70% | −2.91% |
| Granivory (B) | 15.18% | 6.57% | 5.66% | 161.28% | 75.49% |
| Insectivory (M) | 15.01% | 10.22% | 9.33% | 9.61% | −0.60% |
| Terrestrial insectivory (B) | 14.61% | 7.81% | 10.22% | 42.03% | 34.50% |
| Meso-carnivory (M) | 10.25% | 10.95% | 4.13% | 11.37% | −23.42% |
| Ecosystem engineering (B) | 6.87% | 8.11% | 0.57% | −38.57% | 40.51% |
| Pollination (B) | 6.85% | 7.53% | −2.93% | −36.29% | 56.39% |
| Perching insectivory (B) | 6.45% | 7.21% | −1.46% | −31.93% | 26.99% |
| Seed dispersal (B) | 4.64% | 4.52% | −5.33% | −54.78% | 36.80% |
| Carnivory (B) | 2.45% | 3.44% | 0.14% | 21.64% | 9.81% |
| Canopy folivory (M) | −1.10% | 7.93% | −13.17% | −86.97% | −36.02% |
| Scavenging (B) | −4.25% | 1.03% | −4.93% | 2.94% | −49.96% |
| Seed dispersal (M) | −5.38% | 3.97% | −15.17% | −68.87% | −32.07% |
| Scavenging (M) | −10.57% | −3.76% | −5.20% | 39.10% | −33.61% |
| Apex carnivory (M) | −43.43% | −19.70% | −45.90% | −47.07% | −80.11% |
| Grazing (M) | −58.25% | −34.70% | −56.57% | −71.72% | −86.94% |
| Browsing (M) | −58.62% | −31.47% | −58.73% | −82.11% | −86.00% |
| Nutrient cycling (M) | −59.76% | −34.08% | −59.85% | −81.56% | −87.21% |
| Megaherbivore impacts (M) | −60.75% | −33.38% | −61.36% | −79.48% | −86.55% |

**Extended Data Fig. 5 | Winning and losing ecosystem functions after human impacts.** Values represent the change in the proportion of energy flowing through each ecosystem function after accounting for human land use. Values were calculated proportionally, as the percentage change to the historical proportion of energy that flowed through each ecosystem function. Only one function, avian granivory in croplands, experienced an absolute increase in energy flow compared to historical values (Extended Data Fig. 4). But many functions experienced a relative increase in importance, even as absolute energy flows through those ecosystem functions declined. Bluer cells indicate "winning" functions, which experienced increasing relative importance, while redder cells indicate "losing" which experienced decreasing relative importance.

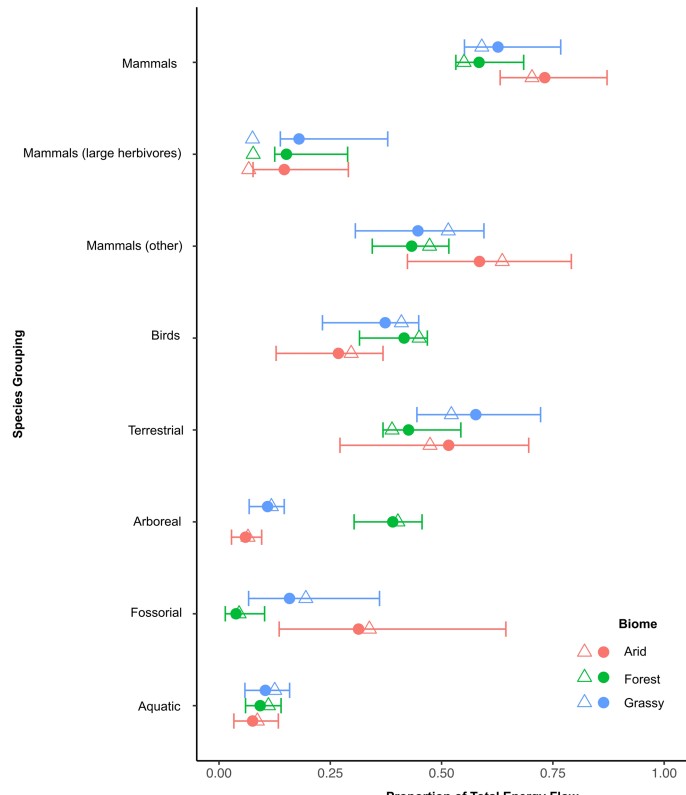

**Extended Data Fig. 6 | Proportional energy flow through species groups.**
Circles indicate the historical fraction of total bird and mammal energy flow
accounted for by a given of group of species. Triangles indicate the current
fraction of total bird and mammal energy flow accounted for by a given of
group of species. Species were grouped based on taxonomic class, lifestyle.
Large herbivores were designated based on Hempson et al.[44]. Error bars denote
mean values and 95% confidence intervals for historical values, derived from
10,000 Monte-Carlo simulation estimates incorporating uncertainty in body
mass, population density, the daily energy expenditure equation, assimilation
efficiency of different food types, composition of the diet of each species, and
the biodiversity intactness of each species within each land use.

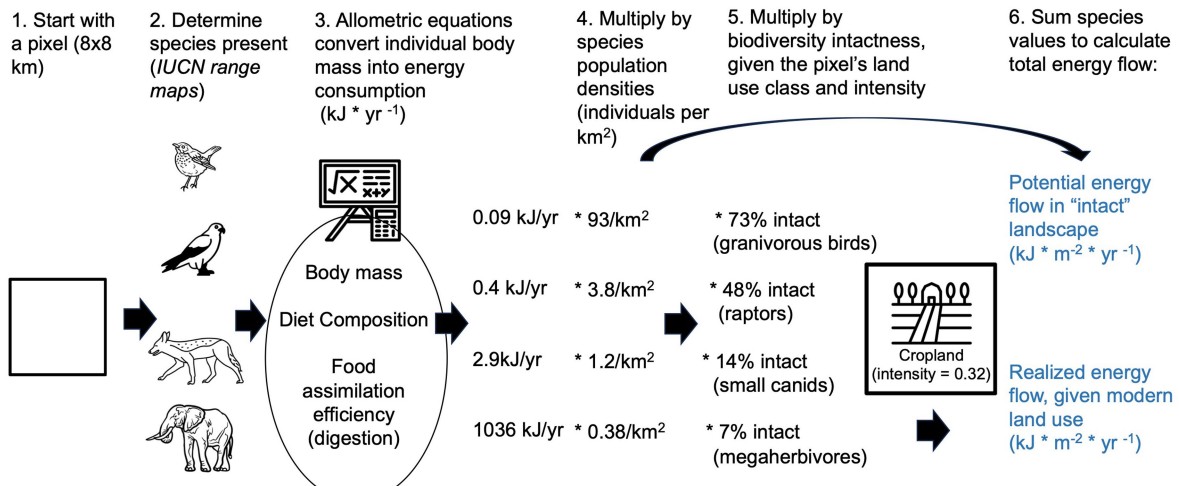

**Extended Data Fig. 7 | Diagram of methods.** The diagram demonstrates the steps that were used to calculate the current and historical energy flows of trophic and functional guilds of species within each cell, as explained in the methods portion of the main text.

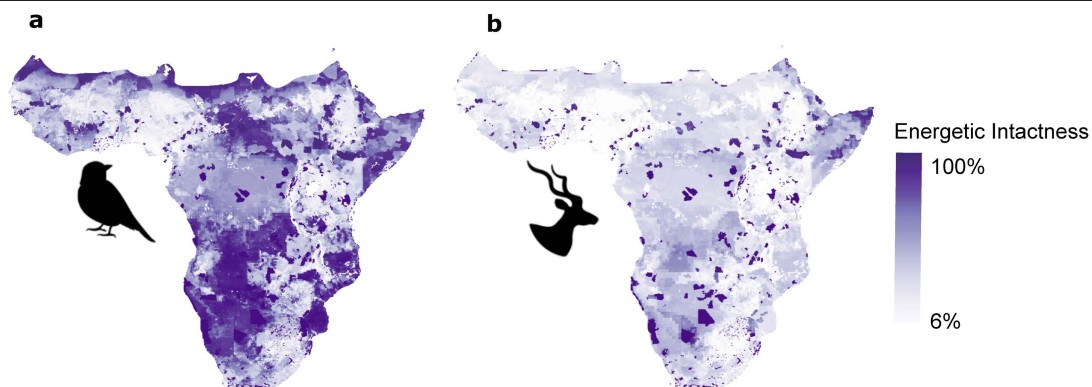

**Extended Data Fig. 8 | Intactness of energy flows through birds and mammals after human impacts.** Maps indicate the proportional intactness of (a) birds and (b) mammals. Fractional energetic intactness was calculated by dividing the current absolute energy flow through each 8 × 8 km grid cell by the historic absolute energy flow through each 8 × 8 km grid cell. Current absolute energy flows were calculated by multiplying the energy flows through each species present in a grid cell by the species' change in abundance, as estimated by the biodiversity intactness index for Africa.

**Extended Data Table 1 | Criteria for allocating bird and mammal species into ecosystem function groups**

| Broad scale ecosystem functions | Fine scale ecosystem function | Species number | Designation Criteria |
|---|---|---|---|
| Pollination | Pollination (B) | 41 | Trophic niche = nectarivore. |
| Seed dispersal (frugivory) | Seed dispersal (B) | 140 | Trophic niche = frugivore. |
| | Seed dispersal (M) | 191 | Diet > 25% fruit. |
| Granivory | Granivory (B) | 239 | Trophic niche = granivore. |
| | Granivory (M) | 131 | Diet > 25% seeds, and body mass < 500g. |
| Insectivory | Aerial insectivory (B) | 96 | Trophic niche = insectivore, and lifestyle = aerial. |
| | Terrestrial insectivory (B) | 361 | Trophic niche = insectivore, and lifestyle = terrestrial. |
| | Perching insectivory (B) | 660 | Trophic niche = insectivore, and lifestyle = insessorial. |
| | Insectivory (M) | 497 | Diet > 25% insects. |
| Soil disturbance | Soil disturbance (M) | 42 | Locomotion = fossorial, or aardvark. |
| Meso-carnivory | Carnivory (B) | 86 | Trophic niche = vertivore. |
| | Aquatic carnivory (B) | 112 | Trophic niche = aquatic predator. |
| | Meso-carnivory (M) | 147 | Diet > 50% vertebrates, and body mass < 15kg, and not fossorial. |
| Apex carnivory | Apex carnivory (M) | 5 | Diet > 50% vertebrates, and body mass > 15kg. |
| Scavenging | Scavenging (B) | 9 | Trophic niche = scavenger (i.e. vultures), or marabou stork. |
| | Scavenging (M) | 4 | Diet > 25% scavenging. Body mass > 500g. These include brown hyena, striped hyena, black backed jackal, and side striped jackal. |
| Grazing and browsing by large herbivores | Grazing (M) | 56 | Large terrestrial herbivores (Hemspon et al., 2015) with diets > 25% monocots, and *Gorilla* spp. |
| | Browsing (M) | 46 | Large terrestrial herbivores (Hemspon et al., 2015) with diets > 25% dicots, and gelada. |
| Nutrient dispersal by large herbivores | Nutrient dispersal (M) | 29 | Large terrestrial herbivores with average herd size ≥ 30. |
| Other | Grazing (B) | 32 | Trophic niche = aquatic herbivore or terrestrial herbivore. |
| | Ecosystem engineering (B) | 33 | Family = *Picidae* (woodpeckers). |
| | Canopy folivory | 66 | Diet >25% leaves and arboreal locomotion. |
| | Megaherbivore impacts | 5 | Mammals with body mass > 1000kg (i.e. invulnerable to predators). These include black and white rhinoceros, savanna and forest elephant, and hippopotamus. |

Species were initially allocated into fine scale ecosystem functions, and these were then aggregated into the broad scale ecosystem functions visualized in the main text figures.

**Extended Data Table 2 | Changes in energy flow through species groups based on body mass and land use**

| Land Use | Size Class | Energy - Historical (KJ * m-2 * yr-1) | Energy - Current (KJ * m-2 * yr-1) | Proportion of Energy-Historical | Proportion of Energy-Current |
|---|---|---|---|---|---|
| all | large | 685 | 186 | 16% | 7% |
| all | medium | 636 | 398 | 15% | 15% |
| all | small | 1818 | 1314 | 43% | 49% |
| all | smallest | 1061 | 776 | 25% | 29% |
| **all** | **all** | **4200** | **2675** | 100% | 100% |
| Croplands | large | 728 | 45 | 16% | 2% |
| Croplands | medium | 722 | 248 | 16% | 13% |
| Croplands | small | 1978 | 979 | 44% | 53% |
| Croplands | smallest | 1086 | 581 | 24% | 31% |
| **Croplands** | **all** | **4514** | **1853** | 100% | 100% |
| Protected Lands | large | 858 | 499 | 17% | 11% |
| Protected Lands | medium | 657 | 613 | 13% | 14% |
| Protected Lands | small | 2269 | 2151 | 46% | 49% |
| Protected Lands | smallest | 1143 | 1095 | 23% | 25% |
| **Protected Lands** | **all** | **4928** | **4358** | 100% | 100% |
| Settlements | large | 672 | 24 | 15% | 2% |
| Settlements | medium | 671 | 129 | 15% | 11% |
| Settlements | small | 1993 | 644 | 44% | 53% |
| Settlements | smallest | 1220 | 414 | 27% | 34% |
| **Settlements** | **all** | **4557** | **1211** | 100% | 100% |
| Untransformed Lands | large | 665 | 188 | 16% | 7% |
| Untransformed Lands | medium | 619 | 410 | 15% | 15% |
| Untransformed Lands | small | 1756 | 1315 | 43% | 49% |
| Untransformed Lands | smallest | 1050 | 789 | 26% | 29% |
| **Untransformed Lands** | **all** | **4090** | **2702** | 100% | 100% |

Species were allocated into body mass classes according to the following thresholds: smallest (<0.5 kg), small (<3 kg), medium (<65 kg), large (> 65 kg). The thresholds between the small, medium, and large groups follow Pringle et al.[13]. The threshold between the small and smallest groups was added to provide additional detail.

# Reporting Summary

## Statistics

For all statistical analyses, confirm that the following items are present in the figure legend, table legend, main text, or Methods section.

| n/a | Confirmed | |
|---|---|---|
| ☐ | ☒ | The exact sample size (*n*) for each experimental group/condition, given as a discrete number and unit of measurement |
| ☐ | ☒ | A statement on whether measurements were taken from distinct samples or whether the same sample was measured repeatedly |
| ☐ | ☒ | The statistical test(s) used AND whether they are one- or two-sided<br>*Only common tests should be described solely by name; describe more complex techniques in the Methods section.* |
| ☐ | ☒ | A description of all covariates tested |
| ☐ | ☒ | A description of any assumptions or corrections, such as tests of normality and adjustment for multiple comparisons |
| ☐ | ☒ | A full description of the statistical parameters including central tendency (e.g. means) or other basic estimates (e.g. regression coefficient) AND variation (e.g. standard deviation) or associated estimates of uncertainty (e.g. confidence intervals) |
| ☐ | ☒ | For null hypothesis testing, the test statistic (e.g. *F*, *t*, *r*) with confidence intervals, effect sizes, degrees of freedom and *P* value noted<br>*Give P values as exact values whenever suitable.* |
| ☒ | ☐ | For Bayesian analysis, information on the choice of priors and Markov chain Monte Carlo settings |
| ☐ | ☒ | For hierarchical and complex designs, identification of the appropriate level for tests and full reporting of outcomes |
| ☐ | ☒ | Estimates of effect sizes (e.g. Cohen's *d*, Pearson's *r*), indicating how they were calculated |

*Our web collection on statistics for biologists contains articles on many of the points above.*

## Software and code

Policy information about availability of computer code

| | |
|---|---|
| Data collection | No software was used in data collection. |
| Data analysis | Data was analyzed and visualized with a custom code using R (version 4.5.0 and earlier), and the following R packages: ggplot2, tidyverse, rgdal, readxl, sp, raster, terra, sf, lwgeom, magrittr, and parallel. |

For manuscripts utilizing custom algorithms or software that are central to the research but not yet described in published literature, software must be made available to editors and reviewers. We strongly encourage code deposition in a community repository (e.g. GitHub). See the Nature Portfolio guidelines for submitting code & software for further information.

## Data

Policy information about availability of data

All manuscripts must include a data availability statement. This statement should provide the following information, where applicable:
- Accession codes, unique identifiers, or web links for publicly available datasets
- A description of any restrictions on data availability
- For clinical datasets or third party data, please ensure that the statement adheres to our policy

The data on energy flows through each species, trophic guild, and functional group are available in Supplementary Data 1 and 2. Input data on species population densities are available through the TetraDENSITY dataset (https://ecaslab.com/tetradensity-database/). Input data from the biodiversity intactness index are available through the BII4Africa project (https://bii4africa.org/). Input data on species ranges are available in the associated data with Beyer and Manica (2020,

https://doi.org/10.1038/s41467-020-19455-9) and from the IUCN database (https://www.iucnredlist.org/resources/spatial-data-download). Input data on species traits (i.e. diet, body mass, lifestyle) are available through the EltonTraits database (mammals) (https://opentraits.org/datasets/elton-traits.html) and through the Avonet database (birds) (https://opentraits.org/datasets/avonet.html). Input data on ecoregions is available through the RESOLVE ecoregion dataset (https://developers.google.com/earth-engine/datasets/catalog/RESOLVE_ECOREGIONS_2017).

# Research involving human participants, their data, or biological material

Policy information about studies with human participants or human data. See also policy information about sex, gender (identity/presentation), and sexual orientation and race, ethnicity and racism.

| | |
|---|---|
| Reporting on sex and gender | n/a |
| Reporting on race, ethnicity, or other socially relevant groupings | n/a |
| Population characteristics | n/a |
| Recruitment | n/a |
| Ethics oversight | n/a |

Note that full information on the approval of the study protocol must also be provided in the manuscript.

# Field-specific reporting

Please select the one below that is the best fit for your research. If you are not sure, read the appropriate sections before making your selection.

☐ Life sciences  ☐ Behavioural & social sciences  ☒ Ecological, evolutionary & environmental sciences

For a reference copy of the document with all sections, see nature.com/documents/nr-reporting-summary-flat.pdf

# Ecological, evolutionary & environmental sciences study design

All studies must disclose on these points even when the disclosure is negative.

| | |
|---|---|
| Study description | The study used statistical analysis to analyze existing datasets using a custom code in R |
| Research sample | n/a |
| Sampling strategy | n/a |
| Data collection | n/a |
| Timing and spatial scale | n/a |
| Data exclusions | n/a |
| Reproducibility | All data and code is documented and publicly available so that the study is reproducible. |
| Randomization | n/a |
| Blinding | n/a |

Did the study involve field work?  ☐ Yes  ☒ No

# Reporting for specific materials, systems and methods

We require information from authors about some types of materials, experimental systems and methods used in many studies. Here, indicate whether each material, system or method listed is relevant to your study. If you are not sure if a list item applies to your research, read the appropriate section before selecting a response.

## Materials & experimental systems

| n/a | Involved in the study |
|-----|----------------------|
| ☒ ☐ | Antibodies |
| ☒ ☐ | Eukaryotic cell lines |
| ☒ ☐ | Palaeontology and archaeology |
| ☒ ☐ | Animals and other organisms |
| ☒ ☐ | Clinical data |
| ☒ ☐ | Dual use research of concern |
| ☒ ☐ | Plants |

## Methods

| n/a | Involved in the study |
|-----|----------------------|
| ☒ ☐ | ChIP-seq |
| ☒ ☐ | Flow cytometry |
| ☒ ☐ | MRI-based neuroimaging |

## Plants

Seed stocks
*Report on the source of all seed stocks or other plant material used. If applicable, state the seed stock centre and catalogue number. If plant specimens were collected from the field, describe the collection location, date and sampling procedures.*

Novel plant genotypes
*Describe the methods by which all novel plant genotypes were produced. This includes those generated by transgenic approaches, gene editing, chemical/radiation-based mutagenesis and hybridization. For transgenic lines, describe the transformation method, the number of independent lines analyzed and the generation upon which experiments were performed. For gene-edited lines, describe the editor used, the endogenous sequence targeted for editing, the targeting guide RNA sequence (if applicable) and how the editor was applied.*

Authentication
*Describe any authentication procedures for each seed stock used or novel genotype generated. Describe any experiments used to assess the effect of a mutation and, where applicable, how potential secondary effects (e.g. second site T-DNA insertions, mosaicism, off-target gene editing) were examined.*

