## [Peer Review file · Nature]

Energy flows reveal declining ecosystem functions by animals across Africa

Corresponding Author: Mr Ty Loft

Version 0:

Reviewer comments:

Referee #1

(Remarks to the Author)

This paper offers a continent wide analysis of how biodiversity loss has changed energy flow (and sometimes other ecosystem services) across Africa. It takes advantage of recent expert derived datasets estimating biodiversity intactness across the continent and modelled population density estimates to undertake this project and notes that energy flows have been changed in various ways by loss of different functional groups. Declines in elephants have had a disproportionate impact on energy flows, while in forests birds have a larger impact than elsewhere. The authors suggest that this combination of Intactness and energetics offers a possible route to incorporation of concrete biodiversity loss limits for planetary boundaries type analyses. All this sounds exciting and potentially important.

Before thinking about interpretation and conclusions, it is important to evaluate the quality of the data. The paper relies heavily on two datasets - one estimating 'natural' abundance of individual species, one estimating at a very fine resolution the decrease in abundance from those natural levels for each species. (Basic metabolic computations are then used to turn biomass of each species into energetic costs.). One of these datasets is already published, but has hardly been widely accepted by the ecological community (indeed, estimates have already been corrected by around 35% as a result of an error the authors noted), the other is not yet available. The population densities come from two global analyses (birds and mammals), and it is reasonable to assume that the data for Africa are among the most uncertain - mostly estimated as a function of species traits and habitats based on data from outside the continent. The authors (sensibly) opt to incorporate uncertainty in body mass and 'natural' population density (but not apparently intactness) using an MCMC approach, and the credible intervals they provide cover an order of magnitude difference (Fig 1), though I rather suspect this is an underestimate given the bias in global datasets towards temperate systems. Similarly, the apparently lack of incorporation of variation in intactness measures (when I understand they are available) seems unwise and would likely even further impact these changes. Given the large uncertainty already involved, and the even larger uncertainty in the expert opinion intactness data the estimates must be considered little more than guesses. I question whether these data are suitable for the analysis, or simply build a house of cards. I also wonder why the authors attempt it in Africa? If their aim really is to describe and test a new energetics based method of computing human impact, why not use a much better documented continent than Africa where the uncertainties (which must multiply) are so large I can't really believe much.

Not only are the African data highly uncertain, but add to this as the authors note the lack of invertebrate data (we know that something like 50% of energy in some grassland systems is consumed by invertebrates, so this is not a trivial issue) and, indeed, the lack of consideration of fire as an energetic consumer in much of Africa, and I am left feeling this work needs significantly more context before it is useful (assuming the underlying data issues are accepted). I don't think we can really interpret the loss of 30% energy cycling through a decline in a subset of vertebrates, when we have no idea what is happening to either invertebrate energy cycling or changes in fire? Is this decline really important?

Some more specific issues that might be useful to expand on:

(1) There's a degree of swapping between proportional change (most of the time) and estimates of absolute energy consumption (in the text). This could be clarified.

(2) Similarly, there is a confusion over when energetics per se are being cited, and when this is being provided as an ecosystem function. For example, is granivory an ecosystem function (as implied in Fig 1)? Or a trait? Or an energy flow? The

writing and description of what is simply energetics and what (like seed dispersal) is really an ecosystem function needs to be much clearer. Why not stick with just energetics and quantification of flows? I think, for example, that seed dispersal impacts of, e.g. empty forests has been estimated much more comprehensively and accurately elsewhere and the authors may be well advised to decide what they really care about.

(3) vibrancy is cited, but needs explanation here

(4) I'm also puzzled by Fig 1 showing apex carnivory is more intact in unprotected than protected areas. (Also grazing & browsing, but maybe that is domestic stock?) Can this be explained?

(5) There seems to be a fundamental problem with spatial resolution of historic analysis - the polygons used are so much smoother than current ones, because they come from a very different methodology, that there are necessarily going to be large declines estimated for pretty much every species. I think some sort of sensitivity to smoothing needs to be applied. For example, each successive IUCN / BirdLife release of maps almost inevitably reduces the estimated areas of occupancy, because some new technique has been developed to exclude unsuitable habitat from the polygon (e.g. the major changes when altitude limits were stamped on all the maps relatively recently). I would like to see how much of the declines estimated are a result of refining range maps, rather than actual range change per se.

(6) Labels for fig 4 c & d are in text and a and b

(7) The authors themselves say "the energetic approach presented here, ..., is highly inaccurate at predicting values for individual species or landscapes" They then continue "Rather, these estimates are useful for large-scale taxonomic and geographic comparisons when values are aggregated across many species and landscapes, and can elucidate how humans are impacting ecosystem function at scales relevant for policy." but I suggest that this is somewhat like suggesting lots of bad data will lead to sensible policy-relevant decision making. I'm not totally convinced and the section on restoration restoration section is pretty unconvincing - what recommendations here are not what people are currently aiming at? Equally, if this is to be useful for discussions around planetary boundaries, don't we need to see how the authors would recommend how to pick one in the face of all the uncertainty in the metrics? It will presumably need to be an absolute metric, not a relative one - but that isn't given much discussion.

Referee #2

(Remarks to the Author)

This manuscript proposes a new metric for quantifying the roles of animals in assessing broadscale ecosystem function and applies this in a case study of sub-Saharan African birds and mammals. The authors postulate that a comparison of energy flows through historical vs. contemporary animal communities provides a measure of change in ecosystem function and explore changes across different trophic guilds, animal functional groups, biomes and in areas of different land use. Their novel approach makes use of two new datasets and draws on a wide variety of other datasets. They describe implications for restoration, global policy and priority-setting, and further research.

I find this approach novel and exciting, and believe that this manuscript presents a major step forward in the field of biodiversity conservation. To date, no broad-scale measure of landscape functionality has previously been available in the global suite, which has resulted in this aspect of ecosystem condition being poorly represented in policy-relevant assessments such as IPCC and IPBES. The paper has been very well-written, the arguments and results clearly presented and the referencing appropriate. The methods and results are complex and I suggest ways to improve their clarity and ease of interpretation. There are also some important caveats that I believe have been neglected, and the extent of some of the bold claims reduced - I make further suggestions. Overall, I think the results are sufficiently robust, at least to an equivalent degree to other established studies and indices of this type. This is excellent work, of global relevance and I recommend its publication if the issues below can be addressed.

Recommended actions

1. The workflows for deriving metrics of energy flow are many-stepped and require detailed scrutiny of long descriptions to decipher. I had to draw a diagram to see how it all fits together. Since a major point of the paper is to propose a new method, I suggest making this more accessible to readers, for example by adding a schematic figure. This could detail steps from acquiring/compiling species range maps through abundance estimation, adding BII, working from body mass through to consumption and then on to functional groups and guilds.... Etc. The text could then refer to them by name or number, making it easier to follow, and providing the opportunity link with explanations, references, dataset examples/suggestions, caveats and suggestions.

Caveats. There are so many steps and nested datasets and layers on which the final measures and claims are built that the idea of a house of cards comes to mind. I've needed to work step-by-step through your steps (and my figure) to assure myself that I've considered the limitations at each. Considering those who may follow your approach based on this paper, what would they need to know, do or not do, bear in mind in interpretation, etc. to use it responsibly? You have bits and pieces of caveats scattered through the ms (e.g. Lines 274-277, which are particularly important but not in the caveats section). I think your caveats section needs to feature more prominently, including both as a short section in the main text and a more thorough and systematic section under methods. With future external users in mind, this could talk systematically to each application step.

Beyond the dataset limitations referred to here and in other underlying datasets from other papers, here are a few additional ones that I believe should be included:

a. IUCN Red List range data. Lns 601-608. It isn't clear whether the authors recognise that the IUCN polygons used vary widely between species and geographic sub-regions in terms of their mapping resolution and accuracy. Ranges of well-known birds and mammals in highly-observed areas, for example, have been mapped accounting for fine-scale habitat heterogeneity (in contradiction to the text), already account for area-of-habitat, and have polygons that identify now-extinct range sections (i.e. good approximation of historical range at circa. 1700). The majority of other sub-Saharan animals, however, are broader range estimations, closer to extents-of-occurrence, and there is no knowledge or inclusion of now-extinct range sections. IUCN does not have /provide meta-data enabling users such as the authors to know which is which, or how these vary across regions, even for a single species.

As a result, historical range extent inaccuracies (considered independently of other variables) could lead to over-estimating decline in poorly known geographic areas, and underestimating decline in poorly-known taxa. This applies and has implications both for this and the BII Africa study and should be discussed.

b. Ln638: cells in PAs are assigned an intensity class of 1, but what about the species groups for which PAs cannot be assumed to represent a healthy historical state (e.g. vultures; human-hunted/poached animals)

c. The BII's limitations, given its prominent role in this study

Minor corrections

Fig 1. Labelling of panels inconsistent format (i.e. needs to be (a) not a.)

Fig 2: a fabulous figure! (OK, so this isn't a correction)

Fig 3: Lns 558 560. Missing a full-stop and the first (d) should be a (c). I suggest adding the bird and mammal icons to the graphs themselves for ease of interpretation.

Line 682: suggest changing to "...Unlike for birds, there is so single authoritative source..."

Line 326: restoring forest birds and primates is seldom feasible – I suggest instead recommending prioritising conservation of or avoidance of loss of..."

Referee #3

(Remarks to the Author)

The Loft et al. study addresses the consequences of changes in bird and mammal populations for energy flows through ecosystems in Sub-Sahara Africa. The authors elegantly integrate new datasets on animal population densities, biodiversity intactness (BII) and energy consumption to provide a picture of where and by how much energy flows have declined during the past approximately three centuries. The uncertainty analysis performed – in which uncertainties of the various databases are combined – seems sophisticated. The authors find an overall decline in energy flows of one third. Main (and most interesting) results are that (1) mega-fauna reduction and extinction have had tremendous impacts on energy flows through Sub-Saharan ecosystems and (2) small mammals are on the rise both in density and in their contribution to ecosystem energetics. Overall, the study convincingly shows that an energetics approach can be applied at continental scale to quantify anthropogenic changes in ecological processes and is a valuable addition to other metrics (e.g., carbon stock, NPP, GPP) because it explicitly considers the role of animals.

1. Ecosystems & land use classes. The study targets energy flows through 'ecosystems', suggesting that is restricted to (semi-)natural ecosystems. Yet, the study also includes cropland and settlement. In my view, this is problematic for three reasons: (1) The inclusion of croplands and settlements in this study seems to be at odds with the use of the terms 'ecosystem' and 'biome'. In Fig 2 and elsewhere biome names are used, e.g. stating that "energy flows are estimated to be [...] intact in forests..." (L 164-5), suggestion that that the analysis is limited to forests, but I guess in the practice of this study the 'forest biome' also includes croplands and settlements. (2) I do not think including croplands and settlements is warranted because the purpose of the study is to assess how shifts in animal abundance affects the energy flows in 'ecosystems'. If the land cover changes from a (semi-)natural vegetation to crops (or to settlements), one does not assess shifts in energy flows through changes in animal abundance alone, but also due to the shift in the plants (crops) growing there. This, in my view, goes beyond the modification of ecosystems. (3) Mammal and bird species that survive in highly altered systems such as settlements and croplands very likely can do so because of (partial) dietary shifts. As far as I can tell, such dietary shifts have not been included in the study. This likely induces bias in the estimates of energy flow in settlements and croplands), which is not accounted for. In sum, I would argue to leave out croplands and settlements from the analyses (also from maps) to provide a more clear focus. One could consider to include agroforestry-type systems in the analyses, if possible.

2. Changing energy flows and ecosystem services. I seriously doubt whether inferences about shifts in ecosystem functions can be made based on changes in energy flows. I read the justification in the Methods section (L707-710) and understand why the authors have opted not to estimate 'functional efficiency'. Yet, without any evidence that a reduction in energy flow through animals involved in a certain ecosystem function (for example seed dispersal) affect that service (number or mass of seeds dispersed), the results of shifts in energy flows through seed dispersing animals are hard to interpret. In spite of this limitation, the authors do present and interpret the shifts in energy flows as shifts in pollination, dispersal, etc. For example, L205-206 read "Seed dispersal is just 58% intact in untransformed lands and 14% intact in croplands." and the summary (L 32-33) mentions "with functions performed by megafauna in particular collapsing outside protected areas". So, implicitly (or

perhaps explicitly?) one-to-one relations between shifts in energy flow and those in ecosystem services are assumed without support.

3. Biodiversity and ecosystem energetics. Part of the analysis and results (Fig 4) is devoted to relating biodiversity to energetics. Clearly, this is an interesting component of the study, and it is logical to verify to what extent changes in species abundance (BII) predict comparatively large changes in energy flows. Yet, the strong relationships and high R² values for Fig 3a-b should not come as a surprise because the shifts in ecosystem energy flows is (i.e., the y-axis) is calculated using values of biodiversity intactness (the reverse of BII, i.e., the x-axis). They are now interpreted (L262-264) as "biodiversity intactness is a strong predictor of total energetic intactness", but given the dependency this a methodological association rather than one that provides insights into the relation between diversity and energetics. I suggest to reframe this part of the text, by asking the question to what extent biodiversity intactness is a good predictor of shifts in energy flows, and for what groups it is not. That way, the added value of the energetics approach compared to biodiversity-related metrics can be highlighted and discussed.

4. Uncertainties. While the uncertainty analysis overall appears to be solid, it does not address two sources of uncertainty. (1) L800-801 states that "we were unable to model geographic variability in population densities within species". This may be problematic for the most important species in terms of energy flows, e.g. elephants. For several of those taxa, species-specific density estimates will likely be available. I'd advise the authors to conduct a robustness analysis for some of the key taxa, in which they replace the 'flat geographic population densities' used now by spatially explicit species-specific population densities to calculate shifts in energy flow. If the overall shift in energetics are robust to this change in densities used, then this provides support for the current assumption of flat densities. (2) Historical population densities: 'historical' estimates are also influenced by anthropogenic effects, as mentioned in Santini et al (ref 24 in ms): "Our estimates are likely to be influenced by anthropogenic pressure". Which means that for many species historical estimates are likely underestimates. I have the impression that this possible bias has not been accounted for in the analyses. What would the effect be?

Other comments

- Summary paragraph: would be good to mention in a few words what data sources are used.

- Ecological vibrancy: to me it remains unclear how this is different from the quantification of energy flows. (The definition in L117-118 remains rather vague.) I would advise removing the term (or defining it very clearly).

- L 193-195: I read (Worldbank Open Data: <https://data.worldbank.org/indicator/AG.LND.AGRI.ZS?locations=ZG>) that about half of Sub-Saharan Africa has an agricultural use. How does that relate to the 80% of untransformed & unprotected land?

- L 253: I find the conclusion that "fossorial rodents may be similarly impactful belowground" (compared to elephants aboveground) to be tricky and poorly supported by evidence, given the abovementioned (L244-249) extremely large uncertainty range. I'd argue that such a claim would only be supported if a realistic low estimate for fossorial rodents still would put them at a similar level (order of magnitude scale) as elephants.

- L 613: good to add for how many species these 10 thousand estimates were done, and that these were then extrapolated to other species

- L 375-378. I value the discussion on proxies for biodiversity to be used in large-scale assessments and analyses of planetary boundaries. I wonder to what extent (geographic) patterns of energetics shifts compare to those of HANPP. Would it be possible to substantiate (and expand) this discussion by adding a Suppl Fig with such a comparison? That would illustrate how and where an energetics approach would be more meaningful than a NPP approach.

- Fig 1: "energy flows through animal-mediated ecosystem functions" this is quite complex to grasp. Any way to simplify or better explain?

- Fig 2: Does this include all land uses (so also agriculture and croplands)? And if so, do the differences tell something about the biological/ecological differences between biomes, or rather about differences between 'biome' in conversion rates or types of land use to which original vegetation is converted? In the latter case, it may not be very meaningful to compare across biomes.

- Fig 2. Given the large uncertainty for fossorial animals and their energetic dominance in arid and grassy systems, I'd argue that it would be important show uncertainties in the estimates. Another option for fossorial animals is to show a conservative estimate, based on species-specific densities of mole rats obtained from other sources? Reporting such very high values requires extra checking.

- Fig 4: Wouldn't it be informative to also show % change maps? I'm quite curious what they would look like. So, a row with three maps per animal group: historical, current and % change.

- Fig 4: What is the reason for the coarse patterns? It seems different layers are overlaid, while I would expect to see the cumulative value of all energy flows..

- Fig 4: I miss values and units for the energy flows, which also makes comparison between historical and current difficult (or actually impossible).

Referee #4

(Remarks to the Author)

The authors draw from a large amount of data coupled to allometric consumption rates, to argue that the ecosystem energetics that run the sub-Saharan African ecosystems have been fundamentally altered compared to historical times (1700 CE). The authors use these energetics estimates of consumptive flow as an argument for declining ecosystem function and a link to disrupted structure-function relationships. I applaud the authors for this work and largely like the direction (the role of structure-function and energetics to sustainability management is critical) but have some concerns of

whether it meets the bar for a Nature publication. I do think that this type of paper would push people to further develop energetics within an ecosystem management perspective. My comments are below, some more curiosity questions.

Major comments

1). My first issue and one that is more serious potentially is that while I am sympathetic to bringing in energy flow and structure-function relationships I think that this paper needs to prove that we get somewhere novel with an energetics application. That is, that when we apply the energetics we learn something that we do not get from biodiversity and changes in density alone. The authors try to do this but to me it needs to be executed more clearly and I feel as stands not confident that they have done this. For example, when I see all the results it seems to me that the results are really just a simple "scaling up" of the "functional diversity" and density results. As a specific example, arboreal species reduction in energy flow in forests are merely the scalar response of a reduction in arboreal species relative densities and diversity. It is fine to say that this impacts energy flow but this feels quite straightforward and implicitly known. As an example, would the diversity*density plots of Figure 2 be identical qualitatively to the flow plots in figure 2? The lessons for ecosystem restoration on lines 320-338 seem to me could all be done from changes in functional diversity and may not require energetics at all.

2). My second comment is on structure and function, I was modestly confused with the use of these key terms. Food web ecologists might argue that structure of an ecosystem comes from i) the species interactions (topology), and ii) the strength of these interactions (defined differently but often with energetic flow more recently). The function comes from this as primary productivity, secondary productivity, nutrient cycling, oxygen levels and so on. I think the authors here are effectively suggesting they are measuring secondary production defined as consumed carbon/biomass after conversion of prey into new biomass. If they are, then this should be made clear. I think a box with some key definitions might go a long way in this including: structure, function, biodiversity intactness, energetic intactness, energetic vibrancy, resilience (used a few times and not clear exactly).

3). The body size shifts in the African ecosystems is very interesting and consistent with other work. I wonder if there are relationships with body size, density, diversity and energy flow? It might be interesting to show us how body size across species has been impacted in these systems and their relationship to secondary productivity or energy flow as the authors point out briefly somewhere, from allometric patterns in consumption rates, we would expect here, that energetic impacts may have a significant changing impact (small species). I was wondering if small species are declining too, just their relative amounts are increasing. If so, this should be made clear.

4). I wonder if the authors could pull out more clearly winners and losers with this change. Often, with ecosystem change there are some species that take over. In this paper, it appears to be the argument by the authors that there is only largely an overall decline although there are modest suggestions of increases of some species. I was left wondering about this. Making this more clear would be useful. The authors discuss an example with logging accelerating vertebrate consumption and suggest this as a possibility. It would be nice to know if they found such anomalies. From an ecosystem stability perspective strong singular energy pathways would likely contribute to instability and so while high diversified energy flow might be increased function, high singular energy flows might increase secondary production but at the peril of lost ecosystem stability. This is an issue ignored by the authors, and is fine, but shows how increased fluxes may not be always a positive to a restored ecosystem.

Anyway, I liked the direction of this paper and feel this has potential. I was left wondering if they really had showed us that energetics pushed the bar of sustainability and impact a long way. If so, it needs to be brought out more strongly and coherently otherwise this paper fits in a more specific environmental journal.

(Remarks on code availability)

My graduate student ran the code and it all appeared to work but there was a longer run that did not have the data to do that run (nor we the patience to run it for 5 days as they claimed they did). All in all, I believe their coding side.

Version 1:

Reviewer comments:

Referee #1

(Remarks to the Author)

I thank the authors for their clear responses. Many of the arguments made in their response are sensible and mostly convincing. I would like to see some of the arguments more fully included in the main text - much of this is done, which I'm pleased to see, but some still needs to be fully addressed. For example, the authors suggest that even if the estimates of historic abundance are well off, the ball park results will be correct. However, I still look at figure ED3 and see that almost any rank order arrangement of species could be supported by the data - with the possible exception of elephants remaining high. I think the confidence intervals (which must be independent by species) mean that in any one MCMC draw almost any rank order is supported - this stochastic ordering of most species is presumably why such regionally restricted species as Salt's Dikdik appear in the figure but, say, Hippo do not. The only thing that I can rule out confidently is that elephant will remain in the top 10 of all such realisations. To convince all readers this is not important, I think you need to convince in the main text that averaging (adding) within groups makes this less rather than more certain, allowing greater confidence at the functional group level than at the individual species level.

I also thank the authors for being more candid about the way they are estimating the intactness of functional groups - this was a concern I raised that another reviewer managed to put their finger on more explicitly. The answer - involving making the assumption that every species contributes equally and there is a 1 to 1 relationship between energy and contribution - makes me question again the value of including this. I think it is so unlikely this assumption holds (indeed, we have lots of evidence that it isn't true at least over the short term in any given ecosystem), I would still suggest cutting and retaining the core story about energy. I think the ecosystem function add-on is something of a distraction from the core message, built on even shakier foundations.

Referee #2

(Remarks to the Author)

Overall, my comments have been well addressed. The authors have reworked their estimates of species' distributions by adding a step of calculating Area-of-Habitat. Their updated approach draws on the work of Beyer and Manica (2020), with Beyer now included as a co-author. A comparison of use of "raw" vs. refined (area of habitat) is convincing in demonstrating that their findings are relatively robust in this regard.

Some specific feedback:

1. Summary figure of methodological steps: this is well done and very helpful
2. Caveats: New section is fine. First line (411) – the framing suggests that caveats are disadvantageous, rather than fundamental aspects of modelling. I suggest rewording it.
3. Ln 702: "For a few small-range species, 1/2° scale habitat-adjusted range maps eliminated all available range; for these species," – Surely the reason for the elimination in the larger grids was due to exclusion from the 1/12 degree grid cells?
4. There is a more recent reference that should inform this: <https://pubmed.ncbi.nlm.nih.gov/31324345/>
5. Fig S1: typo in header a.

Referee #3

(Remarks to the Author)

I have assessed the responses of the authors to the comments of all reviewers, and read the manuscript and additional new materials.

First, I'd like to reiterate what I stated in my review on the original submission (I was reviewer #3): "Overall, the study convincingly shows that an energetics approach can be applied at continental scale to quantify anthropogenic changes in ecological processes and is a valuable addition to other metrics (e.g., carbon stock, NPP, GPP) because it explicitly considers the role of animals." This is still my overall assessment. To this I'd like to add that I find this study to be a valuable contribution to Earth System studies. The revised paper is more balanced than the initial submission: caveats and limitations are better explained and illustrated, more analyses of robustness and sensitivities are now included, and I value the additional methodological discussion in the SI.

Second, my comments are satisfactorily addressed. I particularly like and value the additional analyses done to address comments #3 and 4. For comment #1 (on the contribution of habitat loss/conversion to shifts in energy flows), I would suggest the authors to include their estimate that "conversion ... is responsible for 25% of the total decline in energy flows" (quote from the rebuttal) in the main text. This is an important addition that helps interpreting the findings.

Third, I feel that concerns of other reviewers have also been adequately addressed in the rebuttal and revision. In particular, I find the responses to the concerns of reviewer #1 about the poor quality of the African datasets to be convincing. Plus, I'd like to add to this discussion that the fact that changes in African biomes are so poorly studied is -- in my view -- actually a strong justification for conducting this type of studies there. Of course, while taking uncertainties into account. I truly hope the paper --when published-- will create opportunities for energetics (and similar) studies at African research institutes and organizations. This type of studies have the potential to stimulate new studies, and help improving the data quality situation in understudied regions.

Finally, some minor (textual) comments:

- Fig 2 caption: the explanation what the asterisk for fossorial mammals is missing
- Fig S2: I'd be in favour of making this Figure more visible by including it as an Extended Data Fig
- L 341-342: sentence is a fragment, incomplete
- L 376: citation missing ("CITE")
- L 402 (around): could you refer to the S Fig with the poor association of HANPP vs energy?

Referee #4

(Remarks to the Author)

I have read the authors comments to all reviewers. The authors have very thoroughly responded to all queries and I feel the paper has greatly improved from this process. I, like some of the reviewers, remain a little wary of the data and the layers of assumptions but believe the overall goal of injecting an "energetics" lens on conservation is an important step forward and hopefully a paper like this will charge conservation with a more holistic energetcis perspective.

Response to Reviewers

*We thank the referees for their thoughtful and considered reviews. We have substantially edited the paper, conducting two sensitivity analyses; using new, habitat-adjusted range maps; adding 5 new extended data figures and tables (and even more in the supplementary discussion), some of which reflect new analyses; and writing a new, extended conclusion that justifies the approach's importance and novelty. We have also edited the text throughout to respond to the referee's comments. Below we go through each of the comments in turn and give our responses in *blue italics*.*

Referee #1

This paper offers a continent wide analysis of how biodiversity loss has changed energy flow (and sometimes other ecosystem services) across Africa. It takes advantage of recent expert derived datasets estimating biodiversity intactness across the continent and modelled population density estimates to undertake this project and notes that energy flows have been changed in various ways by loss of different functional groups. Declines in elephants have had a disproportionate impact on energy flows, while in forests birds have a larger impact than elsewhere. The authors suggest that this combination of Intactness and energetics offers a possible route to incorporation of concrete biodiversity loss limits for planetary boundaries type analyses. All this sounds exciting and potentially important.

We thank the referee for their positive appraisal and for recognizing the potential importance of our approach.

Before thinking about interpretation and conclusions, it is important to evaluate the quality of the data. The paper relies heavily on two datasets - one estimating 'natural' abundance of individual species, one estimating at a very fine resolution the decrease in abundance from those natural levels for each species. (Basic metabolic computations are then used to turn biomass of each species into energetic costs.). One of these datasets is already published, but has hardly been widely accepted by the ecological community (indeed, estimates have already been corrected by around 35% as a result of an error the authors noted), the other is not yet available. The population densities come from two global analyses (birds and mammals), and it is reasonable to assume that the data for Africa are among the most uncertain - mostly estimated as a function of species traits and habitats based on data from outside the continent.

The referee makes a good cautionary point and we agree that it is important to evaluate the quality of the underlying datasets, especially as these datasets are relatively new. However, we contest the argument that the dataset of natural abundances is not yet widely accepted in the ecological community. We also contest the argument that the underlying African data are among the most uncertain.

Although the population density estimates are recent (published in 2022 and 2023), they are derived from the more established and widely cited TetraDENSITY database, first

*published in 2018. The new estimates should be more accurate because they are based on a greatly expanded sample of population data. The model for mammals was based on >18K raw estimates for >800 species, and that for birds on >18K raw estimates for >1800 species (Lines 711-712). Both models were validated with robust approaches (i.e. Roberts et al. 2017, *Ecography*) and estimated a predictive error that we have accounted for in our analysis, assessing the effect of its propagation on our conclusions. (Lines 852-893)*

As explained in the original papers, and noted in our discussion (Lines 437-445), such predictions are clearly unsuitable for local applications (e.g. to estimate population sizes in a reserve). These predictions can, however, provide robust approximations for large-scale analyses—and they have repeatedly been used in large-scale analyses published in top ecological journals, including:

- Broekman et al., (*Conservation Biology*, 2022) to simulate the long-term effects of habitat fragmentation on mammal communities;*
- Gonzales-Suarez et al. (*J Animal Ecology*, 2020) to assess the role of brain size on species density*
- Santini et al. (*Ecography*, 2019) to test the centre-abundance hypothesis.*
- Santini et al. (*Ecology Letters*, 2020) to assess temporal trends in allometric scaling.*
- Sykes et al., (*Conservation Biology*, 2019) to assess how rarity moderates species' response to land use.*
- Tucker et al. (*Ecography*, 2020) to assess the impact of human pressures on species abundance.*
- Van Aeden & Dickman (2023) to estimate the impacts of Australia's megafires, published in a book by CSIRO, the Australian Government agency responsible for scientific research;*
- Wang et al. (*Global Change Biology*, 2021) to predict emerging infectious zoonotic diseases risk;*
- Wolff et al., (*One Earth*, 2023) for conservation planning and prioritization;*

These papers address large-scale questions in macroecology and conservation biology. They provide a clear precedent for using the population density estimates in global and regional analyses. In addition, the same estimates have been accepted by other parts of the ecological community: for example, they are now integrated into the sRedList platform (<https://sredlist.eu/#/home>), used to inform Global and National Red List assessments. The mammal and bird population density papers have been well cited since publication (39 times and 10 times, respectively). We are not aware of either receiving published criticism.

Regarding data quality for the study region, we disagree that the estimates are more uncertain for Africa. We address this issue below, further along in our response to reviewers.

The authors (sensibly) opt to incorporate uncertainty in body mass and 'natural' population density (but not apparently intactness) using an MCMC approach, and the credible intervals they provide cover an order of magnitude difference (Fig 1), though I rather suspect this is an underestimate given the bias in global datasets towards temperate systems. Similarly, the apparently lack of incorporation of variation in intactness measures (when I understand they are available) seems unwise and would likely even further impact these changes.

We agree that failing to incorporate variation in intactness measures would be unwise. Fortunately, we did incorporate this source of variation into our uncertainty simulations. We assumed there was there was uncertainty in the estimated intactness of each species in each land use, based on the data provided by Clements et al. (See Lines 867-871 of the draft text, as well as section B.5.1 of the r script analysis, entitled "Uncertainty in BII estimates"). To simulate the uncertainty around intactness values we drew values from a random beta distribution, generated using the mean intactness values and standard deviations recorded for each species. This resulted in 10,000 intactness values for each species in each land use simulating uncertainty around biodiversity intactness. To incorporate this source of uncertainty into the broader uncertainty analysis, we multiplied the 10,000 generated intactness values for each land use with the 10,000 generated energy consumption values for each species, the latter already incorporating uncertainty in body mass, population density, diet composition, assimilation efficiency, and in the allometric equation coefficients. (Lines 857-858; 867-871)

Because these sources of uncertainty must multiply, uncertainty is very large for any given species in a given pixel. However, the uncertainty around summed energy flows is much lower as more species are analyzed over a larger area. First, the uncertainty for each species is independent of other species, so that as the sum of energy flows through more species are analyzed in a guild or functional group, the proportional uncertainty around the energy flow decreases. Second, the proportional uncertainty around energy flows declines when summing the flows through many populations occurring over large geographic areas. This is because some sources of uncertainty are partially independent among the populations present in each 0.5 degree square. We explain how we accounted for this second decline in uncertainty the paper's Methods (Lines 873-893).

As for the BII intactness values, these were estimated through a rigorous and peer-reviewed structured expert elicitation process following the well-established IDEA protocol (Hemming et al. 2018). These values are not "little more than guesses." Rather, the co-production approach "provides a unique form of validation since experts were convened to critically discuss the validity of anonymised, aggregated estimates of intactness for different groups of species, and thereafter (anonymously) revise their independent estimates where they deemed it necessary. Confidence intervals around BII estimates reflect the degree of consensus between experts, providing insights into uncertainty by highlighting taxa and land uses for which knowledge is currently more limited or disputed." (Clements et al., 2024) (Lines 727-730)

Given the large uncertainty already involved, and the even larger uncertainty in the expert opinion intactness data the estimates must be considered little more than guesses. I question whether these data are suitable for the analysis, or simply build a house of cards.

The “house of cards” analogy suggests there may be underlying assumptions which, if proved weak, would lead to invalidation of our key conclusions. The referee’s concerns are mainly around properly scaling uncertainty, and how such robust propagation of this uncertainty in our analysis could increase or decrease the uncertainties in our quantified estimates. In an above section of our response to referee 1 (as well as in our methods – Lines 852-893), we have clarified the Monte Carlo approach we used to propagate the six sources of uncertainty that we identified. It is true that these sources of uncertainty multiply, leading to very high uncertainty around energy flows through any given species or any given cell. However, we think none of our core conclusions will be altered by any plausible expansion of uncertainty, hence this “house of cards” metaphor is inappropriate. We see no evidence that our overall analysis and narrative would be fundamentally changed by expansions in the size of the uncertainty. This is illustrated below, where we list each of our core conclusions with associated 95% confidence intervals.

1. Due to human activity, energy flows (i.e. food consumption) through wild African birds and mammals has decreased to 64% (54 – 74%) of historical values.

2. The greatest changes to trophic structure and ecosystem function resulted from the collapse of large herbivores and their associated functions (nutrient dispersal, grazing, and browsing). Large herbivores are just 27% (15% - 39%) intact. In contrast, birds are 71% (62% – 81%) intact, and other mammals are 71% (62% – 80%) intact.

3. The relationship between species richness and energy flow is fundamentally different for birds and mammals, with flows through birds and their associated functions more evenly supported by many species, and flows through mammals more dominated by a few keystone species including elephants, frugivorous primate, perissodactyls and fossorial rodents. The 5% most energy-consuming mammals consumed 72% (64% – 84%) of energy flows through mammals, while the 5% most energy-consuming birds consumed just 48% (41% – 59%) of flows through birds. In contrast, the bottom 50% of mammals consumed just 0.7% (0.4% – 1%) of mammal-mediated flows, whereas the bottom 50% of birds consumed 7% (4% – 10%) of bird-mediated flows.

4. Compared to richness and abundance-based metrics, an energetics approach highlights the importance of keystone species such as elephants, which contribute an outsized proportion of many ecosystem functions. In total,

elephants historically accounted for 9.9% (6.2% - 25.9%) of total energy consumption by birds and mammals across sub-Saharan Africa.

5. An energetics approach also highlights the ecological importance of smaller species, and their greater relative contributions to ecosystem function as large herbivores decline. Birds in particular historically contributed an outsized proportion of energy given their small body mass, accounting for 42% (32% - 47%) of energy in forests, 37% (23% - 45%) of energy in grassy systems, and 27% (13% - 37%) of energy in arid systems. Despite accounting for 37% of total bird and mammal energy flow, birds accounted for just 24% (15% - 36%) of total bird and mammal biomass.

6. Finally, the key ecosystem functions in landscapes, as well as the most important energetic pathways and largest contributors to declines in energy flows, are moderated by biome. Arboreal birds and mammals are more important in forests: historically, they accounted for 39% (30% - 45%) of energy flow in forests; 11% (7% - 15%) of energy flow in grassy systems, and 6% (3% - 10%) of energy flow in arid systems. In contrast, fossorial mammals show an inverse pattern: they historically accounted for 4% (1% - 10%) of energy in forests, 16% (7% - 36%) of energy in grassy systems and 32% (14% - 64%) of energy in arid systems. Large terrestrial herbivores were important across all systems: historically, the 90 mammalian large herbivore species accounted for 21% (13% - 48%) of energy flow in grassy systems; 17% (8% - 29%) of energy flow in arid systems and 17% (13% - 29%) of energy flow in arid systems.

To better communicate the uncertainty around the analysis, we have added 95% confidence intervals to values in the text. We have also added a new figure (Extended Data Figure 6) that shows confidence intervals around the fractional energy consumption by key species groups. (Lines 852-871)

I also wonder why the authors attempt it in Africa? If their aim really is to describe and test a new energetics based method of computing human impact, why not use a much better documented continent than Africa where the uncertainties (which must multiply) are so large I can't really believe much.

We focus on Africa for three reasons, which concern, (1) Africa's uniquely intact megafauna and (2) the disproportionate focus of studies on data-rich regions, and (3) data quality.

First, Africa presents a unique chance to assess the full suite of bird and mammal-mediated ecosystem functions across species' historical ranges. This is because Africa is the only continent with a relatively intact megafauna. Moreover, the populations of most African megafauna species contracted only in the last few centuries, allowing ecologists to reconstruct species' historical ranges with relative accuracy. On other continents, megafauna species and populations were extirpated millennia earlier,

creating a far greater uncertainty around species' historical ranges (Faurby and Svenning 2015). (Lines 132-136)

Second, too many ecological analyses are driven by a focus on, or bias towards, "data-rich" Europe and North America. Europe is highly depauperate in most taxa, and North America is depauperate in megafauna (Pedersen, Faurby, and Svenning 2023; Ripple et al. 2017). This spatial bias leads to a distorted understanding of animal ecological functioning and its intactness, a bias that understates how large mammals impact landscapes, biomes, and the earth system. We argue that the opportunity to address this bias, coupled with the strength of the BII dataset and of aerial survey data on African mammals, justifies our focus on Africa. We have clarified our justification for choosing Africa as a case study in lines 132-136 of the main text.

Third, we question the assertion that the data on population densities are less accurate in Africa. The database of population densities includes many estimates drawn from African studies (see Santini et al. 2018, 2022, 2024), and we have no reason to believe the quality is lower for Africa than other areas. In fact, for many large and medium-sized mammals, population counts are probably more accurate in Africa than in other regions, because African data is more likely to come from highly accurate aerial censuses. In many African reserves, managers monitor populations annually to combat poaching, a practice rarer in other regions. Uncertainty around small mammal estimates is high everywhere, and we have no reason to believe it is higher in Africa than elsewhere. For birds, it is true that there are less data than in other regions. But we do not see a reason to consider the African data less accurate, as virtually all estimates for birds (in Africa and elsewhere) use the same methodological approach: Distance sampling. It is true that the African data might cover a lower proportion of species than temperate zone data, but the estimated uncertainty for the data reflects such uncertainty (see e.g. methodology in Santini et al. 2022)). In any case, large and medium-sized mammals contribute about 30% of the total energy flow through African ecosystems (see Extended Data Table 2). We suggest that the higher accuracy for these groups in African ecosystems would offset any potentially lower accuracies for birds and small mammals, leaving total energy flows similarly accurate in Africa as elsewhere.

We also suggest our biodiversity intactness estimates are probably better in Africa than they would be in other regions. The expert-derived biodiversity intactness index for Africa is likely much more grounded than global products that estimate declines in species abundances. To our knowledge, this kind of grounded, local knowledge-based biodiversity intactness index is not available for other regions of similar size. Hence our focus on Africa rather than a global analysis.

Not only are the African data highly uncertain, but add to this as the authors note the lack of invertebrate data (we know that something like 50% of energy in some grassland systems is consumed by invertebrates, so this is not a trivial issue) and, indeed, the lack of consideration of fire as an energetic consumer in much of Africa, and I am left feeling this work needs significantly more context before it is useful (assuming the underlying

data issues are accepted). I don't think we can really interpret the loss of 30% energy cycling through a decline in a subset of vertebrates, when we have no idea what is happening to either invertebrate energy cycling or changes in fire? Is this decline really important?

We believe that addressing all birds and mammals at the scale of all of sub-Saharan Africa is a major step forward in our understanding for three reasons. First, many ecosystem functions (e.g. seed dispersal, nutrient flow) are dominated by birds and mammal effects. Second, this paper shows for the first time the general utility of an energetics approach, (as opposed to species diversity or abundance approaches), for mechanistically relating animal biodiversity to ecosystem function at regional and planetary scales. Third, we are able for the first time to make quantitative statements at the scale of a continent about how a suite of bird and mammal mediated functions have changed. We believe these are substantive and novel achievements, which present the power of a framework that can be more widely applied, criteria that make this paper suitable for publication in Nature. (Lines 129-131)

We agree that it would be valuable to try and apply such an approach to invertebrates. But this would probably be a major and multi-year project in itself, and the data uncertainties and gaps highlighted by this reviewer would be at least an order of magnitude greater than for birds and mammals. We would be interested in pursuing such approaches in future years, and if we are successful would certainly be submitting such work to Nature! (Lines 129-131)

Fire is a separate issue and outside the scope of this paper. The referee seems to be highlighting a comparison of the three main agents of vegetation consumption (vertebrates, invertebrates and fire). This is certainly an interesting question but outside the scope of this current manuscript, which investigates historical changes to bird and mammal trophic structures and bird and mammal-mediated ecosystem functions. The framework we present, however, does show a way forward in how to address the question of vegetation consumption at continental scale in the future.

We now carefully refer to bird/mammal herbivory rather than herbivory to avoid any assumed neglect of invertebrates.

Some more specific issues that might be useful to expand on:

(1) There's a degree of swapping between proportional change (most of the time) and estimates of absolute energy consumption (in the text). This could be clarified.

We have added a sentence clarifying the difference between absolute energy consumption and proportional changes in energy consumption. We have also clarified our use of these two terms throughout the text. We now explicitly refer to "absolute energy flows/consumption", whenever discussing absolute flows. (Lines 155-157)

(2) Similarly, there is a confusion over when energetics per se are being cited, and when this is being provided as an ecosystem function. For example, is granivory an ecosystem function (as implied in Fig 1)? Or a trait? Or an energy flow? The writing and description of what is simply energetics and what (like seed dispersal) is really an ecosystem function needs to be much clearer. Why not stick with just energetics and quantification of flows? I think, for example, that seed dispersal impacts of, e.g. empty forests has been estimated much more comprehensively and accurately elsewhere and the authors may be well advised to decide what they really care about.

We agree that our terminology is confusing and have sought to clarify it. In all cases, we quantify energy flows through groups of species: either species in a trophic guild or species that perform an ecosystem function. We now state early in the text (Lines 157-160) that when we refer to the energetic intactness of a particular function, this is a shorthand for “the intactness of energy flows through birds and mammals performing that particular function.” We adopt this shorthand to streamline the text.

We also acknowledge that we quantify the strength of ecosystem functions using an imperfect proxy: the energy flows through the species that perform a ecosystem function. Our proxy approach assumes that all animals in a functional group contribute equally to that function, after accounting for differences in body size, population density, and assimilation efficiency. Our approach does not consider how behavioral differences cause some animals to perform functions more efficiently than other similar species (a caveat we state in lines 151-155). In adopting these assumptions, we follow the methods of other papers that have measured ecosystem functions and are published in high impact journals, including by Doughty et al (Nature Geosciences 2013), Pedersen et al., (Global Ecology and Biogeography 2023) and Antunes et al. (Trends in Ecology and Evolution, 2024). We now note these assumptions in the text. (Lines 151-155, 157-160)

In future work, it would be good to finetune relationships between energy flows and ecosystem function by integrating behavioral information. But this behavioral data would be different for each function, and for many functions is not available for most bird and mammal species. We believe that our current, proxy-dependent approach represents an important step forward, because for the first time it allows ecologists to compare the strength of many functions on a continental scale. In doing so, it provides a blueprint for future studies to integrate more detailed behavioral information.

(3) vibrancy is cited, but needs explanation here

We have cut the use of the term “ecological vibrancy” after deciding that this term is better introduced and defined elsewhere.

(4) I'm also puzzled by Fig 1 showing apex carnivory is more intact in unprotected than protected areas. (Also grazing & browsing, but maybe that is domestic stock?) Can this be explained?

Good catch, thank you. We accidentally mislabeled the horizontal axis title in this figure. We switched the titles for protected areas and untransformed lands. Energy flows through apex carnivores are indeed higher in protected areas than in near-natural lands. (Fig. 1)

(5) There seems to be a fundamental problem with spatial resolution of historic analysis - the polygons used are so much smoother than current ones, because they come from a very different methodology, that there are necessarily going to be large declines estimated for pretty much every species. I think some sort of sensitivity to smoothing needs to be applied. For example, each successive IUCN / BirdLife release of maps almost inevitably reduces the estimated areas of occupancy, because some new technique has been developed to exclude unsuitable habitat from the polygon (e.g. the major changes when altitude limits were stamped on all the maps relatively recently). I would like to see how much of the declines estimated are a result of refining range maps, rather than actual range change per se.

None of the estimated declines in species energy flows are the result of refining range maps. It is true that successive IUCN/ BirdLife maps reduce areas of occupancy. However, our approach uses only one set of IUCN/ BirdLife range maps: the most recently released set of maps showing historical species ranges. To estimate the change in species ranges under contemporary land use conditions, we do not use a separate set of IUCN/Birdlife maps. Rather, we apply to the original maps the changes in species abundances estimated by the Biodiversity Intactness Index for Africa. These changes in species abundances are estimated based on (i) current land use across Africa at the scale of 8x8 km² and (ii) the expert-estimated impact of land use change on the abundance of each species.

In response to referee comments, we have now applied a habitat filter to the historical IUCN/Birdlife range maps, following Beyer and Manica (2020). (Robert Beyer is now a co-author) To do so, the vector range map for each species was first divided into 1/12° grid cells. Each grid cell was assigned a biome (e.g. grassland, savanna, forest) based on its natural vegetation cover in the absence of human land use change. Grid cells were excluded from a species' range wherever a grid cell's biome was not included within a species' habitat preferences. To facilitate the analysis of ~3000 species, the 1/12° grid cells were upscaled to 1/2 ° cells, and cells were excluded from species ranges whenever no suitable habitat was available for a species within a cell. Using the habitat-adjusted Beyer and Manica (Beyer and Manica 2020)) maps, instead of the raw IUCN/Birdlife maps, improves the analysis by standardizing the scale at which habitat is used to demarcate species ranges. (Lines 696-706)

(6) Labels for fig 4 c & d are in text and a and b

Fixed.

I'm not totally convinced and the section on restoration restoration section is pretty unconvincing - what recommendations here are not what people are currently aiming at?

The referee correctly points out that some of our restoration recommendations were not novel. We've therefore reframed our restoration section to better highlight the unique added value of an energetics approach (lines 363-383). In particular, we believe energetics provides a novel and useful way to quantify the recovery of ecosystem functions in situations where species composition has changed substantially. When ecosystems recover, the allocation of biomass among species often changes dramatically, even among species performing the same ecosystem function, and even once the ecosystem's original animal biomass is reattained (See e.g. the recovery of large herbivores in Gorongosa National Park – Daskin et al., 2016) As a common currency, energy flows can be used to quantitatively compare ecosystem a broad suite of functions in recovering animal communities when species composition has changed. This can help restoration decision makers integrate functionalist approaches to restoration into traditional, compositionalist approaches. We have reframed the restoration section to highlight this novelty. (lines 363-383)

Equally, if this is to be useful for discussions around planetary boundaries, don't we need to see how the authors would recommend how to pick one in the face of all the uncertainty in the metrics? It will presumably need to be an absolute metric, not a relative one - but that isn't given much discussion.

The planetary boundaries framework is not a focus of this paper, and is questionable for any diversity metric (as we now state in lines 396-406). However, if a biodiversity planetary boundary is to be applied (as is widely done in much influential literature), we argue that energy flow through key taxa or functional groups might be a more functionally appropriate approach than previously applied metrics such as extinction rate, biodiversity intactness or HANPP. All of these metrics have arbitrary thresholds (e.g. 90%, 30%) based on expert judgment rather than on any mechanistic basis. We also believe our approach directly addresses a gap noted in the most recent planetary boundaries paper (Richardson et al. 2023, Science Advances), which states that "the link of BII to Earth system functions remains poorly understood [as] BII cannot be directly linked to the planetary biogeochemical and energy flows relevant for establishing Earth system state." (lines 396-406).

(7) The authors themselves say "the energetic approach presented here, ..., is highly inaccurate at predicting values for individual species or landscapes" They then continue "Rather, these estimates are useful for large-scale taxonomic and geographic comparisons when values are aggregated across many species and landscapes, and can elucidate how humans are impacting ecosystem function at scales relevant for policy." but I suggest that this is somewhat like suggesting lots of bad data will lead to sensible policy-relevant decision making.

Sometimes we do need to aggregate for policy relevance (as indeed the BII does) and aggregation over multiple taxa with independent sources of error can indeed result in

reduced relative error the total, a key principle of error propagation. This does not of course apply if biases are systematically in the same direction across large numbers of taxa. But we believe that systematic biases are unlikely, based on the robustness of our error propagation analysis, and on our sensitivity analyses, which investigated the impacts of changes from flat to variable species population densities and in changes to range map habitat scale.

Referee #2

This manuscript proposes a new metric for quantifying the roles of animals in assessing broadscale ecosystem function and applies this in a case study of sub-Saharan African birds and mammals. The authors postulate that a comparison of energy flows through historical vs. contemporary animal communities provides a measure of change in ecosystem function and explore changes across different trophic guilds, animal functional groups, biomes and in areas of different land use. Their novel approach makes use of two new datasets and draws on a wide variety of other datasets. They describe implications for restoration, global policy and priority-setting, and further research.

I find this approach novel and exciting, and believe that this manuscript presents a major step forward in the field of biodiversity conservation. To date, no broad-scale measure of landscape functionality has previously been available in the global suite, which has resulted in this aspect of ecosystem condition being poorly represented in policy-relevant assessments such as IPCC and IPBES. The paper has been very well-written, the arguments and results clearly presented and the referencing appropriate. The methods and results are complex and I suggest ways to improve their clarity and ease of interpretation. There are also some important caveats that I believe have been neglected, and the extent of some of the bold claims reduced - I make further suggestions. Overall, I think the results are sufficiently robust, at least to an equivalent degree to other established studies and indices if this type. This is excellent work, of global relevance and I recommend its publication if the issues below can be addressed.

We thank the referee for recognizing and highlighting the novelty and robustness of our approach—in particular, the novelty of producing a broad-scale measure of animal-mediated landscape functionality. Beyond this particular study, we think we have demonstrated a pathway for incorporating animal-mediated ecosystem functions into policy-relevant assessments more broadly, and perhaps ultimately into the IPBES or IPCC assessments. (Lines 388-395).

Recommended actions

1. The workflows for deriving metrics of energy flow are many-stepped and require detailed scrutiny of long descriptions to decipher. I had to draw a diagram to see how it all fits together. Since a major point of the paper is to propose a new method, I suggest making this more accessible to readers, for example by adding a schematic figure. This could detail steps from acquiring/compiling species range maps through abundance estimation, adding BII, working from body mass through to consumption and then on to

functional groups and guilds.... Etc. The text could then refer to them by name or number, making it easier to follow, and providing the opportunity link with explanations, references, dataset examples/suggestions, caveats and suggestions.

Following the referee's advice, we have included a methods diagram in the paper (Extended Data Figure 7).

Caveats. There are so many steps and nested datasets and layers on which the final measures and claims are built that the idea of a house of cards comes to mind. I've needed to work step-by-step through your steps (and my figure) to assure myself that I've considered the limitations at each. Considering those who may follow your approach based on this paper, what would they need to know, do or not do, bear in mind in interpretation, etc. to use it responsibly? You have bits and pieces of caveats scattered through the ms (e.g. Lines 274-277, which are particularly important but not in the caveats section). I think your caveats section needs to feature more prominently, including both as a short section in the main text and a more thorough and systematic section under methods. With future external users in mind, this could talk systematically to each application step.

Following the referee's advice, we have added a short caveats section in the main text (Lines 411-445) as well as a more thorough caveats section in the methods (Lines 897-966). This section systematically states the caveats associated with each included dataset and analysis. We have also used this section to propose future analyses that can be taken to address these caveats and improve large-scale energetics analyses if new and improved datasets become available.

Beyond the dataset limitations referred to here and in other underlying datasets from other papers, here are a few additional ones that I believe should be included:

a. IUCN Red List range data. Lns 601-608. It isn't clear whether the authors recognise that the IUCN polygons used vary widely between species and geographic sub-regions in terms of their mapping resolution and accuracy. Ranges of well-known birds and mammals in highly-observed areas, for example, have been mapped accounting for fine-scale habitat heterogeneity (in contradiction to the text), already account for area-of-habitat, and have polygons that identify now-extinct range sections (i.e. good approximation of historical range at circa. 1700). The majority of other sub-Saharan animals, however, are broader range estimations, closer to extents-of-occurrence, and there is no knowledge or inclusion of now-extinct range sections. IUCN does not have /provide meta-data enabling users such as the authors to know which is which, or how these vary across regions, even for a single species. As a result, historical range extent inaccuracies (considered independently of other variables) could lead to over-estimating decline in poorly known geographic areas, and underestimating decline in poorly-known taxa. This applies and has implications both for this and the BII Africa study and should be discussed.

Thank you for this comment. As the referee says, it is well established in the literature that to reduce the coarse approximation of range maps one should combine them with available habitat area maps, derived from land cover maps and the habitat preferences of species (Lumbierres et al. 2022). We have therefore switched from using “raw” birdlife/IUCN maps, to using the higher quality area of habitat maps presented in Beyer and Manica (2020). We have also added Robert Beyer as a co-author because of data and his intellectual contribution to this response.

These Beyer and Manica maps refine the IUCN/birdlife range maps, by eliminating area that does not align with a given species’ habitat requirements. To create these area of habitat maps, Beyer and Manica first aligned the biomes used in land cover maps with the habitat preferences of each species, as reported by the IUCN. Then, the vector range maps for each species were divided into 1/12° grid cells. Each grid cell was assigned a biome (e.g. grassland, savanna, forest) based on its natural vegetation cover in the absence of human land use change. Grid cells were excluded from a species’ range wherever a grid cell’s biome was not included within a species’ habitat preferences. To facilitate the analysis of ~3000 species, grid cells were upscaled to 1/2 ° cells, and excluded from species ranges when no suitable habitat was available for a species within a cell. For a few small-range species, 1/2° scale habitat-adjusted range maps eliminated all available range; for these species, unadjusted IUCN/Birdlife ranges were used. Adopting the habitat-adjusted Beyer and Manica (2020) maps, instead of the raw IUCN/Birdlife maps, improved the analysis by standardizing the scale at which habitat is used to demarcate species ranges.

Adopting the new area of habitat maps did not substantially change the study’s results (Figure R1). In no case did the intactness of energy flows through an ecosystem function group change by more than 2.5%. This analysis demonstrates that the study results are not highly sensitive to changes in the resolution of the underlying range and habitat maps, a concern raised by some of the referees. We have added language explaining these choices, and their consequences, to our methods and Caveats section (Lines 696-706, 913-921).

a) Intactness of energy flows using unadjusted IUCN/Birdlife speceis range maps

Ecosystem Function	Intactness of energy flows using unadjusted IUCN/Birdlife speceis range maps				
	All	Protected Lands	Untransformed Lands	Croplands	Settlements
Pollination	65%	96%	63%	25%	31%
Seed dispersal (frugivory)	62%	92%	58%	14%	24%
Granivory	74%	96%	73%	86%	38%
Insectivory	72%	96%	70%	41%	33%
Soil formation	75%	94%	95%	72%	27%
Meso-carnivory	72%	95%	72%	52%	34%
Apex carnivory	32%	69%	34%	8%	4%
Scavenging	61%	89%	63%	43%	14%
Grazing and browsing	27%	58%	29%	8%	4%
Nutrient dispersal	26%	58%	27%	8%	3%
Total bird	71%	94%	71%	50%	37%
Total mammal	60%	85%	65%	38%	20%

b) Intactness of energy flows using habitat-adjusted IUCN/Birdlife maps from Beyer and Manica (2020)

Ecosystem Function	Intactness of energy flows using habitat-adjusted IUCN/Birdlife maps from Beyer and Manica (2020)				
	All	Protected Lands	Untransformed Lands	Croplands	Settlements
Pollination	64%	96%	61%	25%	31%
Seed dispersal (frugivory)	62%	92%	58%	14%	24%
Granivory	75%	96%	74%	85%	38%
Insectivory	72%	96%	70%	40%	33%
Soil formation	75%	94%	95%	77%	26%
Meso-carnivory	72%	95%	72%	50%	34%
Apex carnivory	32%	69%	34%	8%	4%
Scavenging	59%	89%	61%	64%	13%
Grazing and browsing	27%	58%	29%	8%	4%
Nutrient dispersal	26%	58%	26%	8%	3%
Total bird	71%	94%	70%	51%	37%
Total mammal	59%	85%	64%	38%	19%

Figure S1. Switching from unadjusted IUCN/Birdlife range maps to habitat-adjusted IUCN/Birdlife range maps does not substantially alter the study's results.

b. Ln638: cells in PAs are assigned an intensity class of 1, but what about the species groups for which PAs cannot be assumed to represent a healthy historical state (e.g. vultures; human-hunted/poached animals)

A score of 1 represents species abundances in a healthy historical state, rather than in modern protected areas. Some experts chose to envision abundances in exceptionally well-managed protected areas as a proxy for healthy historical ecosystems (Clements et al. 2024). But they produced separate estimates for protected areas in general, which account for mis-management, poaching, and range size effects. The result is that for large, wide-ranging species (e.g. vultures, large carnivores, mega-herbivores) experts estimated that intactness scores within protected areas are substantially below 1. (Lines 727-730)

c. The BII's limitations, given its prominent role in this study

We've added language in our new, expanded caveats section (Lines 935-949) that explains the BII's limitations. The key limitations of the BII concern local and national-scale analyses, because the BII does not account for national-scale factors such as war, protected area management capacity, and cultural attitudes toward hunting. We

argue that in our sub-continental scale study, these limitations are less important. We state in the text that neither our study nor the BII should be used to assess ecosystem change at local scales. (Ls 437-445). In addition, we integrate into our uncertainty analyses the key source of uncertainty within the BII: the uncertainty around the expert-derived BII scores, which estimate how land use affects species abundance (Lines 867-871).

Minor corrections

Fig 1. Labelling of panels inconsistent format (i.e. needs to be (a) not a.)

Fixed.

Fig 2: a fabulous figure! (OK, so this isn't a correction)

Thank you!

Fig 3: Lns 558 560. Missing a full-stop and the first (d) should be a (c). I suggest adding the bird and mammal icons to the graphs themselves for ease of interpretation.

We've edited the text and added the bird and mammal icons to the graphs.

Line 682: suggest changing to "...Unlike for birds, there is so single authoritative source..."

Fixed.

Line 326: restoring forest birds and primates is seldom feasible – I suggest instead recommending prioritising conservation of or avoidance of loss of..."

We've reframed our section on restoration more broadly, in response to comments by referee 1. These revisions eliminated language on restoring forest birds and primates. (Lines 363-383).

Referee #3

The Loft et al. study addresses the consequences of changes in bird and mammal populations for energy flows through ecosystems in Sub-Saharan Africa. The authors elegantly integrate new datasets on animal population densities, biodiversity intactness (BII) and energy consumption to provide a picture of where and by how much energy flows have declined during the past approximately three centuries. The uncertainty analysis performed – in which uncertainties of the various databases are combined – seems sophisticated. The authors find an overall decline in energy flows of one third. Main (and most interesting) results are that (1) mega-fauna reduction and extinction have had tremendous impacts on energy flows through Sub-Saharan ecosystems and (2) small mammals are on the rise both in density and in their contribution to ecosystem

energetics. Overall, the study convincingly shows that an energetics approach can be applied at continental scale to quantify anthropogenic changes in ecological processes and is a valuable addition to other metrics (e.g., carbon stock, NPP, GPP) because it explicitly considers the role of animals.

We thank the referee for highlighting the value of our energetics approach as an addition to other large-scale metrics of ecosystem function, as well as for complimenting the sophistication of the uncertainty analysis. We think that an energetics approach presents a novel and useful addition to macro-ecology and earth system science, in that it explicitly considers the role of animals in ecosystem function over large spatial scales. (Lines 479-489).

1. Ecosystems & land use classes. The study targets energy flows through 'ecosystems', suggesting that is restricted to (semi-)natural ecosystems. Yet, the study also includes cropland and settlement. In my view, this is problematic for three reasons: (1) The inclusion of croplands and settlements in this study seems to be at odds with the use of the terms 'ecosystem' and 'biome'. In Fig 2 and elsewhere biome names are used, e.g. stating that "energy flows are estimated to be [...] intact in forests..." (L 164-5), suggestion that that the analysis is limited to forests, but I guess in the practice of this study the 'forest biome' also includes croplands and settlements. (2) I do not think including croplands and settlements is warranted because the purpose of the study is to assess how shifts in animal abundance affects the energy flows in 'ecosystems'. If the land cover changes from a (semi-)natural vegetation to crops (or to settlements), one does not assess shifts in energy flows through changes in animal abundance alone, but also due to the shift in the plants (crops) growing there. This, in my view, goes beyond the modification of ecosystems. (3) Mammal and bird species that survive in highly altered systems such as settlements and croplands very likely can do so because of (partial) dietary shifts. As far as I can tell, such dietary shifts have not been included in the study. This likely induces bias in the estimates of energy flow in settlements and croplands), which is not accounted for. In sum, I would argue to leave out croplands and settlements from the analyses (also from maps) to provide a more clear focus. One could consider to include agroforestry-type systems in the analyses, if possible.

We understand the referee's concern about our decision to classify anthropogenic landscapes (i.e. croplands and settlements) as parts of ecosystems and biomes. The referee is right that these landscapes have been dramatically transformed. We can see how it might be confusing to write that energy flows have declined by x% in a biome, when a portion of that loss is due to the biome's conversion into croplands. When we write about declines in forests, for example, we mean areas that were historically forests (in the 1700s baseline), but might now include anthropogenic land uses. To clear up this confusion, we have carefully edited the text to read "historical biomes" (e.g. historical forests, historical grasslands), when we discuss anthropogenic landscapes occurring in once-natural ecosystems.

However, we have chosen not to omit croplands and settlements entirely, because we believe that doing so would decrease the study's value to ecologists and policymakers.

We calculated that conversion of natural habitat to cropland and settlements is responsible for 25% of the total decline in energy flows through birds and mammals across sub-Saharan Africa. The conversion of natural habitat to croplands and settlements is a major cause of biodiversity loss in the region (Powers and Jetz 2019). We think it is important to understand how such biodiversity loss changes ecosystem function, including in areas that have been converted from natural to anthropogenic landscapes. Excluding such categories from the analysis would preclude us from making statements about region-level changes, for example that energy flows have declined by 36% through African birds and mammals. It would also prevent us from assessing how a major driver of biodiversity loss (land use change) is altering ecosystem function across Africa.

We agree with the referee that it is valuable to understand the impacts on plant energy flows of converting natural ecosystems to croplands. However, such an analysis of plant energy flows is outside the scope of this study, and indeed has been undertaken elsewhere within studies estimating human appropriated net primary productivity or HANPP (see e.g. Haberl, Erb, and Krausmann 2014; see also our supplementary discussion). We think that assessing changes to bird and mammal-mediated energy on a continental scale marks a novel and significant contribution to understanding how biodiversity loss impacts ecosystem function. Many ecosystem functions are dominated by birds and mammals, including seed dispersal, grazing and browsing, and nutrient dispersal (Sekercioglu 2006; Pringle et al. 2023; Malhi et al. 2016). These functions can substantially impact agricultural systems, by facilitating ecosystem services and disservices. We therefore argue that it is important to estimate how these functions change when natural ecosystems are converted into croplands and settlements (Lines 446-449).

Finally, the referee suggests that mammals and birds survive in anthropogenic landscapes by altering their diets. We do include a random uncertainty in our assignment of diet preferences, which leaves some room for potential dietary shifts in our uncertainty estimates (Lines 866-867). However, because we focus on the energetic uptake of various food pathways, the only way changes in diet affect our calculations is through variations in assimilation efficiency. The assimilation efficiency is very similar among most food types (around 70-90%; see Table S3), with the notable exception being leaves, which are less nutritious and palatable (assimilation efficiency around 30-50% in mammals, apparently higher in birds where they are a minimal part of diets) (Malhi et al. 2022). Hence the only dietary shift that would have a noticeable impact on our calculations would be one from leaves to other foodstuffs. There is very unlikely to be a major shift in diet from leaves to other food types because of constraints of feeding and gut morphology. To test the sensitivity of bird-mediated energy flows to dietary shifts, Malhi et al., (2022) assessed the impacts of a 30% shift toward arthropods among mixed feeder bird species. The increase in arthropod consumption occurred at the expense of seeds, nuts, and fruits. They found that total energy consumption increased by just 2.3% compared with no shift in their diet. We therefore suggest that the potential changes in diet composition are (i) accounted for within our wide uncertainty intervals and (ii) unlikely to change the paper's core conclusions.

2. Changing energy flows and ecosystem services. I seriously doubt whether inferences about shifts in ecosystem functions can be made based on changes in energy flows. I read the justification in the Methods section (L707-710) and understand why the authors have opted not to estimate 'functional efficiency'. Yet, without any evidence that a reduction in energy flow through animals involved in a certain ecosystem function (for example seed dispersal) affect that service (number or mass of seeds dispersed), the results of shifts in energy flows through seed dispersing animals are hard to interpret. In spite of this limitation, the authors do present and interpret the shifts in energy flows as shifts in pollination, dispersal, etc. For example, L205-206 read "Seed dispersal is just 58% intact in untransformed lands and 14% intact in croplands." and the summary (L 32-33) mentions "with functions performed by megafauna in particular collapsing outside protected areas". So, implicitly (or perhaps explicitly?) one-to-one relations between shifts in energy flow and those in ecosystem services are assumed without support.

We agree that our terminology is confusing and have sought to clarify it. In all cases, we quantify energy flows through groups of species: either species in a trophic guild or species that perform an ecosystem function. We now state early in the text that when we refer to the energetic intactness of a particular function, this is a shorthand for "the intactness of energy flows through birds and mammals performing that particular function." We adopt this shorthand to streamline the text. (Lines 198-207)

We also acknowledge that we quantify the strength of ecosystem functions using an imperfect proxy: the energy flows through the species that perform an ecosystem function. Our proxy approach assumes that all animals in a functional group contribute equally to that function, after accounting for differences in body size, population density, and assimilation efficiency. Our approach does not consider how behavioral differences cause some animals to perform functions more efficiently than other similar species (a caveat we state in lines 198-202). In adopting these assumptions, we follow the methods of other papers that have measured ecosystem functions and are published in high impact journals, including by Doughty et al (Nature Geosciences 2013), Pedersen et al., (Global Ecology and Biogeography 2023) and Antunes et al. (Trends in Ecology and Evolution, 2024). We now note these assumptions in the text (Lines 198-202, 1456-1464).

In future work, it would be good to finetune relationships between energy flows and ecosystem function by integrating behavioral information. But this behavioral data would be different for each function, and for many functions is not available for most bird and mammal species. We believe that our current, proxy-dependent approach represents an important step forward, because for the first time it allows ecologists to compare the strength of many functions on a continental scale. In doing so, it provides a blueprint for future studies to integrate more detailed behavioral information.

3. Biodiversity and ecosystem energetics. Part of the analysis and results (Fig 4) is devoted to relating biodiversity to energetics. Clearly, this is an interesting component of the study, and it is logical to verify to what extent changes in species abundance (BII) predict comparatively large changes in energy flows. Yet, the strong relationships and high R² values for Fig 3a-b should not come as a surprise because the shifts in ecosystem energy flows is (i.e., the y-axis) is calculated using values of biodiversity intactness (the reverse of BII, i.e., the x-axis). They are now interpreted (L262-264) as “biodiversity intactness is a strong predictor of total energetic intactness”, but given the dependency this a methodological association rather than one that provides insights into the relation between diversity and energetics. I suggest to reframe this part of the text, by asking the question to what extent biodiversity intactness is a good predictor of shifts in energy flows, and for what groups it is not. That way, the added value of the energetics approach compared to biodiversity-related metrics can be highlighted and discussed.

We thank the referee for this excellent suggestion. We have reframed the text to ask for which groups biodiversity intactness is a good predictor of shifts in energy flows. We have distilled three key points. First, for ecological functioning overall, BII and species richness are good proxies for energy flows through birds, because of species redundancy and small body size range among birds (Lines 582-588). Second, this is not the case for mammals, because the much wider range in body sizes, population densities, and assimilation efficiencies means that some mammals are keystone species that contribute disproportionately to total energy flows (Lines 674-678. We've added a quantification of the different distributions of energy among birds and mammal species (lines 662-683). Third, when we look at individual functions, there are some where BII or species richness are good proxies, some where they are noisy proxies, and some where they are poor proxies (Lines 568-582). We have made these points clearer in the text (Lines 568-588, 662-683).

4. Uncertainties. While the uncertainty analysis overall appears to be solid, it does not address two sources of uncertainty. (1) L800-801 states that “we were unable to model geographic variability in population densities within species”. This may be problematic for the most important species in terms of energy flows, e.g. elephants. For several of those taxa, species-specific density estimates will likely be available. I'd advise the authors to conduct a robustness analysis for some of the key taxa, in which they replace the ‘flat geographic population densities’ used now by spatially explicit species-specific population densities to calculate shifts in energy flow. If the overall shift in energetics are robust to this change in densities used, then this provides support for the current assumption of flat densities.

To calculate the energy flows through African birds and mammals, we used a single, range-wide or “flat” estimate of each species’ population density. In reality, intra-species population densities vary across species’ ranges, according to environmental and biotic variables: they are not “flat” but “variable”. We used flat densities, because there are only a few species for which there are available a sufficient number of estimates to run species-specific models accounting for environmental and biotic variables (e.g. Pranzini,

Bertolino, and Santini 2023; Hempson, Archibald, and Bond 2015). We assumed that our simplifying assumption would not substantially alter our results, hypothesizing that intra-species variation in population densities would even out when summing energy flows across hundreds to thousands of species. (Lines 1396-1409)

To test this assumption, we ran a sensitivity test, which calculated the declines in energy flows through 92 large herbivore species for which spatially variable population densities are available over the whole of sub-Saharan Africa (Hempson, Archibald, and Bond 2015). The 92 analyzed large herbivore species account for over one quarter of the total energy flow through mammals, and are therefore a meaningful proxy for the larger set of species assessed in the main analysis. We calculated energy flows twice: first, using the spatially explicit variable densities provided by Hempson et al.; and second, using flat densities calculated by averaging the spatially-explicit densities, and then applying average values across the whole of each species' range. We calculated the difference between (i) the total range-wide energy flows through the guild; (ii) the range-wide energy flows through each individual large herbivore species, and (iii) the guild-wide energy flows through our four major land uses and three major biomes. (Supplementary Discussion, Table S1)

We found that assuming flat densities did not meaningfully change the decline in energy flow across any of these categories, except for a few individual species (Table S1). The total, range-wide energy flow through large herbivores was 0.8% more intact when calculated using flat densities than using variable densities (25.75% versus 24.96%). In no land use or biome was the difference in energy flows greater than 1%. In addition, the difference in intactness was less than 1% for 72 of the 92 assessed species, with the largest differences apparent for sable antelope (*Hippotragus equinus*, 4.9%), impala (*Aepyceros melampus*, 4.8%) and steenbok (*Raphicercus campestris*, 4.3%). These results indicate that assuming flat densities does not meaningfully change the results of the study, at least for the important large herbivore guild. The results further suggest that although many landscapes have been highly transformed by humans, these transformed landscapes do not correlate with regions that feature exceptionally high or low intra-species population densities across a wide suite of species. In other words, intra-species population densities do not vary consistently across species in a way that correlates with highly transformed or protected landscapes. We therefore conclude that using flat densities does not substantially change the key results of our studies. We present these findings in the supplementary discussion.

Table S1. Differences in energy flows through 92 large herbivore species calculated using flat range-wide average population densities versus variable, spatially explicit population densities. For the calculations using flat density, each species was assumed to occur in its range-wide mean population density across its entire range. For the variable population densities, we used the spatially explicit estimates in Hempson et al. (Science, 2014); these densities varied across the range of each species according to environmental variables (e.g. climate, soil, biome).

Group	Intactness –flat densities	Intactness – variable densities	Difference in intactness
Total energy flow (large herbivores)	25.75%	24.96%	0.78%
Mean energy flow for 92 large herbivores species	32.94%	32.88%	0.07%
Land Use			
Settlements	3.68%	3.68%	0.00%
Croplands	5.87%	5.87%	0.00%
Untransformed lands	26.90%	26.2%	0.70%
Protected lands	58.90%	58.90%	0.00%
Biome			
Forest	29.99%	29.15%	0.84%
Grassy	24.60%	24.25%	0.35%
Arid	27.78%	28.39%	-0.60%

(2) Historical population densities: ‘historical’ estimates are also influenced by anthropogenic effects, as mentioned in Santini et al (ref 24 in ms): “Our estimates are likely to be influenced by anthropogenic pressure”. Which means that for many species historical estimates are likely underestimates. I have the impression that this possible bias has not been accounted for in the analyses. What would the effect be?
Other comments

It is true that the species population densities we observe today cannot be considered intrinsic properties of species, even within protected areas. But apart from including uncertainty around population densities in our error estimation (which we have done), this problem is challenging to address. Species population densities do not respond consistently to human impacts: many population densities have declined, but many others have increased, at least locally, in human-modified landscapes, or in near-natural landscapes in which predators or competitors have been eliminated (see e.g. Tucker et al. 2021). We now state this caveat text in our Methods section (Lines 1203-1206).

- Summary paragraph: would be good to mention in a few words what data sources are used.

We have added this text. (Lines 33-34)

- Ecological vibrancy: to me it remains unclear how this is different from the quantification of energy flows. (The definition in L117-118 remains rather vague.) I would advice removing the term (or defining it very clearly).

We’ve eliminated the term “ecological vibrancy” throughout the text.

- L 193-195: I read (Worldbank Open Data: <https://data.worldbank.org/indicator/AG.LND.AGRI.ZS?locations=ZG>) that about half of Sub-Saharan Africa has an agricultural use. How does that relate to the 80% of untransformed & unprotected land?

This is a good question, which we investigated in response, and to which we provide a two-part answer. First, untransformed and unprotected lands include both rangelands and near-natural lands. Rangelands contain livestock and are thus classified by the World Bank as agricultural lands. Second, a cell is only classified by the BII as “cropland” when crops cover at least 20% of the cell. Consequently, some areas classified by the World Bank as having an agricultural use will fall into the unprotected, untransformed land category, in cases where cropland covers less than 20% of a cell.

- L 253: I find the conclusion that “fossorial rodents may be similarly impactful belowground” (compared to elephants aboveground) to be tricky and poorly supported by evidence, given the abovementioned (L244-249) extremely large uncertainty range. I’d argue that such a claim would only be supported if a realistic low estimate for fossorial rodents still would put them at a similar level (order of magnitude scale) as elephants.

We have changed the text so that it is far more conservative about mole rat energy flow estimates, and so that it better highlights the high uncertainty around these numbers. We have also added a figure (Extended Data Figure 6), which clearly demonstrates the exceptionally high uncertainty around the proportion of total energy consumption accounted for by fossorial mammal. We have also added a note to Figure 2, clarifying that uncertainty around fossorial mammals is exceptionally high.

Underlying our results is a strong but highly uncertain signal that small fossorial mammals have high population densities. This signal is based on relatively few empirical population density counts, as fossorial mammal densities are extremely difficult to measure. We think it is possible that fossorial rodents do consume a substantial proportion of energy, particularly in more arid ecosystems, where plants allocate a large proportion of their biomass underground, and where burrowing is a common strategy to escape extreme temperatures (Eldridge and Soliveres 2023). Given this possibility—and we now make clear that it is only a possibility—we have reframed the text to highlight the need for further research on fossorial mammal energetics (lines 279-282, 551-559).

- L 613: good to add for how many species these 10 thousand estimates were done, and that these were then extrapolated to other species

We’ve added this information into the methods. Globally, these estimates were done for 737 mammal and 1853 bird species (2590 species total). They were extrapolated to the other species using the models described in Santini et al., (2022, 2023). (Lines 1218-1220)

- L 375-378. I value the discussion on proxies for biodiversity to be used in large-scale assessments and analyses of planetary boundaries. I wonder to what extent (geographic) patterns of energetics shifts compare to those of HANPP. Would it be possible to substantiate (and expand) this discussion by adding a Suppl Fig with such a comparison? That would illustrate how and where an energetics approach would be more meaningful than a NPP approach.

We have added supplementary discussion that (a) quantitatively assesses the spatially explicit relationship between patterns of HANPP and animal energetics shifts, and (b) hypothesizes why these shifts are only weakly and noisily correlated, and thus why there is added value in using an animal energetics-based metric alongside a HANPP-based metric to assess changes to biosphere integrity (See Supplementary Discussion). For the sake of convenience, that discussion is reproduced here:

Human-appropriated net primary productivity (HANPP) has recently been proposed as suitable metric to use for determining a planetary boundary for biosphere integrity², replacing the previous metric, biodiversity intactness (BII). HANPP estimates the amount of potential plant energy appropriated by humans through agricultural harvests and land use change³. A proportionate HANPP value (between 0% and 100%) can be calculated by dividing the absolute HANPP in a cell by the absolute potential NPP of a cell. We suggest that HANPP is a reasonable proxy for how much humans have impacted the plant component of the biosphere, given that HANPP measure the proportion of plant energy that humans have appropriated. In contrast, HANPP is unlikely to accurately capture change to the integrity of the animal component of the biosphere, for three reasons. First, populations of different animal species, and their associated ecosystem functions, decline asynchronously in response to different kinds of land use change. For example, rising HANPP in rangelands, where vegetation is consumed by livestock, may simultaneously benefit species adapted to open grassy vegetation, and harm species adapted to closed, woody vegetation. Second, a key driver of change to animal populations—overharvesting—is uncorrelated or very weakly correlated with changes to NPP. Ecosystems with intact vegetation, and thus low HANPP, can be “empty” or largely devoid of birds and mammals, where hunting pressure is high but land clearance and logging is low. Third, there is no established mechanistic basis through which changes to animal abundances and functions can be inferred based on changes to HANPP; at best, efforts to use HANPP as a proxy for the animal component of a biosphere integrity planetary boundary would rely on correlations. For all these reasons, we suggest that an energetics-based measure of animal-mediated ecosystem function would complement HANPP by revealing a different aspect of biosphere change.

To test our assumption that HANPP is not an accurate proxy for changes to animal-mediated ecosystem functions, we used a linear regression to test the strength of the relationship between proportional HANPP and the proportional intactness of energy flows through birds and mammals. We expected to find a weak correlation between the two metrics, indicating their correlation with aggregate human impacts on a landscape.

We expected this relationship to be noisy, given the different ways plants and animals respond to human activity. As expected, we found a weak, noisy relationship between increasing HANPP and declining energetic intactness of birds and mammals (slope = -0.24, $p < 0.01$, $r^2 = 0.15$).

Figure S2. Relationship between HANPP and the intactness of energy flows through birds and mammals. Across all 8x8 km cells in sub-Saharan Africa, proportional human appropriated net primary productivity (HANPP) was related to the proportional intactness of energy flows through birds and mammals.

We suggest the weakness of the relationships is not only a consequence of the different methodologies used to map HANPP and biodiversity intactness, but also reflects the distinct ways animal versus plant energetics respond to human land use. Factors such as overharvesting, protected area management, and competition with livestock are likely to affect animals and the functions they perform in ways that diverge from their impacts on plant primary productivity. We therefore conclude that mapping changes to energy flows through animal functional groups is likely to provide useful information about changes to the earth system, information not captured by changes to HANPP. Integrating HANPP and animal energetics provides a way forward for setting a

planetary boundary for biosphere integrity that captures changes to both the plant and animal components of the biosphere. (See lines 957-967 for our discussion of the added value of estimating energy flows through animals).

- Fig 1: “energy flows through animal-mediated ecosystem functions” this is quite complex to grasp. Any way to simplify or better explain?

We’ve changed the figure title to “Intactness of energy flows through birds and mammals performing key ecosystem functions,” and added an additional sentence in the figure caption explaining that “ecosystem function intactness” is a shorthand for the intactness of energy flows through the birds and mammals that perform these ecosystem functions. (Lines 1138-1141).

- Fig 2: Does this include all land uses (so also agriculture and croplands)? And if so, do the differences tell something about the biological/ecological differences between biomes, or rather about differences between ‘biome’ in conversion rates or types of land use to which original vegetation is converted? In the latter case, it may not be very meaningful to compare across biomes.

The outer circles in Figure 2 show how trophic structures vary across historical, “intact” biomes. These represent the intrinsic differences between biomes’ physical structures and plant communities, which together affect the resources and habitat available for birds and mammals. By contrast, the inner circles represent the energy flows remaining after human impacts. Like the outer circles, the inner circles reflect the different distribution of trophic guilds between biomes. But as the reviewer points out, the inner circles also reflect the uneven conversion rates of the biomes (Fig. R1). In particular, 19% of grassy ecosystems have been converted to high intensity land uses, versus just 5% of forests and 2% of arid systems (Figure R2).

We believe it is useful for conservationists to see how land use change has altered the trophic structure of each biome. The biome has been proposed as the appropriate unit to assess biodiversity intactness, because biomes respond differently to human land use (Mace et al. 2014). Knowledge about the actual changes in each biome allows decision makers to prioritize conservation of the most threatened guilds in the most threatened biomes. For example, it adds to growing evidence that intact grassy ecosystems are disappearing more rapidly than forests (Stevens et al. 2022). The alternative would be to display a “corrected” figure, showing the theoretical changes to trophic structure in a scenario where each biome is converted equally. But we think such a theoretical figure would be less useful to decisionmakers than one showing how global change is actually occurring.

Figure R2 | The pace of land use conversion varies between major biomes. The proportion of cells in each land use in the year 2020 was estimated for each of the three major biomes.

- Fig 2. Given the large uncertainty for fossorial animals and their energetic dominance in arid and grassy systems, I'd argue that it would be important show uncertainties in the estimates. Another option for fossorial animals is to show a conservative estimate, based on species-specific densities of mole rats obtained from other sources? Reporting such very high values requires extra checking.

We've added a new figure showing the large uncertainties in the proportion of energy consumed by fossorial mammals (Extended data figure 6). We've also added an asterisk to the fossorial mammals guild in Figure 2, linked to a sentence noting the high uncertainty associated with the guild.

- Fig 4: Wouldn't it be informative to also show % change maps? I'm quite curious what they would look like. So, a row with three maps per animal group: historical, current and % change.

We have added a % change map as Supplementary Figure 2, which is reproduced below.

FIGURE REDACTED

Figure S2. | Intactness of energy flows through birds and mammals after human impacts. *Maps indicate the proportional intactness of (a) birds and (b) mammals. Fractional energetic intactness was calculated by dividing the current absolute energy flow through each 8x8 km grid cell by the historic absolute energy flow through each 8x8 km grid cell. Current absolute energy flows were calculated by multiplying the energy flows through each species present in a grid cell by the species' change in abundance, as estimated by the biodiversity intactness index for Africa.*

- Fig 4: What is the reason for the coarse patterns? It seems different layers are overlaid, while I would expect to see the cumulative value of all energy flows..

The cause of the coarse patterns differs between the maps showing historical or "intact" energy flows, and the maps showing current or "human-transformed" energy flows. For the intact maps, only the mammal map shows this "layering" affect. This is because a few keystone species (e.g. elephant, plains zebra, rhinos, fossorial rodents) historically contributed disproportionately to total energy flow through mammals. The referee is right that the maps show the cumulative values of all energy flows. But because a few mammals contribute a substantial portion of the cumulative values, the ranges of these keystone species appear as layers on the map.

For the current maps, the layering effect reflects the land use map. A land use map was used to estimate the declines in species abundances, following their values in the

Biodiversity Intactness Index (Clements et al. 2024). Land uses often change abruptly, for example when a protected area abuts a heavily farmed landscape. In these instances, species abundances change abruptly as well, and thus energy flows through species appear to fall off in layers. (See lines 1241-1247 for text on the BII maps).

- Fig 4: I miss values and units for the energy flows, which also makes comparison between historical and current difficult (or actually impossible).

We've added units to these figures.

Referee #4

The authors draw from a large amount of data coupled to allometric consumption rates, to argue that the ecosystem energetics that run the sub-Saharan African ecosystems have been fundamentally altered compared to historical times (1700 CE). The authors use these energetics estimates of consumptive flow as an argument for declining ecosystem function and a link to disrupted structure-function relationships. I applaud the authors for this work and largely like the direction (the role of structure-function and energetics to sustainability management is critical) but have some concerns of whether it meets the bar for a Nature publication. I do think that this type of paper would push people to further develop energetics within an ecosystem management perspective. My comments are below, some more curiosity questions.

We thank the referee for highlighting the critical role that clarifying ecosystem energetics and structure-function relationships can play in sustainable management. We have reframed the paper's discussion section to clarify the novelty and importance of our energetics lens. Below, in the next section of this response to the referees, we have outlined the key points that explain why we think this energetics approach is novel and important enough to meet the bar for a nature publication. We have also added a new conclusion outlining these points (Lines 928-980)

Major comments

1). My first issue and one that is more serious potentially is that while I am sympathetic to bringing in energy flow and structure-function relationships I think that this paper needs to prove that we get somewhere novel with an energetics application. That is, that when we apply the energetics we learn something that we do not get from biodiversity and changes in density alone. The authors try to do this but to me it needs to be executed more clearly and I feel as stands not confident that they have done this. For example, when I see all the results it seems to me that the results are really just a simple "scaling up" of the "functional diversity" and density results. As a specific example, arboreal species reduction in energy flow in forests are merely the scalar response of a reduction in arboreal species relative densities and diversity. It is fine to say that this impacts energy flow but this feels quite straightforward and implicitly known. As an example, would the diversity*density plots of Figure 2 be identical

qualitatively to the flow plots in figure 2? The lessons for ecosystem restoration on lines 320-338 seem to me could all be done from changes in functional diversity and may not require energetics at all.

We have added a new, expanded conclusion (Lines 928-980) that more clearly justifies the novelty and significance of our energetics approach. In it we present three arguments outlining the advantages of using energetics: one argument each at the scale of the species, the ecosystem, and the earth system. We also explain why calculating energy flows presents a unique opportunity to relate ecosystem function across these three scales using a quantitative, ecologically meaningful metric. We summarize these points about novelty and significance in Lines 119-124 of our introduction as well.

In addition, we have added a methods figure (Extended Data Fig. 7) to illustrate the multi-step process required to estimate energy flows from population densities. We believe that this figure helps to clarify that estimating ecosystems energetics is more complicated (and informative) than simply “scaling up” changes in population density and functional diversity.

We argue that the core value of an ecosystem energetics approach is that it quantitatively estimates how species are affecting an ecosystem through their food consumption. The approach measures the amount of kilojoules (i.e. calories) each species consumes per square kilometer per year ($\text{kJ m}^{-2} \text{yr}^{-1}$). The result is a physically meaningful metric, which quantifies how much grazing, browsing, carnivory, granivory, or insectivory each species does in an ecosystem, in a unit that is transferable between any species or landscape. Knowing the magnitude of these functions allows ecologists to quantify how species impacts vegetation, trophic structure, nutrient dispersal, and other ecosystem processes. Ecologists can also compare energy flows through animals to energy flows through plants (net primary productivity), or to energy conserved in fossil fuels, or to the total energy that reaches the Earth from the Sun, allowing ecologists to estimate animals' relative impact on earth system processes. By contrast, when ecologists estimate changes using species biodiversity and density alone, they produce a non-fungible metric, which cannot be quantitatively compared to other ecosystem processes. While these metrics can tell ecologists about species biomass or diversity, it is unclear what aspect of ecosystem function these metrics capture if any.

Compared to approaches based on scaling population densities and functional diversity, an energetics approach incorporates more information, which results in a much better approximation of how species actually impact ecosystems. The energetics approach not only scales up population densities, but also weights individuals of each species based on their food consumption, which is itself a function of each individual's body mass and diet composition, as well as the digestibility of diet components, and the assimilation efficiency of each species' digestive system. Scaling approaches that ignore this additional data provide misleading results. For example, a scaling approach that weights species based on their population density, (which we think is what the referee is proposing here), would dramatically overstate the ecological importance of high-density,

small-bodied species such as rodents and passerines. To account for this issue, one could instead scale up species based on their biomass, by multiplying population densities by body mass. But this would create the opposite problem, overrating the importance of large species, because it would not account for smaller species' lower energetic efficiency, which causes them to consume more food per unit mass. In addition, neither of the aforementioned approaches would account for the important differences in food assimilation efficiency between food types and animal digestive systems, factors that dramatically impact the total amount of energy a species consumes in an ecosystem, and thus a species' impacts on an ecosystem's vegetation and trophic structure.

Elephants, Africa's most energy-consuming wild mammal, provide a clarifying example. Elephants consume almost entirely leaves, from trees, shrubs, and grass. On average, leaves have a much lower assimilation efficiency (~40%) than other food sources (70-90%). But this assimilation efficiency varies widely based on herbivores' digestive systems: ruminants assimilate ~60-70% of leaf energy, perissodactyls and small herbivores (e.g. rabbits) around 40%, and elephants, which are highly inefficient digesters, just 20%! If elephant energy consumption was estimated via a simple scaling process, without accounting for elephants' anomalously low digestion efficiency and the low digestibility of leaves, it would lead to a three-to-fourfold underestimate in elephants' total energy consumption. Such a result would dramatically underrate elephants' contribution to ecosystem function through their grazing, browsing, and nutrient dispersal, and ultimately understate the impact of elephants (and of their extirpation) on landscapes, biomes, and the earth system.

See introduction and conclusion (Lines 119-124 and 928-980).

2). My second comment is on structure and function, I was modestly confused with the use of these key terms. Food web ecologists might argue that structure of an ecosystem comes from i) the species interactions (topology), and ii) the strength of these interactions (defined differently but often with energetic flow more recently). The function comes from this as primary productivity, secondary productivity, nutrient cycling, oxygen levels and so on. I think the authors here are effectively suggesting they are measuring secondary production defined as consumed carbon/biomass after conversion of prey into new biomass. If they are, then this should be made clear. I think a box with some key definitions might go a long way in this including: structure, function, biodiversity intactness, energetic intactness, energetic vibrancy, resilience (used a few times and not clear exactly).

We have edited the text so that we use the term "structure" to refer exclusively to ecosystems' trophic structures, which we define as the partitioning of energy and biomass between plant and animal guilds. The referee is correct that this measure of structure is synonymous with the distribution of secondary production among animal guilds. As the referee rightly points out, our use of the term structure differs from its use by food web ecologists, who are instead concerned with the strength of interactions

between species or groups, a concern that falls outside the scope of this paper. We use ecosystem functions to refer to how mammal and bird activity impacts ecosystem patterns and processes, including vegetation cover, plant-animal interactions, trophic structure, trophic interactions, fire regime, biogeochemical cycling, and climate. (Lines 68-80, 210)

3). The body size shifts in the African ecosystems is very interesting and consistent with other work. I wonder if there are relationships with body size, density, diversity and energy flow? It might be interesting to show us how body size across species has been impacted in these systems and their relationship to secondary productivity or energy flow as the authors point out briefly somewhere, from allometric patterns in consumption rates, we would expect here, that energetic impacts may have a significant changing impact (small species). I was wondering if small species are declining too, just their relative amounts are increasing. If so, this should be made clear.

Good questions! We have taken your suggestion, and analyzed how body size impacts changes in energy flows. We present the results in the main text (lines 496-503; Extended Data Table 2), as well as here. We found that on average, across sub-Saharan Africa, the proportion of energy consumed by small birds and mammals (<3 kg) rose from 69% of total energy to 78% of total energy, while the proportion consumed by megafauna (>65kg) fell from 16% to 7%. In croplands, the proportion of energy consumed by large mammals from 16% to 2%, a dramatic shift in favor of birds and small mammals.

Extended Data Table 2 | *Changes in energy flow through species groups based on body mass and land use.*

Land Use	Size Class	Energy - Historical (KJ * m-2 * yr-1)	Energy - Current (KJ * m-2 * yr-1)	Proportion of Energy-Historical	Proportion of Energy-Current
all	large	685	186	16%	7%
all	medium	636	398	15%	15%
all	small	1818	1314	43%	49%
all	smallest	1061	776	25%	29%
all	all	4200	2675	100%	100%
Croplands	large	728	45	16%	2%
Croplands	medium	722	248	16%	13%
Croplands	small	1978	979	44%	53%
Croplands	smallest	1086	581	24%	31%
Croplands	all	4514	1853	100%	100%
Protected Lands	large	858	499	17%	11%
Protected Lands	medium	657	613	13%	14%
Protected Lands	small	2269	2151	46%	49%
Protected Lands	smallest	1143	1095	23%	25%

Protected Lands	all	4928	4358	100%	100%
Settlements	large	672	24	15%	2%
Settlements	medium	671	129	15%	11%
Settlements	small	1993	644	44%	53%
Settlements	smallest	1220	414	27%	34%
Settlements	all	4557	1211	100%	100%
Untransformed Lands	large	665	188	16%	7%
Untransformed Lands	medium	619	410	15%	15%
Untransformed Lands	small	1756	1315	43%	49%
Untransformed Lands	smallest	1050	789	26%	29%
Untransformed Lands	all	4090	2702	100%	100%

Species were allocated into body mass classes according to the following thresholds: smallest (<0.5 kg), small (< 3kg), medium (<65 kg), large (> 65 kg). The thresholds between the small, medium, and large groups follow Pringle et al., 2023. The threshold between the small and smallest groups was added to provide additional detail.

4. I wonder if the authors could pull out more clearly winners and losers with this change. Often, with ecosystem change there are some species that take over. In this paper, it appears to be the argument by the authors that there is only largely an overall decline although there are modest suggestions of increases of some species. I was left wondering about this. Making this more clear would be useful. The authors discuss an example with logging accelerating vertebrate consumption and suggest this as a possibility. It would be nice to know if they found such anomalies.

Thank you for another excellent suggestion. We have undertaken an analysis of the relative winning and losing ecosystem functions, and present the results in Extended Data Figure 5 (and below). Only one ecosystem function experienced an absolute increase in energy flow under land use change: in croplands, energy flows through bird-mediated granivores increased to above their historic levels. But many ecosystem functions, particularly those mediated by birds and small mammals, were relative winners, they accounted for a higher proportion of total energy flows under human land use than under historical, pre-industrial/colonial conditions (see heatmap figure below). The distribution of winning and losing ecosystem functions varied substantially by land use. For example, the proportion of energy flowing through perching insectivorous birds increased by 7% overall after land use change, and by 27% in settlements, even as it decreased by 32% in croplands. We believe that empirically verifying these patterns through fieldwork, in an effort to reveal their causes, would be a rich area for future study. (Lines 1609-1618, 441-448)

Ecosystem Function	Human Land Use Category				
	All	Protected Lands	Untransformed Lands	Croplands	Settlements
Aerial insectivory (B)	24.74%	9.35%	14.86%	16.97%	80.40%
Grazing (B)	18.06%	9.12%	22.17%	66.15%	41.61%
Aquatic carnivory (B)	16.57%	8.11%	13.03%	30.93%	40.51%
Granivory (M)	16.15%	10.23%	10.83%	85.31%	17.56%
Soil formation (M)	15.94%	6.60%	42.64%	75.70%	-2.91%
Granivory (B)	15.18%	6.57%	5.66%	161.28%	75.49%
Insectivory (M)	15.01%	10.22%	9.33%	9.61%	-0.60%
Terrestrial insectivory (B)	14.61%	7.81%	10.22%	42.03%	34.50%
Meso-carnivory (M)	10.25%	10.95%	4.13%	11.37%	-23.42%
Ecosystem engineering (B)	6.87%	8.11%	0.57%	-38.57%	40.51%
Pollination (B)	6.85%	7.53%	-2.93%	-36.29%	56.39%
Perching insectivory (B)	6.45%	7.21%	-1.46%	-31.93%	26.99%
Seed dispersal (B)	4.64%	4.52%	-5.33%	-54.78%	36.80%
Carnivory (B)	2.45%	3.44%	0.14%	21.64%	9.81%
Canopy folivory (M)	-1.10%	7.93%	-13.17%	-86.97%	-36.02%
Scavenging (B)	-4.25%	1.03%	-4.93%	2.94%	-49.96%
Seed dispersal (M)	-5.38%	3.97%	-15.17%	-68.87%	-32.07%
Scavenging (M)	-10.57%	-3.76%	-5.20%	39.10%	-33.61%
Apex carnivory (M)	-43.43%	-19.70%	-45.90%	-47.07%	-80.11%
Grazing (M)	-58.25%	-34.70%	-56.57%	-71.72%	-86.94%
Browsing (M)	-58.62%	-31.47%	-58.73%	-82.11%	-86.00%
Nutrient cycling (M)	-59.76%	-34.08%	-59.85%	-81.56%	-87.21%
Megaherbivore impacts (M)	-60.75%	-33.38%	-61.36%	-79.48%	-86.55%

Extended Data Fig. 5 | Winning and losing ecosystem functions after human impacts. Values represent the change in the proportion of energy flowing through each ecosystem function after accounting for human land use. Values were calculated proportionally, as the percentage change to the historical proportion of energy that flowed through each ecosystem function. Only one function, avian granivory in croplands, experienced an absolute increase in energy flow compared to historical values. But many functions experienced a relative increase in importance, even as absolute energy flows through those ecosystem functions declined. Bluer cells indicate “winning” functions, which experienced increasing relative importance, while redder cells indicate “losing” which experienced decreasing relative importance.

From an ecosystem stability perspective strong singular energy pathways would likely contribute to instability and so while high diversified energy flow might be increased function, high singular energy flows might increase secondary production but at the peril

of lost ecosystem stability. This is an issue ignored by the authors, and is fine, but shows how increased fluxes may not be always a positive to a restored ecosystem.

We have added language addressing this point in the caveats section of the main text (Lines 872-879)

Anyway, I liked the direction of this paper and feel this has potential. I was left wondering if they really had showed us that energetics pushed the bar of sustainability and impact a long way. If so, it needs to be brought out more strongly and coherently otherwise this paper fits in a more specific environmental journal.

We have added a new, expanded conclusion that more clearly justifies the novelty and significance of our energetics approach. We have also summarized these key justifications in the paper's introduction (Lines 928-980).

References

- Antunes, Ana Carolina, Emilio Berti, Ulrich Brose, Myriam R. Hirt, Dirk N. Karger, Louise M. J. O'Connor, Laura J. Pollock, Wilfried Thuiller, and Benoit Gauzens. 2024. "Linking Biodiversity, Ecosystem Function, and Nature's Contributions to People: A Macroecological Energy Flux Perspective." *Trends in Ecology & Evolution* 39 (5): 427–34. <https://doi.org/10.1016/j.tree.2024.01.004>.
- Beyer, Robert M., and Andrea Manica. 2020. "Historical and Projected Future Range Sizes of the World's Mammals, Birds, and Amphibians." *Nature Communications* 11 (1): 5633. <https://doi.org/10.1038/s41467-020-19455-9>.
- Clements, Hayley S., Emmanuel Do Linh San, Gareth Hempson, Birthe Linden, Bryan Maritz, Ara Monadjem, Chevonne Reynolds, Frances Siebert, Nicola Stevens, and Reinette Biggs. 2024. "The Bii4africa Dataset of Faunal and Floral Population Intactness Estimates across Africa's Major Land Uses." *Scientific Data* 11 (1): 191.
- Daskin, Joshua H., Marc Stalmans, and Robert M. Pringle. 2016. "Ecological Legacies of Civil War: 35-Year Increase in Savanna Tree Cover Following Wholesale Large-Mammal Declines." *Journal of Ecology* 104 (1): 79–89. <https://doi.org/10.1111/1365-2745.12483>.
- Doughty, Christopher E., Adam Wolf, and Yadvinder Malhi. 2013. "The Legacy of the Pleistocene Megafauna Extinctions on Nutrient Availability in Amazonia." *Nature Geoscience* 6 (9): 761–64. <https://doi.org/10.1038/ngeo1895>.
- Eldridge, David J., and Santiago Soliveres. 2023. "Rewilding Soil-Disturbing Vertebrates to Rehabilitate Degraded Landscapes: Benefits and Risks." *Biology Letters* 19 (4): 20220544. <https://doi.org/10.1098/rsbl.2022.0544>.
- Faurby, S., and J.-C. Svenning. 2015. "Historic and Prehistoric Human-Driven Extinctions Have Reshaped Global Mammal Diversity Patterns." *Diversity and Distributions* 21 (10): 1155–66. <https://doi.org/10.1111/ddi.12369>.

- Haberl, Helmut, Karl-Heinz Erb, and Fridolin Krausmann. 2014. "Human Appropriation of Net Primary Production: Patterns, Trends, and Planetary Boundaries." *Annual Review of Environment and Resources* 39 (1): 363–91. <https://doi.org/10.1146/annurev-environ-121912-094620>.
- Hemming, Victoria, Mark A. Burgman, Anca M. Hanea, Marissa F. McBride, and Bonnie C. Wintle. 2018. "A Practical Guide to Structured Expert Elicitation Using the IDEA Protocol." Edited by Barbara Anderson. *Methods in Ecology and Evolution* 9 (1): 169–80. <https://doi.org/10.1111/2041-210X.12857>.
- Hempson, G. P., S. Archibald, and W. J. Bond. 2015. "A Continent-Wide Assessment of the Form and Intensity of Large Mammal Herbivory in Africa." *Science* 350 (6264): 1056–61. <https://doi.org/10.1126/science.aac7978>.
- Lumbierres, Maria, Prabhat Raj Dahal, Carmen D. Soria, Moreno Di Marco, Stuart H. M. Butchart, Paul F. Donald, and Carlo Rondinini. 2022. "Area of Habitat Maps for the World's Terrestrial Birds and Mammals." *Scientific Data* 9 (1): 749. <https://doi.org/10.1038/s41597-022-01838-w>.
- Mace, Georgina M., Belinda Reyers, Rob Alkemade, Reinette Biggs, F. Stuart Chapin, Sarah E. Cornell, Sandra Díaz, et al. 2014. "Approaches to Defining a Planetary Boundary for Biodiversity." *Global Environmental Change* 28 (September):289–97. <https://doi.org/10.1016/j.gloenvcha.2014.07.009>.
- Malhi, Yadvinder, Christopher E. Doughty, Mauro Galetti, Felisa A. Smith, Jens-Christian Svenning, and John W. Terborgh. 2016. "Megafauna and Ecosystem Function from the Pleistocene to the Anthropocene." *Proceedings of the National Academy of Sciences* 113 (4): 838–46. <https://doi.org/10.1073/pnas.1502540113>.
- Malhi, Yadvinder, Terhi Riutta, Oliver R. Wearn, Nicolas J. Deere, Simon L. Mitchell, Henry Bernard, Noreen Majalap, et al. 2022. "Logged Tropical Forests Have Amplified and Diverse Ecosystem Energetics." *Nature* 612 (7941): 707–13. <https://doi.org/10.1038/s41586-022-05523-1>.
- Pedersen, Rasmus Østergaard, Søren Faurby, and Jens-Christian Svenning. 2023. "Late-Quaternary Megafauna Extinctions Have Strongly Reduced Mammalian Vegetation Consumption." *Global Ecology and Biogeography* 32 (10): 1814–26. <https://doi.org/10.1111/geb.13723>.
- Powers, Ryan P., and Walter Jetz. 2019. "Global Habitat Loss and Extinction Risk of Terrestrial Vertebrates under Future Land-Use-Change Scenarios." *Nature Climate Change* 9 (4): 323–29. <https://doi.org/10.1038/s41558-019-0406-z>.
- Pranzini, N., S. Bertolino, and L. Santini. 2023. "Predicting Population Size at Large Scale: The Case of Two Large Felids." *Global Ecology and Conservation* 48 (December):e02677. <https://doi.org/10.1016/j.gecco.2023.e02677>.
- Pringle, Robert M., Joel O. Abraham, T. Michael Anderson, Tyler C. Coverdale, Andrew B. Davies, Christopher L. Dutton, Angela Gaylard, et al. 2023. "Impacts of Large Herbivores on Terrestrial Ecosystems." *Current Biology* 33 (11): R584–610. <https://doi.org/10.1016/j.cub.2023.04.024>.
- Richardson, Katherine, Will Steffen, Wolfgang Lucht, Jørgen Bendtsen, Sarah E. Cornell, Jonathan F. Donges, Markus Drüke, et al. 2023. "Earth beyond Six of Nine Planetary Boundaries." *Science Advances* 9 (37): eadh2458. <https://doi.org/10.1126/sciadv.adh2458>.

- Ripple, William J., Guillaume Chapron, José Vicente López-Bao, Sarah M. Durant, David W. Macdonald, Peter A. Lindsey, Elizabeth L. Bennett, et al. 2017. "Conserving the World's Megafauna and Biodiversity: The Fierce Urgency of Now." *BioScience*, January, biw168. <https://doi.org/10.1093/biosci/biw168>.
- Santini, Luca, Ana Benítez-López, Carsten F. Dormann, and Mark A. J. Huijbregts. 2022. "Population Density Estimates for Terrestrial Mammal Species." *Global Ecology and Biogeography* 31 (5): 978–94. <https://doi.org/10.1111/geb.13476>.
- Sekercioglu, Cagan H. 2006. "Increasing Awareness of Avian Ecological Function." *Trends in Ecology & Evolution* 21 (8): 464–71. <https://doi.org/10.1016/j.tree.2006.05.007>.
- Stevens, Nicola, William Bond, Angelica Feurdean, and Caroline ER Lehmann. 2022. "Grassy Ecosystems in the Anthropocene." *Annual Review of Environment and Resources* 47.
- Tucker, Marlee A., Luca Santini, Chris Carbone, and Thomas Mueller. 2021. "Mammal Population Densities at a Global Scale Are Higher in Human-Modified Areas." *Ecography* 44 (1): 1–13. <https://doi.org/10.1111/ecog.05126>.

We thank the referees for reviewing this paper again, and for their largely positive feedback. Below we go through each of the comments in turn and give our responses in blue italics.

Reviewer 1:

I thank the authors for their clear responses. Many of the arguments made in their response are sensible and mostly convincing. I would like to see some of the arguments more fully included in the main text - much of this is done, which I'm pleased to see, but some still needs to be fully addressed.

We are glad you found our responses clear and largely convincing. We are glad to include more of our response in the main text, bearing in mind limits on text length.

For example, the authors suggest that even if the estimates of historic abundance are well off, the ball park results will be correct. However, I still look at figure ED3 and see that almost any rank order arrangement of species could be supported by the data - with the possible exception of elephants remaining high. I think the confidence intervals (which must be independent by species) mean that in any one MCMC draw almost any rank order is supported - this stochastic ordering of most species is presumably why such regionally restricted species as Salt's Dikdik appear in the figure but, say, Hippo do not. The only thing that I can rule out confidently is that elephant will remain in the top 10 of all such realisations. To convince all readers this is not important, I think you need to convince in the main text that averaging (adding) within groups makes this less rather than more certain, allowing greater confidence at the functional group level than at the individual species level.

We thank the referee for their good catch on Salt's dik dik. We went back and checked this result, and found that the arid and grassy headings on the sub-figures were switched. While Salt's dik dik is range-restricted across Africa, it occurs widely and at high densities in the Horn of Africa, sub-Saharan Africa's largest arid region. (The Sahel is classified as a grassy region, and the Sahara is excluded from the analysis). So the dik dik's inclusions in the list is perhaps less anomalous than it initially appears.

We agree with the referee's broader point, however: that aside from elephants, species-level energy consumption is associated with such high uncertainty that it makes any species-level analysis unhelpful. We've added clearer text in the methods paragraphs of the main text (Lines 161-162) and in the caveats section of the main text (Lines 449-450) to emphasize that (a) uncertainty around energy flow size declines as a greater number of species are analyzed together and that (b) this paper should therefore be used to understand energy flows through functional groups, guilds, and body size classes rather than to compare energy flows through species.

I also thank the authors for being more candid about the way they are estimating the intactness of functional groups - this was a concern I raised that another reviewer

managed to put their finger on more explicitly. The answer - involving making the assumption that every species contributes equally and there is a 1 to 1 relationship between energy and contribution - makes me question again the value of including this. I think it is so unlikely this assumption holds (indeed, we have lots of evidence that it isn't true at least over the short term in any given ecosystem), I would still suggest cutting and retaining the core story about energy. I think the ecosystem function add-on is something of a distraction from the core message, built on even shakier foundations.

We understand the referee's concern that there is often not a 1:1 relationship between the energy consumption of a group of species and the ecosystem function those species perform. We believe that we made this limitation much clearer in the text, in response to the first round of referee comments (see Lines 151-160, 963-971), . We note that there is a 1:1 relationship between energy consumption and dietary functions, which include grazing and browsing, insectivory, meso-carnivory, granivory, scavenging, and apex carnivory: 6 of the 10 major functions included in the paper.

While we understand this concern, we are hesitant to eliminate the discussion of ecosystem functions, because we think that an energetics approach, while imperfect, substantially advances ecologists' ability to relate biodiversity decline to animal-mediated functions. In particular we provide a pathway for researchers to combine energetics data with animal behavior data, and thereby refine our estimates of how much each species contributes to ecosystem functions. We acknowledge that this paper does not resolve the problem of quantifying animal-mediated functions at scale, but we think it is a major step toward resolving that problem. And we fear that if we eliminated our discussion of ecosystem function, we would understate one of the key novel benefits that an energetics approach brings: it's use as a quantitative metric that can relate biodiversity change, and changes in species abundance, to animal-mediated ecosystem functions, ideally in a model that also includes clarifying information on animal behavior. We hope that this paper opens a pathway for researchers in Africa to launch studies aimed at quantifying animal-mediated ecosystem functions. By eliminating our discussion of function, we fear we would obscure that pathway.

Reviewer 2:

Overall, my comments have been well addressed. The authors have reworked their estimates of species' distributions by adding a step of calculating Area-of-Habitat. Their updated approach draws on the work of Beyer and Manica (2020), with Beyer now included as a co-author. A comparison of use of "raw" vs. refined (area of habitat) is convincing in demonstrating that their findings are relatively robust in this regard.

Some specific feedback:

1. Summary figure of methodological steps: this is well done and very helpful

Thank you.

2. Caveats: New section is fine. First line (411) – the framing suggests that caveats are disadvantageous, rather than fundamental aspects of modelling. I suggest rewording it.

Good advice, thank you. We have reworded this sentence. (Line 413)

3. Ln 702: “For a few small-range species, 1/2° scale habitat-adjusted range maps eliminated all available range; for these species,”- Surely the reason for the elimination in the larger grids was due to exclusion from the 1/12 degree grid cells?

You are correct. We have added this explanation to the methods. (Lines 707-709)

4. There is a more recent reference that should inform this: <https://pubmed.ncbi.nlm.nih.gov/31324345/>

Thanks. We would be happy to swap in this reference, but we are not sure which reference you are referring to as outdated.

5. Fig S1: typo in header a.

Fixed.

Reviewer 3:

I have assessed the responses of the authors to the comments of all reviewers, and read the manuscript and additional new materials.

First, I'd like to reiterate what I stated in my review on the original submission (I was reviewer #3): "Overall, the study convincingly shows that an energetics approach can be applied at continental scale to quantify anthropogenic changes in ecological processes and is a valuable addition to other metrics (e.g., carbon stock, NPP, GPP) because it explicitly considers the role of animals." This is still my overall assessment. To this I'd like to add that I find this study to be a valuable contribution to Earth System studies. The revised paper is more balanced than the initial submission: caveats and limitations are better explained and illustrated, more analyses of robustness and sensitivities are now included, and I value the additional methodological discussion in the SI.

Thank you for the positive feedback.

Second, my comments are satisfactorily addressed. I particularly like and value the additional analyses done to address comments #3 and 4. For comment #1 (on the contribution of habitat loss/conversion to shifts in energy flows), I would suggest the authors to include their estimate that "conversion ... is responsible for 25% of the total decline in energy flows" (quote from the rebuttal) in the main text. This is an important addition that helps interpreting the findings.

We have added this sentence into the results section. (Lines 173-174).

Third, I feel that concerns of other reviewers have also been adequately addressed in the rebuttal and revision. In particular, I find the responses to the concerns of reviewer #1 about the poor quality of the African datasets to be convincing. Plus, I'd like to add to this discussion that the fact that changes in African biomes are so poorly studied is -- in my view -- actually a strong justification for conducting this type of studies there. Of course, while taking uncertainties into account. I truly hope the paper --when published -- will create opportunities for energetics (and similar) studies at African research institutes and organizations. This type of studies have the potential to stimulate new studies, and help improving the data quality situation in understudied regions.

We agree! And we also hope this study helps launch further studies on ecosystem energetics, animal ecosystem function, and biodiversity loss and gain in Africa.

Finally, some minor (textual) comments:

- Fig 2 caption: the explanation what the asterisk for fossorial mammals is missing
- Fig S2: I'd be in favour of making this Figure more visible by including it as an Extended Data Fig
- L 341-342: sentence is a fragment, incomplete
- L 376: citation missing ("CITE")
- L 402 (around): could you refer to the S Fig with the poor association of HANPP vs energy?

We have fixed all these errors and added a reference to the S Figure about HANPP. (Line 404)

Reviewer 4:

I have read the authors comments to all reviewers. The authors have very thoroughly responded to all queries and I feel the paper has greatly improved from this process. I, like some of the reviewers, remain a little wary of the data and the layers of assumptions but believe the overall goal of injecting an "energetics" lens on conservation is an important step forward and hopefully a paper like this will charge conservation with a more holistic energetics perspective.

Thank you. We hope so too!